# Mitochondrial membrane hyperpolarization modulates nuclear DNA methylation and gene expression through phospholipid remodeling

Mateus Prates Mori [1], Oswaldo A. Lozoya [2], Ashley M. Brooks[3], Carl D. Bortner[4], Cristina A. Nadalutti[1], Birgitta Ryback[5], Brittany P. Rickard [6], Marta Overchuk[7], Imran Rizvi[7,8], Tatiana Rogasevskaia[9], Kai Ting Huang[10], Prottoy Hasan[10], György Hajnóczky [10] & Janine H. Santos [1] ✉

Maintenance of the mitochondrial inner membrane potential (ΔΨm) is critical for many aspects of mitochondrial function. While ΔΨm loss and its consequences are well studied, little is known about the effects of mitochondrial hyperpolarization. In this study, we used cells deleted of *ATP5IF1* (IF1), a natural inhibitor of the hydrolytic activity of the ATP synthase, as a genetic model of increased resting ΔΨm. We found that the nuclear DNA hypermethylates when the ΔΨm is chronically high, regulating the transcription of mitochondrial, carbohydrate and lipid genes. These effects can be reversed by decreasing the ΔΨm and recapitulated in wild-type (WT) cells exposed to environmental chemicals that cause hyperpolarization. Surprisingly, phospholipid changes, but not redox or metabolic alterations, linked the ΔΨm to the epigenome. Sorted hyperpolarized WT and ovarian cancer cells naturally depleted of IF1 also showed phospholipid remodeling, indicating this as an adaptation to mitochondrial hyperpolarization. These data provide a new framework for how mitochondria can impact epigenetics and cellular biology to influence health outcomes, including through chemical exposures and in disease states.

The mitochondrial inner membrane potential (ΔΨm) is primarily maintained by the proton pumping activity of the electron transport chain (ETC) although it can rely on ATP hydrolysis by reverse rotation of the F1 subunit of the ATP synthase under conditions of stress such as ischemia or ETC inhibition[1]. Maintenance of the ΔΨm is critical for mitochondrial function as protein import, ion homeostasis, and many other biochemical processes that occur in these organelles largely depend on it[2–4]. There is a vast body of work describing the

[1]Mechanistic Toxicology Branch, Division of Translational Toxicology, National Institute of Environmental Health Sciences (NIEHS), National Institutes of Health (NIH), Durham, NC, USA. [2]Genome Integrity and Structural Biology Laboratory, National Institute of Environmental Health Sciences (NIEHS), National Institutes of Health (NIH), Durham, NC, USA. [3]Biostatistics and Computational Biology Branch, Integrative Bioinformatics Support Group, National Institute of Environmental Health Sciences (NIEHS), National Institutes of Health (NIH), Durham, NC, USA. [4]Flow Cytometry Center, National Institute of Environmental Health Sciences (NIEHS), National Institutes of Health (NIH), Durham, NC, USA. [5]Dana Farber Cancer Institute, Harvard Medical School, Boston, MA, USA. [6]Curriculum in Toxicology & Environmental Medicine, University of North Carolina (UNC), Chapel Hill, NC, USA. [7]Department of Biomedical Engineering, North Carolina State University, Raleigh, NC, USA. [8]Lineberger Comprehensive Cancer Center, UNC, Chapel Hill, NC, USA. [9]Department of Biology, Mount Royal University, Calgary, AB, Canada. [10]MitoCare Center, Department of Pathology and Genomic Medicine, Thomas Jefferson University, Philadelphia, PA, USA. ✉e-mail: janine.santos@nih.gov

consequences of loss of the ΔΨm, which can range from mitochondrial dysfunction (e.g., caused by loss of protein import, ATP production, or Ca²⁺ homeostasis) to a signal for removal through mitophagy. Conversely, little is known about the effects associated with increases in the ΔΨm. In pathology, higher resting ΔΨm was reported in smooth muscle cells isolated from patients with pulmonary hypertension relative to matched healthy controls and in rodent models of the disease[5]; it was also found in glioblastoma[6] and ovarian cancer[7]. It remains unknown whether the increased resting ΔΨm contributes to, or is a secondary effect of, disease pathophysiology.

At the organellar level, studies using isolated mitochondria showed that membrane hyperpolarization promotes reactive oxygen species (ROS) production, which seems threshold-dependent[8]. Higher ΔΨm is known to facilitate mitochondrial Ca²⁺ uptake, activating bioenergetics by its effects on tricarboxylic acid (TCA) cycle dehydrogenases[9]. At the cellular level, deletion of IF1 (or ATPI-F1–*ATP5IF1* official gene symbol), a natural inhibitor of the hydrolase activity of the ATP synthase[10,11], increased cell viability caused by ETC inhibition and supported cell proliferation during mitochondrial DNA (mtDNA) loss, conditions that depolarized the ΔΨm in wild-type (WT) cells but not in the KO counterparts[12,13]. Most recently, the ΔΨm was shown to impact cell cycle progression[14], and to be modulated by intracellular stimuli. For instance, the growth of cells in phosphate-depleted conditions led to mitochondrial hyperpolarization in yeast and in mammalian cells. Mechanistically, this effect was shown to be partially dependent on the ETC, and to restore mitochondrial protein import in a mutant yeast strain[15]. This collective body of work indicates that the ΔΨm itself can affect diverse biological outcomes. Also, they indicate that polarization of the mitochondrial membrane might be responsive to intracellular signals and environmental cues, leading to the hypothesis that it can be modulated to influence cellular outcomes. Nonetheless, how cells respond and adapt to a chronic rise in ΔΨm remains largely unknown.

In this study, we used cells genetically depleted of IF1 as a model to study the effects associated with a chronic rise in resting ΔΨm. Our data highlight the pervasive molecular, metabolic, and genomic changes associated with sustained mitochondrial hyperpolarization, which might be relevant in the response to environmental exposures and in disease states. As such, it can be potentially targeted to improve health outcomes.

## Results

### Chronic loss of IF1 supports a model of increased resting ΔΨm

To define the suitability of using cells depleted of IF1 as a model of chronic mitochondrial hyperpolarization, we started by characterizing the resting ΔΨm of HEK293 IF1-KO relative to the isogenic WT counterparts[13]. Using intact cells and the fluorescent dye TMRE (tetramethylrhodamine ethyl ester) that loads into mitochondria based on the ΔΨm and normalizing the data to MitoTracker Green (MTG), which accumulates in mitochondria independent of ΔΨm, we found that IF1-KO cells had higher resting ΔΨm than WT controls (Fig. 1A). Similar results were observed when deleting IF1 in an independent cell line (Supplementary Fig. 1A, B). Mitochondrial hyperpolarization was confirmed by evaluating permeabilized cells, in which the plasma membrane was removed, and the organelle was energized by adding the complex II substrate succinate. In these experiments, we followed ΔΨm using the fluorescent indicator tetramethylrhodamine methyl ester (TMRM), and concomitantly, the clearance of Ca²⁺ added to the cytoplasmic buffer with FuraFF. The ΔΨm was higher in KO relative to WT cells (Fig. 1B upper panel), which was paralleled by faster clearance of cytosolic Ca²⁺ into the mitochondria (Fig. 1B lower panel). Unlike previously reported[16], we found no changes at the protein levels of the different subunits of the mitochondrial Ca²⁺ uniporter complex (mtCU), which mediate the import of Ca²⁺ in these organelles (Supplementary Fig. 1C); we also did not find changes in total or

phosphorylated AMPK (Fig. S1D). Our data are thus consistent with the higher ΔΨm facilitating the faster uptake of cytosolic Ca²⁺ into the mitochondria of the IF1-KO cells.

IF1 was thought to bind to the ATP synthase to inhibit its hydrolase activity only under stress conditions, such as hypoxia or inhibition of the ETC. However, more recent work suggests that heterogenous populations of the enzyme engaged in either ATP synthesis or hydrolysis co-exist in cells; the presence of IF1 in the complex seems to regulate ATP levels[17–19]. Previous studies showed that ATP synthase monomers have a molecular weight of ~600 kDa while dimers run at ~1150 kDa[20]. Using blue native gels and antibodies against subunit c-ring ATP5O, ATP8, or ATPβ to visualize the holoenzyme, we found that IF1 was detected in the native ATP synthase complex under baseline conditions in WT but not KO cells (Fig. 1C and Supplementary Fig. 1E). We then isolated mitochondria and performed in-gel activity assays to define whether the absence of IF1 modulated ATP hydrolysis. We found significantly increased levels of ATP hydrolytic activity in the KO cells compared to the WT counterparts (Fig. 1D). Together, these data are consistent with a fraction of IF1 being bound to complex V in WT cells grown under standard cell culture conditions to inhibit its reverse rotation.

Different mechanisms operate to maintain ΔΨm beyond the ETC, including glycolytic ATP hydrolysis by reverse rotation of the F1 subunit, the exchange of ATP⁴⁻/ADP²⁻ by the adenine nucleotide translocator, the exchange of H⁺ for ions (Na⁺, K⁺ or Ca²⁺) by mitochondrial ion exchangers and uncoupling of oxidative phosphorylation (OXPHOS) by specific mitochondrial proteins[21]. To define the extent to which ATP hydrolysis contributed to the higher ΔΨm of IF1-KO cells, we cultured WT and KO counterparts in galactose medium to limit the pool of glycolytic ATP. We found that the ΔΨm decreased in both cell types in galactose relative to glucose medium, but the effects were significantly more pronounced in the KO cells (Fig. 1E). Thus, the increased resting ΔΨm in the KO cells relies on a significant contribution of hydrolysis of glycolytic ATP. The effects of galactose in the ΔΨm of WT cells were surprising, but seem in line with a more significant role of IF1 in regulating ATP synthesis under normal physiological conditions than originally anticipated[19].

### Chronic loss of IF1 leads to a robust transcriptional response

Next, we used RNA-seq to understand the broader cellular adaptations resulting from a higher resting ΔΨm. We found over 6000 differentially expressed genes (DEGs) between the KO and WT cells, of which 3884 genes were upregulated (red) and 2669 were downregulated (blue) (Fig. 2A, Supplementary Data 1). This large transcriptional response in the absence of exogenous stress was unexpected but suggests that the higher resting ΔΨm might provide a signal that cells not only sense but can respond to. Genes associated with glycolysis were upregulated in the IF1-KO cells (Supplementary Data 1), consistent with unrestrained hydrolysis of glycolytic ATP contributing to the maintenance of a higher resting ΔΨm. The expression of genes involved in transcription, signaling, cell cycle, DNA repair, and lipid metabolism was also increased (Supplementary Data 1). Interestingly, 8.7% of DEGs in the entire data set coded for mitochondrial proteins, which were generally downregulated (Supplementary Data 1). Gene Ontology analysis identified several mitochondrial processes involved in OXPHOS, ETC, and the proton motive force as enriched by the downregulated DEGs (Fig. 2B), including 54% of the nuclear subunits of the ETC and 46% of genes involved with the mitoribosome (Fig. 2C). Inhibition of ETC complexes was reported to compensate for oligomycin toxicity as a rheostat mechanism[22]; notably, oligomycin also leads to increased ΔΨm when ATP synthase is in forward operation.

We then stably re-introduced IF1 in the KO cells through a lentiviral vector. Levels of the ectopically-expressed protein were about sixfold those of the endogenous as per the WT cells (Supplementary Fig. 2A). Herein, we call these cells OE for overexpressors. RNA-seq

analysis identified over 9000 DEGs between the KO and OE (Supplementary Data 1). Of these, 4185 were DEGs between KO vs WT and broadly enriched for mitochondria (Fig. 2D). The expression of 2365

out of the 4185 common genes was fully reversed by the reintroduction of IF1 (Fig. 2E), which we interpret to reflect those responsive specifically to changes in ATP hydrolytic activity by the

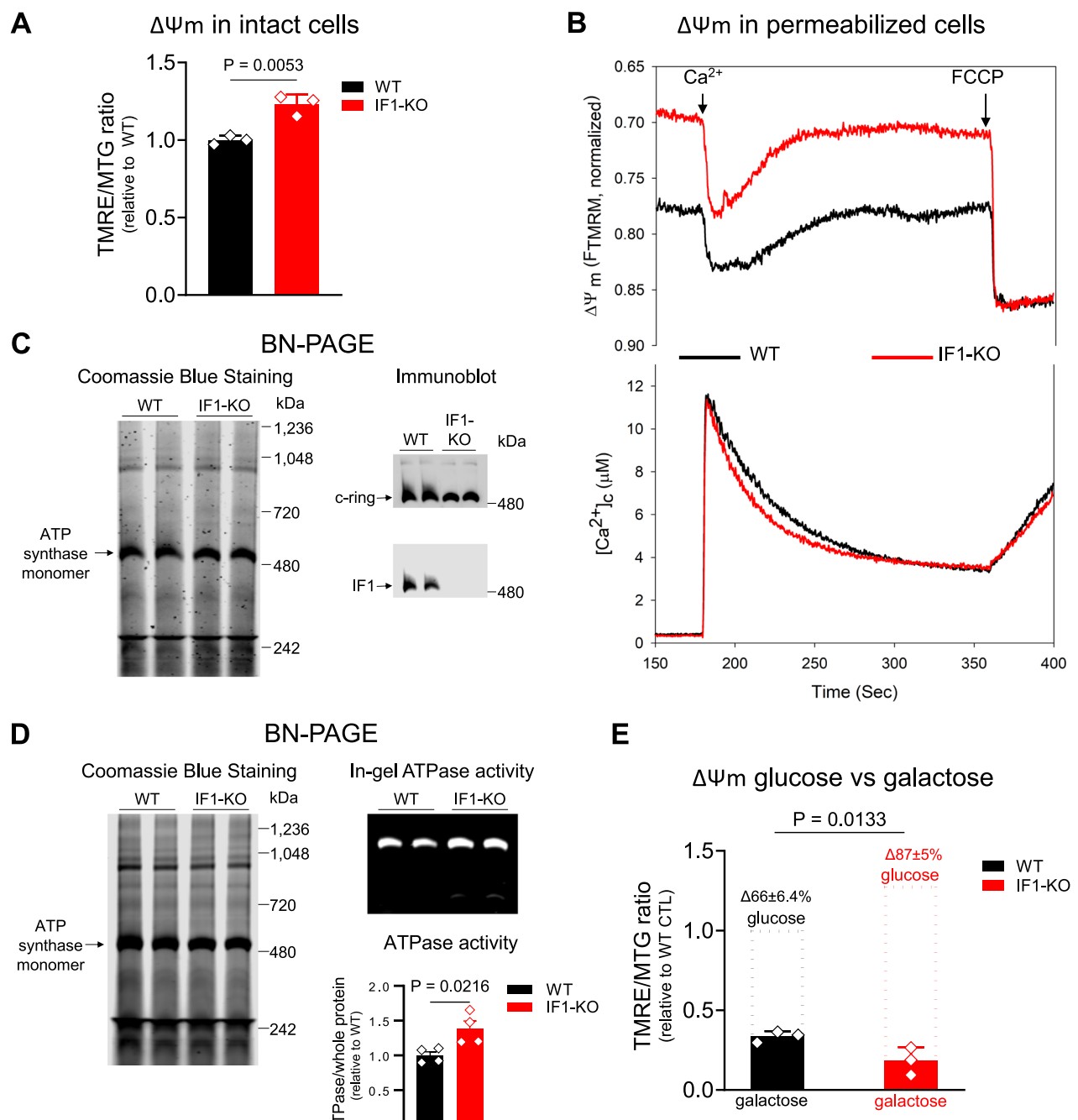

**Fig. 1 | Ablation of IF1 increases resting ΔΨm via ATP synthase hydrolytic activity. A** ΔΨm was measured in intact cells using TMRE; data were normalized to mitochondrial content using the ΔΨm-insensitive dye MitoTracker® Green (MTG). On y axis, TMRE/MTG ratio. Data are presented as mean values ± SEM of n = 3 biological replicates. Statistical difference by two-sided unpaired Student's t test. **B** Representative trace of concomitant ΔΨm and Ca²⁺ clearance measurements in permeabilized cells with succinate; n = 4 independent biological replicates. The black line represents WT cells, and the red line represents IF1-KO cells. FCCP was added at the end of the experiment to fully depolarize the ΔΨm and release the intramitochondrial pool of Ca²⁺. **C** Representative immunoblots of blue native (BN)-PAGE. Left panel, digitonin-permeabilized WT (lanes 1, 2) and IF1-KO (lanes 3, 4) mitochondria stained with Coomassie Blue. The predicted size of ATP synthase is pointed out by the arrow. Right upper panel, immunoblots of WT and IF1-KO

mitochondria blotted for ATP synthase $F_o$ rotor (c-ring). Right lower panel, WT and IF1-KO mitochondria blotted for IF1. Results were reproduced independently three times and with additional antibodies probing other ATP synthase subunits (Supplementary Fig. 1). **D** Coomassie-stained BN-PAGE showing ATP synthase. Samples were evaluated for ATP hydrolase activity using in-gel activity assays (upper right gel). Quantification of the hydrolase activity is shown in the bar graphs. Data are presented as mean values ± SEM of n = 4 biological replicates. Statistical difference by two-sided unpaired Student's t test. **E** Bar graph of ΔΨm as measured in **A** in cells grown in glucose (dotted lines) or 5 mM galactose (filled bars). Y axis depicts the TMRE/MTG ratio. Numbers beside each bar represent the % difference (Δ) of the ΔΨm when each cell was grown in high glucose vs. in galactose. Data are presented as mean values ± SEM of n = 3 biological replicates per cell type/condition. Statistical difference by two-sided unpaired Student's t test.

presence/absence of IF1. Notably, the 1293 genes that were downregulated in the KO and upregulated in the OE enriched for mitochondrial processes, while the 1072 upregulated genes in the KO that were reciprocally downregulated in the rescue enriched for glucose, phospholipid transport and metabolism (Fig. 2F–H). The modulation of phospholipid genes was interesting as a recent study linked

phosphatidylethanolamine (PE), a main phospholipid of mitochondrial membranes, with the regulation of proton flux by the uncoupling protein 1 (UCP1) in brown adipose tissue[23]. Uncoupling through proton dissipation is a known mechanism to regulate the $\Delta\Psi m$[24]. Comparison of WT and OE cells revealed the differential regulation of ~3600 genes, out of which 2339 were common to the OE vs the KO and likely reflect

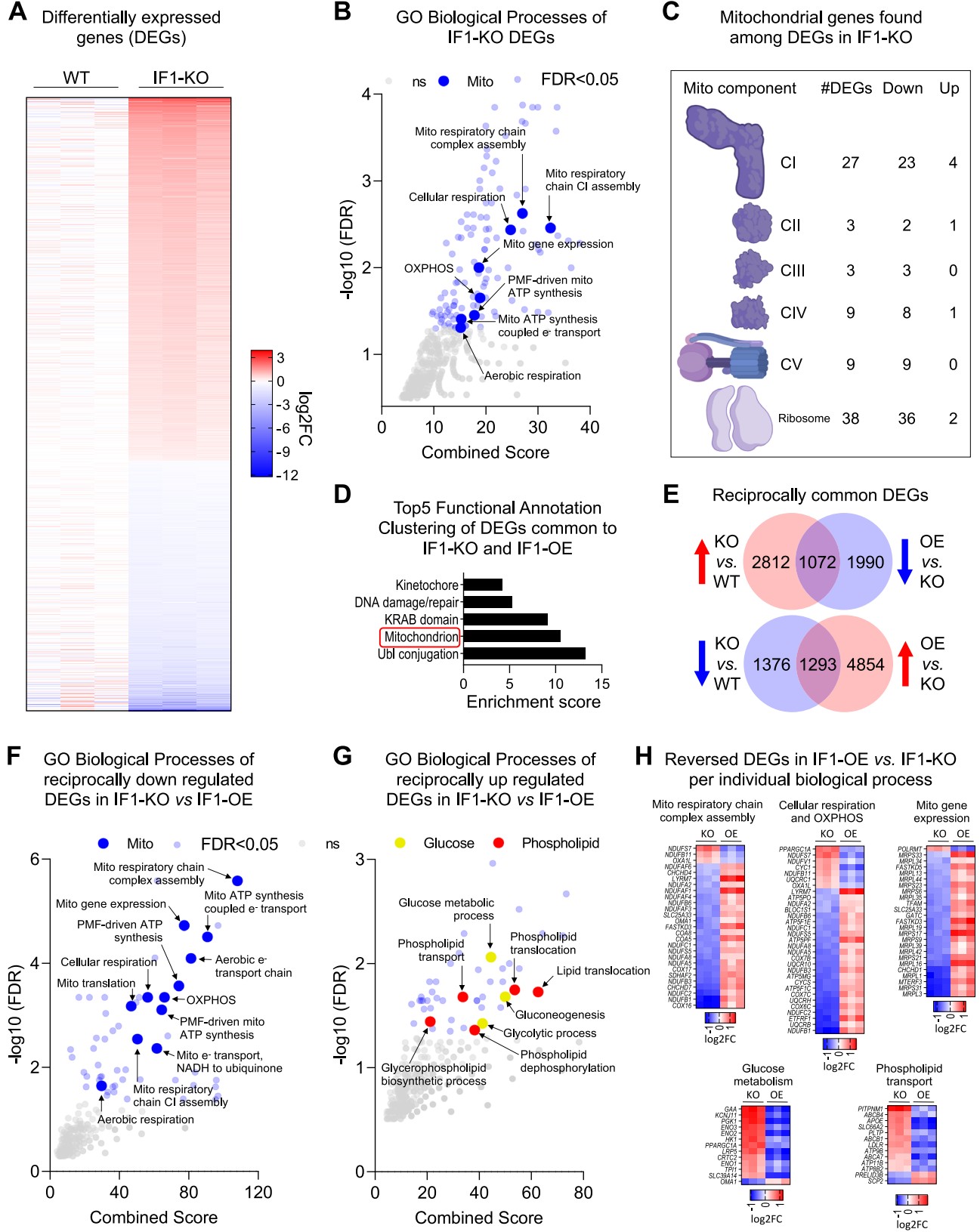

**A** Differentially expressed genes (DEGs)

**B** GO Biological Processes of IF1-KO DEGs

**C** Mitochondrial genes found among DEGs in IF1-KO

| Mito component | #DEGs | Down | Up |
|---|---|---|---|
| CI | 27 | 23 | 4 |
| CII | 3 | 2 | 1 |
| CIII | 3 | 3 | 0 |
| CIV | 9 | 8 | 1 |
| CV | 9 | 9 | 0 |
| Ribosome | 38 | 36 | 2 |

**D** Top5 Functional Annotation Clustering of DEGs common to IF1-KO and IF1-OE

**E** Reciprocally common DEGs

**F** GO Biological Processes of reciprocally down regulated DEGs in IF1-KO vs IF1-OE

**G** GO Biological Processes of reciprocally up regulated DEGs in IF1-KO vs IF1-OE

**H** Reversed DEGs in IF1-OE vs. IF1-KO per individual biological process

**Fig. 2 | IF1-KO cells with increased ΔΨm engage in a transcriptional feedback loop repressing a mitochondrial gene expression program. A** Heatmap of differentially expressed genes (DEGs) in IF1-KO vs. WT (False Discovery Rate, FDR < 0.05). The color scheme is presented as a log2 fold change (log2FC). Shades of red represent upregulated genes and shades of blue represent downregulated genes. **B** Gene Ontology (GO) analysis of DEGs in IF1-KO vs. WT cells. −Log10 FDR values on the y axis, and combined score on the x axis. In bigger blue circles, significantly enriched pathways (FDR < 0.05) related to mitochondria. In smaller translucid blue and gray circles, other significantly enriched pathways (FDR < 0.05) and non-significant pathways (FDR > 0.05), respectively. **C** Structural depiction of individual complexes of the electron transport chain and mitochondrial ribosomes. The number of DEGs associated with these complexes and their transcriptional directionality are also depicted. Created in BioRender. Mori, M. (2025) https://BioRender.com/wfvugau. **D** Functional Annotation Clustering of common DEGs reciprocally inverse changed between IF1-KO vs. WT and IF1-OE vs. IF1-KO. Keywords and terms found in each cluster are shown on the y axis, and the Enrichment score is on the x axis. **E** Venn diagrams show the intersection between reciprocally regulated DEGs in IF1-KO vs. WT and IF1-OE vs IF1-KO. Upper circles: upregulated DEGs based on IF1-KO vs. WT comparison, lower circles represent the down-regulated counterparts. **F, G** GO analysis of reciprocally regulated DEGs in IF1-KO cells, as described in (**B**). **H** Heatmap of representative pathways reversed in IF1-OE vs. IF1-KO as identified by GO analysis. Color scheme as described in (**A**). KO is IF1-KO vs. WT, and OE is IF1-OE vs. IF1-KO.

the cellular effects of IF1 overexpression on ATP synthesis[25]. Consistent with this interpretation, mitochondrial ATP synthesis coupled to electron transport was the only mitochondria-associated pathway commonly enriched between WT and KO cells when IF1 was overexpressed (Supplementary Data 1). The significant modulation of mitochondrial genes in the genotypes analyzed prompted us to examine mitochondrial function using the Seahorse Flux Analyzer. We followed oxygen consumption in glucose- or galactose-supplemented media and found that, despite minor changes observed in glucose conditions, no significant mitochondrial impairment was observed in the KO cells (Supplementary Fig. 2B, C). Consistent with IF1 overexpression affecting ATP synthesis, a modest but significant decrease in basal respiration and ATP production was observed in the OE relative to the WT cells under glucose conditions (Supplementary Fig. 2C). Transmission electron microscopy failed to identify gross changes in mitochondrial ultrastructure among the three genotypes (Supplementary Fig. 2D, E). Despite the hyperpolarized state, KO cells were as proficient in undergoing mitophagy as the WT or OE counterparts (Supplementary Fig. 2F). Based on these collective findings, we interpret the downregulation of genes involved in mitochondrial processes, including those of the ETC, as adaptations to maintain the ΔΨm at an 'optimal' state that supports mitochondrial and cellular function in the KO cells. In agreement, we found that mitochondria could further hyperpolarize when KO or WT cells were put under phosphate-depleted conditions (Supplementary Fig. 2G), recapitulating effects recently reported for yeast and other cells[15]. Thus, at baseline, the ΔΨm of IF1-KO cells is not maintained at its maximal polarized state.

## Chronic rise in ΔΨm leads to nuclear DNA hypermethylation regulating a fraction of gene expression

It is increasingly evident that mitochondria communicate with the epigenome[26], including modulating DNA methylation to influence transcription[27,28]. To define whether the broad transcriptional responses found in the KO cells were associated with changes in DNA methylation, we profiled the methylation status of the nuclear genome using the Illumina 850 K array. Remarkably, we found that the nuclear DNA of the KO cells was significantly hypermethylated relative to the WT counterparts in promoters, gene bodies, 5'- and 3'-UTRs, as well as intergenic regions (Supplementary Fig. 3A, B). We then focused on promoter methylation to establish a relationship with transcription. In a simplistic model, promoter hypermethylation is associated with gene repression while promoter hypomethylation leads to gene upregulation. We identified promoters based on genomic coordinates of the transcription start site (TSS) ±1500 bases (Supplementary Fig. 3A) and found a total of 8774 unique gene promoters differentially methylated in the KO cells (Supplementary Data 2). Overlaying these promoter coordinates with those of the DEGs revealed that 2531 were differentially methylated and expressed genes (DMEGs, Supplementary Data 2), out of which 1875 were hypermethylated and 656 were hypomethylated. From these, 672 genes had the promoter hypermethylated and were repressed (Fig. 3A, cluster 2, Supplementary

Fig. 3C) while 395 genes had the promoter hypomethylated and were upregulated (Fig. 3A, cluster 3; Supplementary Fig. 3C). Genes from cluster 2 enriched for mitochondrial processes (Fig. 3B, Supplementary Data 2) whereas those from cluster 3 were involved in phospholipid transport (Fig. 3C, Supplementary Data 2). It is worth noting that another 22 DEGs associated with mitochondrial or phospholipid metabolism were differentially methylated, but outside of the TSS (Supplementary Data 2). Pathways enriched by DMEGs from clusters 1–4 can be found in Supplementary Data 2. Collectively, these data suggest that the differential nuclear DNA methylation contributed to the transcriptional response of a fraction of the genes responding to the increased resting ΔΨm.

## Reversal of hyperpolarization reinstates the epigenetic and transcription landscapes to WT levels

If the higher ΔΨm drives nuclear DNA hypermethylation to influence transcription, then decreasing the ΔΨm of IF1-KO cells should reverse the epigenetic and transcriptional changes. We pharmacologically decreased the ΔΨm of IF1-KO cells by treatment with the mild protonophore dinitrophenol (DNP, Supplementary Fig. 4A). We also genetically decreased the ΔΨm of IF1-KO cells by re-introducing IF1 or by ectopically expressing UCP4 (*SLC25A27*, Supplementary Fig. 4B, C), which, by decreasing ΔΨm to different extents (Fig. 3D, E) provided independent means to chronically alter the ΔΨm. UCP4 had been previously shown to mildly decrease the ΔΨm of cultured cells by uncoupling[29]. We then profiled nuclear DNA methylation using the Illumina array; we focused on the genetic models to avoid potential confounding effects and toxicity associated with long-term exposure to DNP. Strikingly, KO cells with a lower ΔΨm either through IF1 reintroduction or UCP4 ectopic expression had genome-wide TSS methylation levels that were close to those observed in the WT counterparts (Fig. 3F, Supplementary Fig. 4D). The reversal in DNA methylation was observed at specific loci, including those involved in mitochondrial and phospholipid genes (Fig. 3G, Supplementary Fig. 4E, Supplementary Data 3). The ectopic expression of UCP4 also normalized or reversed the expression of specific mitochondrial, glucose metabolism, and phospholipid transport genes (Fig. 3H), essentially mirroring the partial effects on the transcriptome observed upon the re-introduction of IF1 (Fig. 2H). Curiously, DML-containing nuclear-encoded mitochondrial genes are enriched for regulation of the ΔΨm in the KO cells, irrespective of the means utilized to decrease ΔΨm (Fig. 3I). Collectively, these data strongly support the hypothesis that increased ΔΨm is the upstream signal that regulates the epigenetic landscape and the transcriptional output of a subset of genes differentially expressed.

## ΔΨm-regulated DNA methylation is independent of metabolites or redox changes

Previous work connected dysfunctional mitochondria to nuclear DNA methylation, which was primarily driven by modulation of TCA cycle-associated metabolites[26]. Because no mitochondrial dysfunction was observed in the KO cells, we questioned whether the levels or activity

of the enzymes involved in nuclear genome methylation were altered, resulting in the epigenetic changes observed. DNA methylation is maintained by opposing reactions that deposit or remove methyl groups from cytosines. DNA methyltransferases (DNMTs) use S-adenosylmethionine (SAM) to methylate the DNA, while the removal of that methyl group is initiated through its oxidation by the Ten Eleven Translocation (TETs) enzymes, followed by downstream thymine

DNA glycosylase (TDG)-initiated base excision repair[30,31]. TETs are α-ketoglutarate (α-KG)-, oxygen- and iron-dependent dioxygenases[32] that use ascorbate as a co-factor; they are inhibited by succinate or the structure-related metabolites fumarate and 2-hydroxyglutarate[33,34]. TETs are also inhibited by oxidative stress[35]. As no significant changes at the transcription (Supplementary Data 1) or protein levels of DNMT, TETs, or TDG (Supplementary Fig. 5A) that could explain the DNA

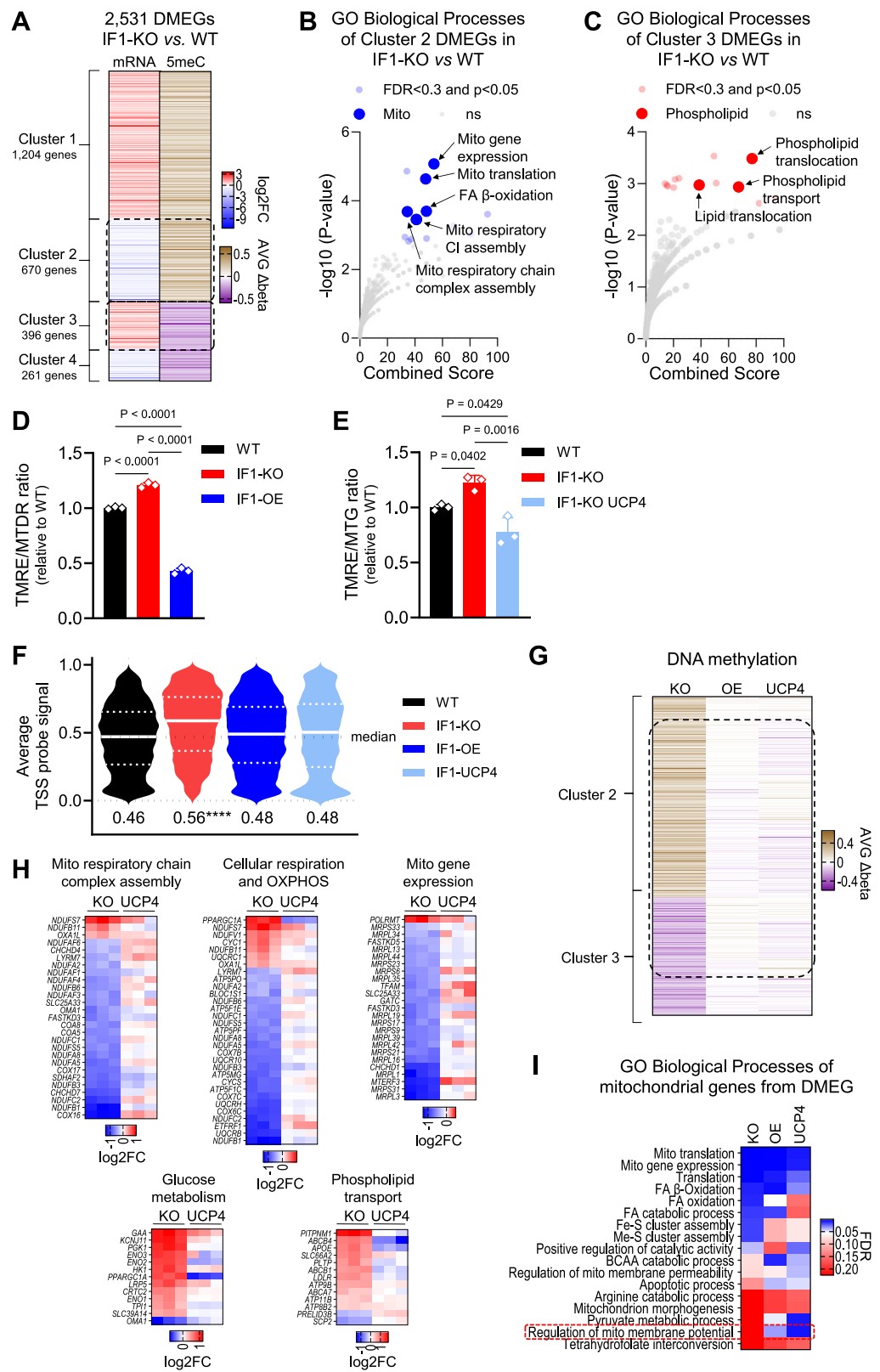

**Fig. 3 | Promoters of mitochondrial and phospholipid genes are hypermethylated in IF1-KO cells. A** Heatmap of differentially methylated and expressed genes (DMEGs); left track data from RNA-seq and right track data from DNA methylation (5meC). Data are presented as a log2 fold change (log2FC) for gene expression, the average change in DNA methylation is shown as average delta beta (AVGΔbeta). In red: upregulated genes, and downregulated in blue; hypermethylated gene promoters are shown in brown and hypomethylated in purple. DMEGs were grouped in 4 clusters: 1) hypermethylated and upregulated, 2) hypermethylated and downregulated, 3) hypomethylated and upregulated, and 4) hypomethylated and downregulated. The dashed lines highlight DMEGs from clusters 2 and 3. **B, C** GO analysis of DMEGs of clusters 2 and 3, respectively. −Log10 FDR values on the y axis and combined score on the *x* axis. **D, E** Bar graph of ΔΨm as measured by TMRE normalized to MTG. *Y* axis depicts the TMRE/MTG ratio. Data are presented as mean values ± SEM of *n* = 3 biological replicates. Statistical difference by two-sided one-way ANOVA with Tukey's post-test. **F** Violin plots showing average probe signal in the transcriptional start site (TSS) across the genome in the four genotypes; the *y* axis depicts average probe values; white lines indicate the median and dotted lines the quartiles. Numerical values below the violin plot are the mean DNA methylation level of TSSs for each genotype (*n* = 4). Significance based on on two-sided one-way ANOVA with Dunnett's correction ****$p$ = <0.0001. **G** Heatmap of locus-specific DNA methylation from clusters 2 and 3 genes (from **A**); each line represents the same locus. The color scheme is presented as AVGΔbeta. Hypermethylated gene promoters in brown, hypomethylated in purple. Left track (KO) IF1-KO vs. WT, middle track (OE) IF1-OE vs. IF1-KO, and right track (UCP4) IF1-KO UCP4 vs. IF1-KO (*n* = 4). **H** Heatmap of representative pathways as in Fig. 2H (*n* = 3). **I** Heatmap of biological processes of mitochondrial genes from DMEGs of each paired comparison. The color scheme is presented as FDR. Comparisons are: left track (KO) IF1-KO vs. WT, middle track (OE) IF1-OE vs. IF1-KO, and right track (UCP4) IF1-KO UCP4 vs. IF1-KO.

hypermethylation phenotype were observed, we next focused on metabolites.

Steady-state metabolomics did not reveal changes in the abundance of metabolites that might influence DNMT or TET activity, including SAM, 5′-deoxy-5′-methylthioadenosine, α-KG, or its competitive inhibitors succinate and fumarate (Fig. 4A), suggesting metabolic adaptation to a higher resting ΔΨm. While SAM is primarily generated in the cytosol, from where it is available for the many reactions that utilize it, including in the nucleus[36], a smaller pool of SAM is present in the mitochondria[37]. The mitochondrial solute carrier 25A26 (SLC2A26) imports cytosolic-synthesized SAM in exchange for matrix SAH; decreases in mitochondrial SAM were shown to impair OXPHOS assembly and induce complex I instability[38]. Conceptually, a decrease in the mitochondrial pool could increase cytosolic levels in the KO cells, differently impacting the epigenome without altering the total amount of SAM. To gain insights into this possibility, we first mined our RNA-seq data as SLCs are known to be regulated primarily transcriptionally, although their activity can be modulated based on substrate availability or on membrane fluidity[39]. We found that while ~100 genes coding for SLCs were differentially transcribed between the WT and the KO cells, *SLC25A26* was not among them (Supplementary Data 1). Because SAM was not limiting in the cytosol (as per metabolomics), and there was no detectable mitochondrial dysfunction (Supplementary Fig. 2B, C) in the KO cells, it seems unlikely that differences in the compartmentalized pool of SAM caused the epigenetic effects observed.

We then considered the possibility that redox changes drove the nuclear DNA hypermethylation. TETs are redox sensitive, and a rise in ΔΨm is expected to increase ROS production, which could presumably inhibit these enzymes, increasing DNA methylation. Consistent with changes in the cellular redox state, joint-pathway analysis using both the metabolomics and RNA-seq data indicated that glutathione metabolism was enriched in IF1-KO cells compared to WT (Fig. 4B). However, contrary to the expectations that cells would be under oxidative stress, several lines of evidence indicated that IF1-KO cells adapted to the chronic higher resting ΔΨm by reducing the cellular redox environment. For example, levels of glutathione, NADH, NAD+, and NADP were increased in the KO cells (Supplementary Fig. 5B), whereas the ratio of oxidized (GSSG) to reduced (GSH) glutathione decreased (Fig. 4C). Targeting the redox probe GRX1-roGFP2 to the cytosol or mitochondria of WT and KO cells revealed that the mitochondrial matrix, but not the cytosol, was oxidized (Fig. 4D), inconsistent with the cellular compartments directly continuous with the nucleus being under a state of oxidative stress. Levels of the redox couple NADP+/NADPH were increased (Fig. 4E), and the amount of hydrogen peroxide (H₂O₂) released in the medium, as gauged by Amplex® Red, was decreased in the KO cells (Fig. 4F)−collectively indicating increased antioxidant capacity.

## Changes in phospholipids link the effects of nuclear DNA methylation to the ΔΨm

Further inspection of the metabolomics data pointed to significant changes in phospholipids between WT and KO cells (Supplementary Fig. 5C). Specifically, levels of PE and phosphatidylcholine (PC), the two major phospholipid components of membranes, including that of mitochondria, were reversibly altered between the KO and OE cells (Supplementary Fig. 5D). These changes affected the global cellular PC/PE ratio (Supplementary Fig. 5E). Decreases in PC were quantitatively confirmed by high performance thin layer chromatography (HPTLC) both at the whole cell as well as isolated mitochondria levels (Fig. 5A). This approach showed no significant changes in the levels of PE (Fig. 5B), but the PC/PE ratio was nevertheless decreased (Fig. 5C). HPTLC also revealed increased levels of phosphatidyl serine (PS), a precursor of PE in mitochondria, in both whole cell and isolated mitochondria (Supplementary Fig. 5F, G). Parallel changes in phospholipid content in whole cells and the mitochondrial membrane were recently reported by others[40].

From an epigenetic perspective, a decrease in the cellular PC/PE ratio in yeast was previously shown to result in histone hypermethylation[41], which prompted us to evaluate histone methylation in our cells. No changes in levels of histone 3 (H3) methylation were observed in the KO, OE, or UCP4-overexpressor relative to the isogenic WT counterparts (Supplementary Fig. 5H). As yeast does not seem to have enzymatically driven DNA methylation[42], these data suggest that the methylation of the DNA, but not of histones, is primarily affected by a decrease in the cellular PC/PE ratio in mammalian cells. The previous report[41] relied on the deletion of phosphatidylethanolamine methyltransferase (PEMT), the enzyme that methylates PE into PC in one of the pathways of PC production. Given HEK293 cells express PEMT at relatively high levels compared to other cell types (Supplementary Fig. 5I), these results raised the intriguing possibility that methyl groups might be diverted from this reaction into nuclear DNA under our experimental conditions.

A premise of this hypothesis is that PE methylation should be decreased while cytosine methylation should be increased in the KO cells relative to the WT controls. We tested these by following the fate of methyl groups using L-methionine-(*methyl*-d3), a methionine isotopologue deuterated at C5 from which carbons will eventually be incorporated into PC and the DNA (Fig. 5D). If methylation of PE to generate PC was diverted to the DNA in the KO cells, we expected to identify decreased levels of +3- or +6-labeled PC along with increased +3 labeling of cytosine. Remarkably, concomitant to the decreased labeling of PC (Fig. 5E) were increases in labeling of 5meC (Fig. 5F), suggesting that a decrease in the PC/PE ratio, likely driven by modulation of PEMT activity, might underlie hypermethylation of the nuclear epigenome. We were able to detect +3 and +6 isotopologues of PC(16:0/16:0), but not +9. This is consistent with methionine being

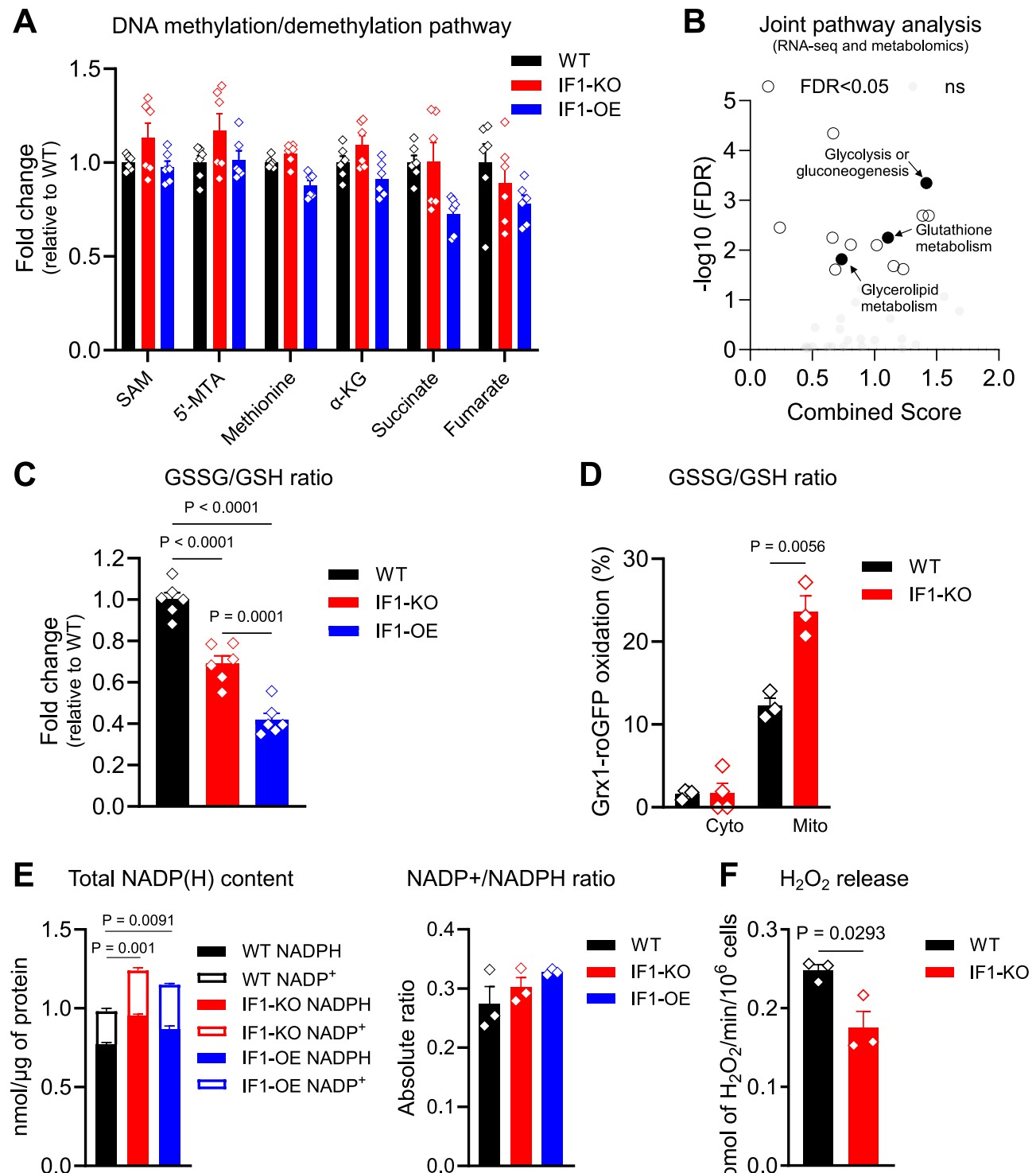

**Fig. 4 | Neither metabolites or redox changes seem associated with epigenetic changes upon mitochondria hyperpolarization. A** Measurement of metabolites involved in DNA (de)methylation reactions per whole cell steady-state metabolomics. Mean WT was set as 1 ($n = 6$ per genotype; error bars represent ±SEM). On the $y$ axis, fold changes (FC) of metabolites relative to WT (black), red represents the IF1-KO and blue the IF1-OE counterparts. **B** Joint-pathway analysis using genes and metabolites differently enriched as per RNA-seq and metabolomics analyses, respectively. −Log10 FDR values on the y axis, and combined score on the $x$ axis. Highlighted in black circles are the pathways of interest. **C** Oxidized/reduced glutathione (GSSG/GSH) ratio based on metabolomics data. On the y axis, GSSG/GSH ratio relative to WT. Data are presented as mean values ± SEM of $n = 6$ biological replicates. Statistical difference by two-sided one-way ANOVA with Tukey's post-test. **D** GSSG/GSH as per degree of Grx1-roGFP probe oxidation in the cytosol (cyto, construct is targeted to the cytosol) ($n = 4$) and mitochondria (mito, probe is targeted to the mitochondria) ($n = 3$). Data are presented as mean values ± SEM. Statistical difference by two-sided unpaired Student's t test. **E** Total NADP(H) content was estimated using a commercially available kit. Left graph, on the y axis, nmol NADPH or NADP$^+$ per µg protein. WT black fill and outline, IF1-KO red fill and outline, IF1-OE blue fill and outline. Filled and outlined bars represent NAPDH and NADP$^+$, respectively. Right graph, NADP$^+$/NADPH ratio. Data are presented as mean values ± SEM of $n = 3$ biological replicates with two technical replicates. Statistical difference by two-sided one-way ANOVA with Tukey's post-test. **F** H$_2$O$_2$ release in the medium as accessed by Amplex® Red. On the y axis, pmol of H$_2$O$_2$/min/10$^6$ cells. Data are presented as mean values ± SEM of $n = 3$ biological replicates. Statistical difference by two-sided unpaired Student's t test.

recycled via homocysteine with methyl donor 5-methyltetrahydro folate through serine-glycine one-carbon metabolism[43]. Indeed, low levels of +0 methionine were detected in the WT and IF1-KO cells (Supplementary Fig. 5J), even though the only exogenous source of methionine was the deuterated (+3) form.

To define whether a decrease in PEMT activity was mechanistically linked to the epigenetic phenotype, we overexpressed PEMT in the KO cells. We flag-tagged PEMT to visualize the protein by Western blots

(Supplementary Fig. 5K) since commercially available antibodies were, at least in our hands, highly unspecific. Next, we performed whole-genome methylation analysis in the KO and PEMT-overexpressing isogenic counterparts using the Illumina array. We found that 320 promoters were differentially methylated by overexpression of PEMT in the KO background relative to the KO cells alone; 86 promoters were hypomethylated and 234 hypermethylated between the cells (Supplementary Data 3). Notably, 190 of these differentially methylated

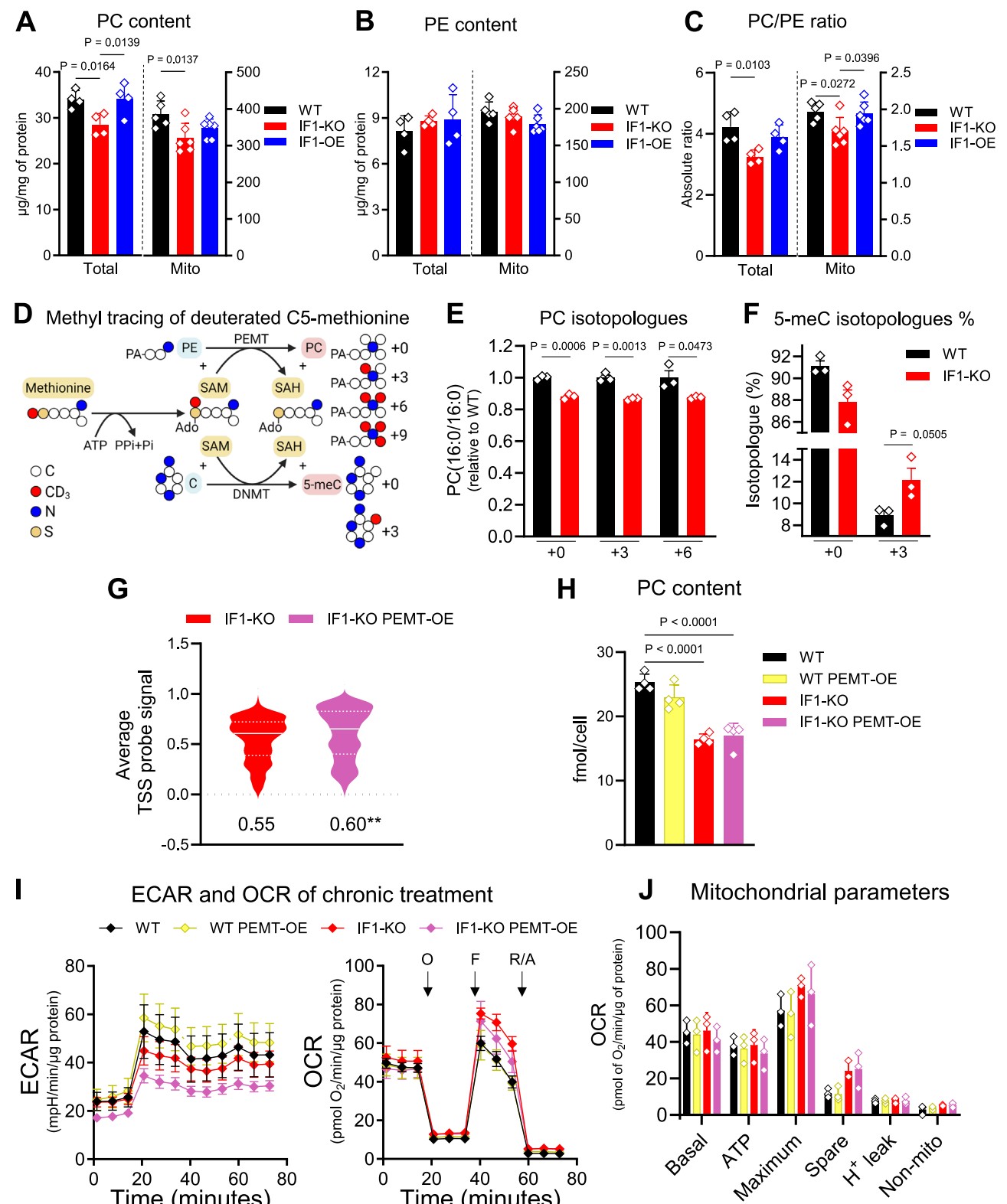

**Fig. 5 | Methyl groups are re-routed from phospholipid to DNA methylation when mitochondria are hyperpolarized. A**, **B** Total (whole cell) and mitochondrial (mito) PC and PE content were analyzed by HPTLC. On $y$ axis, PC and PE in µg/mg of protein. **C** PC/PE ratio from data in (**A**, **B**). On $y$ axis, absolute PC/PE ratio. Data are presented as mean values ± SEM of $n = 4$ biological replicates (whole cell), and $n = 6$ biological replicates (mito). Statistical difference by two-sided one-way ANOVA with Tukey post-test. **D** Schematic diagram for tracing methyl group using deuterated C5-methionine. Carbon (C) is depicted in white circles, nitrogen (N) in blue, sulfur (S) in yellow, and deuterated methyl group ($CD_3$) in red. Deuterated SAM ($CD_3$-SAM) is formed from deuterated C5-methionine. Methyl transfer from $CD_3$-SAM to: 1) PE to form PC yields +3, +6, or +9 isotopologues; and 2) C to form 5meC yields +3 isotopologue. Created in BioRender. Mori, M. (2025) https://BioRender.com/36yccol. **E** Measurement of PC(16:0/16:0) isotopologues; mean WT was set as 1. On the $y$ axis, fold changes (FC) of PC relative to WT. On the $x$ axis,

isotopologues +0, +3, and +6. Black bars, WT; red bar, IF1-KO. **F** Total 5meC was set as 100% for each sample, and isotopologues +0 and +3 were calculated as % from total. On the $y$ axis, relative abundance of each isotopologue. On the $x$ axis, isotopologues +0 and +3. Black bars, WT; red bar, IF1-KO. Data are presented as mean values ± SEM of $n = 3$ biological replicates. Statistical difference by two-sided unpaired Student's $t$ test. **G** Average probe signal at the TSS of genes in IF1-KO (red) relative to IF1-KO PEMT-OE (violet). Statistical difference by two-sided unpaired Student's $t$ test ($n = 4$) **$p = 0.0028$. **H** Total PC content in WT (black), WT PEMT-OE (yellow), IF1-KO (red), and IF1-KO PEMT-OE (violet). Data are presented as mean ± SEM of $n = 4$ biological replicates. Statistical difference by two-sided one-way ANOVA with Tukey's post-test. **I** Seahorse flux analyzer was used to estimate oxygen consumption in the cells used in (**H**). $n = 3$/genotype. O is oligomycin, F is FCCP, and R/A refer to rotenone/antimycin. **J** Graphical representation of oxygen consumption rates-derived parameters from I.

promoters were already changed by loss of IF1 alone (Supplementary Data 2). The average change in DNA methylation when overexpressing PEMT was ±10% (Supplementary Data 3), while these changes ranged ±80% by loss of IF1 alone (Supplementary Data 2). While a comparative increase in the levels of DNA methylation were observed (Fig. 5G), we conclude that constitutive overexpression of PEMT on top of IF1 loss did not result in a DNA methylation profile adjustment large enough, in magnitude or number of TSSs, to supersede the effect of IF1 deletion alone. Locus-specific changes over the 320 promoters are depicted in Supplementary Fig. 5L.

Feedback regulation between PEMT activity and the Kennedy pathway, which is the primary source of PC production in most cells, is a well-established phenomenon. Deletion of PEMT was shown to activate the Kennedy and vice versa, yielding no net changes in PC[44]. Lack of PC recovery could explain these data as it would, presumably, not impact the PC/PE ratio. Overexpression of PEMT indeed did not increase PC levels (Fig. 5H). Loss of PE in mitochondria has been shown to cause mitochondrial dysfunction[45]. Next, we estimated oxygen consumption using the Seahorse Flux analyzer as a proxy for potential changes in PE. We found no differences in mitochondrial function in any of the genotypes analyzed (Fig. 5I, J), suggesting that PE levels were not affected by overexpression of PEMT in ways that could reverse the PC/PE ratio. Collectively, these results suggest that the decreased PC/PE ratio is critical in influencing the methylation status of the nuclear epigenome under our experimental conditions. It is worth noting that genes associated with PC metabolism through the Kennedy pathway were modulated in the KO cells relative to the WT controls, and were reversed in the OE (Supplementary Fig. 5M). Whether this was an adaption to decreased PEMT activity or vice-versa is unclear.

## Phospholipid remodeling and nuclear DNA hypermethylation are also observed upon exposure to environmental agents that chronically hyperpolarize the mitochondria

If changes in the PC/PE ratio due to a higher resting ΔΨm mediate nuclear DNA hypermethylation in the KO cells, then chronically increasing the ΔΨm of WT cells should similarly affect the PC/PE ratio and hypermethylate the nuclear DNA. To test this, we first identified chemicals that can increase the ΔΨm without changes in mitochondrial volume or toxicity. To this end, we took advantage of data provided by Tox21, a consortium of different US Federal Agencies that aims at developing better toxicity assessment methods to evaluate the safety of chemicals, pesticides, food additives, contaminants, and medical products. In one phase, Tox21 leveraged high-throughput screenings, using various cell types and assays, to evaluate the activities of 10,000 chemicals. ΔΨm was among the parameters evaluated using the ratiometric fluorescent dye JC-10. It was found that about 30% of the 10,000 chemicals screened changed JC-10 fluorescence ratio, out of which ~5% increased the ΔΨm[4,56]. We selected a few of the chemicals (Supplementary Fig. 6A), including pharmaceutical drugs,

to confirm their ability to acutely increase the ΔΨm using TMRE, normalizing data to mitochondrial content (Fig. 6A).

Next, we performed time-course experiments to establish doses to which WT cells could be chronically exposed without toxicity, thus allowing them to adapt to the rise in resting ΔΨm. The agents ultimately selected for further analyses were telmisartan, a medication to control blood pressure, and annatto—a food additive used in South American cuisine. These test articles provided the means to model a physiological setting in which the resting ΔΨm may be chronically increased through voluntary environmental exposure. WT cells were exposed for 10 days to either of these chemicals, which proved to be well-tolerated (i.e., no loss of cell viability; Supplementary Fig. 6B), at which point we analyzed the ΔΨm, PC, and PE content, as well as nuclear DNA methylation. Exposure to telmisartan or annatto for 10 consecutive days led to significant increases in the resting ΔΨm of WT cells when compared to vehicle-treated controls (Fig. 6B). Like in the KO cells, this chronic rise in resting ΔΨm did not lead to mitochondrial dysfunction as judged by lack of defects on oxygen consumption or increased extracellular acidification (Fig. 6C, D). Analysis of phospholipids showed that the levels of PC were decreased in the treated cells while PE levels did not change, decreasing the cellular PC/PE ratio (Fig. 6G). More importantly, the nuclear DNA of treated cells was hypermethylated (Fig. 6H), collectively recapitulating the main phenotypes reported for the chronically hyperpolarized IF1-KO cells. These data lend further support to the notion that a decrease in PC methylation with effects on the PC/PE ratio links the epigenetic effects to the chronically increased ΔΨm.

## Phospholipid content alteration seems to be a hallmark adaptation to increase resting ΔΨm, including in disease

Why levels of phospholipids change when mitochondria are chronically hyperpolarized is unclear. PE and PC are the main phospholipids on mitochondrial membranes that can alter electron transfer and proton pumping efficiency[46], thus potentially regulating ΔΨm. Therefore, it is conceivable that phospholipid remodeling is a required adaptation that responds to and sustains a chronic state of hyperpolarization to support proper mitochondrial function. If this is true, then any cell that has the mitochondria hyperpolarized should show concomitant decreases in PC that affect the relative PC/PE ratio whilst maintaining mitochondrial respiration. To test this hypothesis, we took two approaches. First, we leveraged the known heterogeneity of mitochondria within cell populations to sort WT cells based on their ΔΨm; we then established phospholipid content and oxygen consumption rates (OCRs) in the sorted cells. Secondly, we found an ovarian cell line naturally depleted of IF1; we determined resting ΔΨm and the PC/PE ratio relative to a non-isogenic IF1-positive counterpart derived from a donor of the same race and tumor grade. These not only eliminated the possibility that phospholipid changes were an off-target effect of the chemical- and/or genetic manipulations, but also

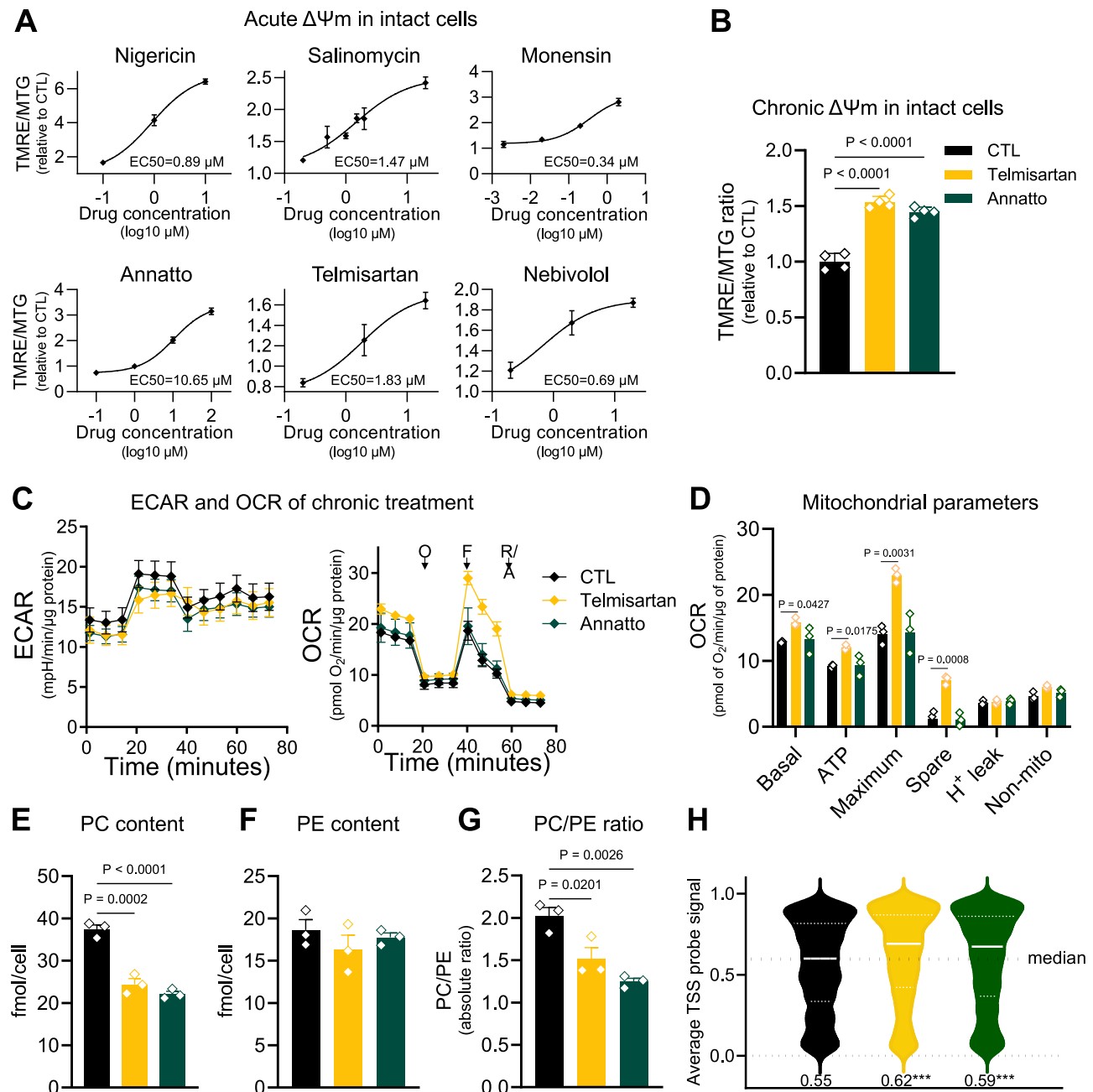

**Fig. 6 | Chronic exposure of WT cells to chemicals that hyperpolarize mitochondria leads to loss in PC, decreased PC/PE ratio, and nuclear DNA hypermethylation. A** Hill slope was used to determine EC50 of chemicals tested. On the y axis, the TMRE/MTG ratio, and on the x axis chemical concentration in log10 μM. Error bars represent ±SD. **B** Chemical-induced ΔΨm hyperpolarization in cells treated for 10 days. Statistical difference by two-sided one-way ANOVA with Dunnett's post-test compared to CTL; error bars represent ±SEM of n = 4 biological replicates. **C** Oxygen consumption rate (OCR) as per the Seahorse Flux analyzer; on the y axis in pmol O₂/min/μg protein. Right graph shows extracellular acidification rate (ECAR) on the y axis in mpH/min/μg protein. O is oligomycin, F is FCCP, and R/A refer to rotenone/antimycin (n = 3); error bars represent ±SEM. **D** Graphical representation of oxygen consumption rates-derived parameters. Statistical difference by two-sided one-way ANOVA with Dunnett's post-test compared to CTL; error bars represent ±SEM of n = 3 biological replicates with 6 or 8 experimental

replicates based on (**C**). **E**, **F** PC and PE content, respectively, in cells chronically treated with ΔΨm hyperpolarizing chemicals. On the y axis, PC or PE content in nmol/10⁶ cells. **G** Absolute PC/PE ratio from (**E**, **F**) in cells chronically treated with ΔΨm hyperpolarizing chemicals. CTL (black), telmisartan 2 μM (yellow) and annatto 10 μM (green) for (**E**–**G**). Statistical difference by two-sided One-way ANOVA with Dunnett's post-test correction; error bars represent ±SEM of n = 3 biological replicates with 2 experimental replicates for (**E**–**G**). **H** Violin plots showing average TSS probe signal across the genome of cells treated with telmisartan or annatto relative to the vehicle-only-treated WT. White lines indicate the median and dotted lines the quartiles. Numerical values below the violin plot represent the mean methylation level of all TSSs for each treatment; n = 5 biological replicates, statistical analysis done using two-sided one-way ANOVA with Dunnett's correction. ****p = <0.0001.

provided independent models of mitochondrial hyperpolarization, including in the context of a disease.

We started by sorting untreated WT cells maintained under standard culture conditions. We set a first gate to include only MTG-positive cells within ±10% of the fluorescence intensity of the mean, which consistently resulted in approximately 30% of all MTG-positive cells (Fig. 7A, upper panel). We then set a second gate for TMRE-positive cells, sorting between cells with TMRE fluorescence in the

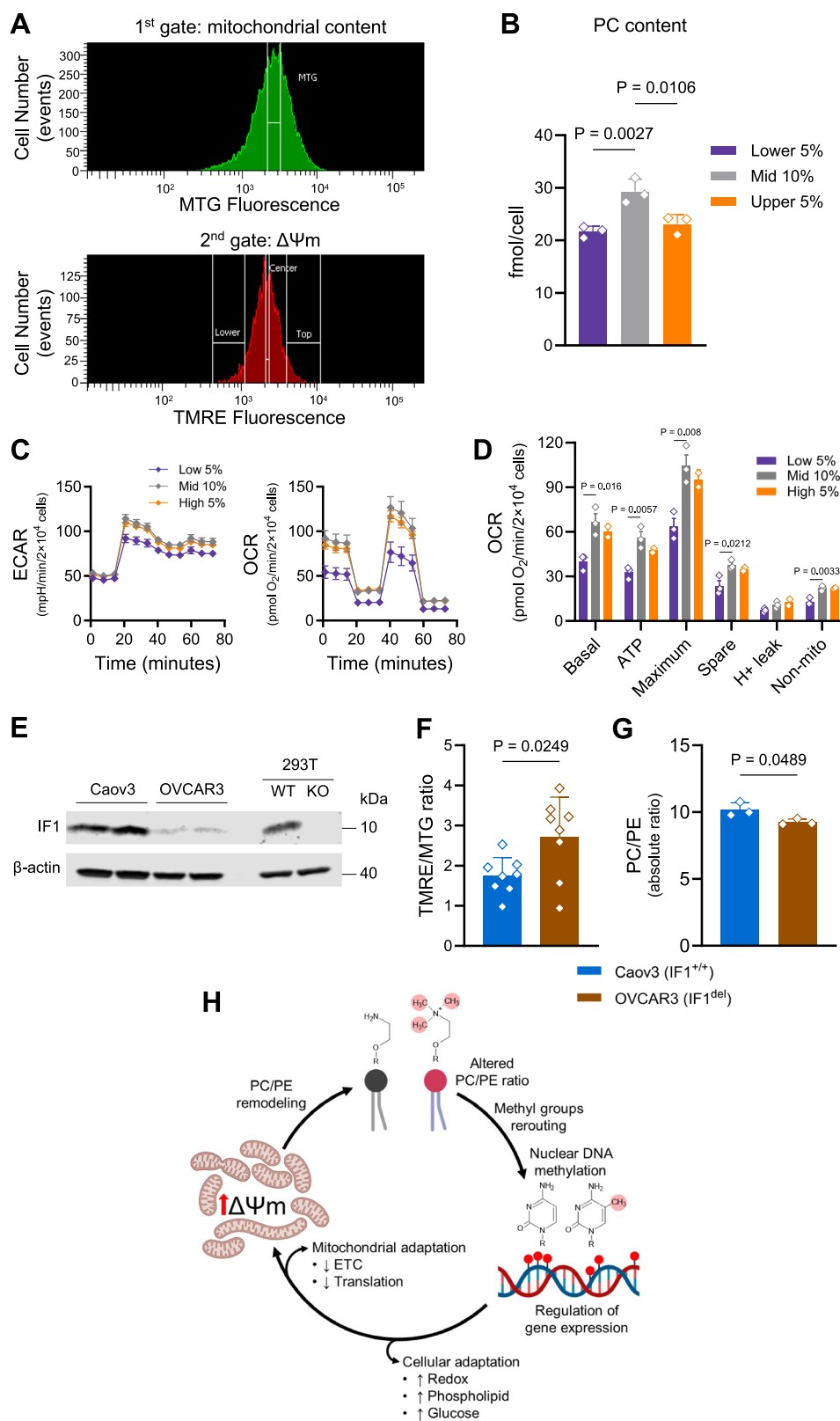

lower (depolarized), mid (average intensity of polarized mitochondria), and upper 5% (hyperpolarized) range (Fig. 7A, lower panel). These strict parameters assured no overlap between the populations, while controlling for changes in TMRE fluorescence based on ΔΨm and not the number of mitochondria. Cells from the gated population were then collected, and PC content was estimated as a proxy for phospholipid remodeling. We measured only PC because of limited

material, and because it was the phospholipid consistently decreased in all models tested herein. The sorted population of cells with hyperpolarized mitochondria had significantly less PC than the sorted population representing the average (10% mid) level of TMRE fluorescence (Fig. 7B). No differences in oxygen consumption were observed among these populations, consistent with preservation of mitochondrial function being a feature of hyperpolarized cells

**Fig. 7 | Naturally hyperpolarized cells remodel PC/PE content. A** Fluorescence-activated cell sorting gating parameters. Upper panel: 1st gate for MTG-positive cells with ±10% of the mean, resulting in 30% of the all MTG-positive population. Lower panel: 2nd gate for TMRE-positive cells with lower-, mid-, and upper 5% of TMRE-positive population. *Y* axis, cell count (events), *x* axis MTG or TMRE fluorescence in log scale. **B** PC content in the low-, mid- and upper 5% sorted cells. On the *y* axis, PC content in fmol/cell. Statistical difference by two-sided one-way ANOVA with Dunnett's post-test compared to mid; error bars represent ±SEM of $n = 3$ biological replicates with 2 experimental replicates. **C** Left graph shows ECAR, on the *y* axis in mpH/min/µg protein; right graph shows OCR, on the *y* axis in pmol $O_2$/min/µg protein ($n = 3$ for low and mid, $n = 2$ for high); error bars represent ±SEM. **D** Graphical representation of OCR-derived parameters from (**C**). Statistical difference by two-sided one-way ANOVA with Dunnett's post-test compared to mid; error bars represent ±SEM. **E)** Representative immunoblots of lysates from Caov3 and OVCAR3 cell lines, WT and IF1-KO cells were used as controls. **F** ΔΨm measured

in intact Caov3 and OVCAR3 cells; *y* axis, TMRE/MTG ratio. Statistical difference by two-sided unpaired Student's *t* test; error bars represent ±SEM of $n = 8$ biological replicates. **G** Absolute PC/PE ratio in ovarian cancer cells using lipidomics. Statistical difference by two-sided unpaired Student's *t* test; error bars represent ±SEM of $n = 3$ biological replicates. **H** Working model: chronic ΔΨm hyperpolarization leads to mitochondrial phospholipid remodeling by decreasing methylation of PC from PE, impacting the cellular PC/PE ratio and leading to a relative increase in PE. This helps regulate proton flux and maintain an optimal hyperpolarized state that is supportive of mitochondrial function. Methyl groups from SAM not used to generate PC are re-routed to the nucleus, rewiring DNA methylation. Differentially methylated loci drive an adaptive gene expression program, including inhibition of OXPHOS that further regulates proton flux and the ΔΨm, redox homeostasis, and glucose/lipid metabolism. Created in BioRender. Mori, M. (2025) https://BioRender.com/96n70n2.

(Fig. 7C, D). Interestingly, the depolarized population also had decreased PC (Fig. 7B), but this was accompanied by significant loss of oxygen consumption, ATP production, and maximum and spare respiratory capacities (Fig. 7C, D). Mitochondrial phospholipid content in depolarized cells has not been reported, but loss of PE and cardiolipin have been shown to have strong implications in the activity and efficiency of OXPHOS[45]. Therefore, it seems that PC levels alone and ΔΨm do not follow a continuum, and that it is likely that depolarized cells have reductions in other phospholipids important to maintain OXPHOS function that are not changed by mitochondrial hyperpolarization.

cBioPortal for Cancer Genomics is a publicly available compendium of large-scale cancer genomic data sets, including from The Cancer Genomic Atlas (TCGA), which provides the ability to visualize, analyze, and download data. As ovarian cancers have been shown to have different bioenergetics profiles, including increased ΔΨm[7], we used this tool to analyze the genomic landscape of ovarian cell lines. We identified OVCAR3 cells as bearing a deep homozygous deletion for IF1[47], which was not present in Caov3—a cell line from the same cancer sub-type, grade and race of the donor (Supplementary Fig. 7A). This genomic deletion resulted in significantly decreased IF1 protein levels in OVCAR3 relative to Caov3 (Fig. 7E, Supplementary Fig. 7B, C), which was accompanied by increased resting ΔΨm (Fig. 7F). Notably, while Caov3 were previously shown to use primarily OXPHOS for ATP production, OVCAR3 were found to rely both on OXPHOS and glycolysis[7], a phenotype likely associated with the deletion of IF1. We then generated lipidomics data and established the PC/PE ratio within each cell, which was used for comparison purposes since they were not isogenic. The PC/PE ratio of OVCAR3 was significantly decreased compared to that of Caov3 (Fig. 7G), consistent with the hyperpolarization of their mitochondria. Together, these results lend further support to the hypothesis that phospholipid remodeling is an adaptation to a chronic state of increased resting ΔΨm.

## Discussion

In this paper, we established that modulation of the ΔΨm is mechanistically linked to nuclear DNA methylation. We provide several lines of evidence to support that phospholipid remodeling, and not metabolites or oxidative stress, underlies the epigenetic changes. Our data also reveals a large-scale adaptation in transcriptional reprogramming, including through ETC and phospholipid metabolism genes, as a result of chronic mitochondrial hyperpolarization. Remarkably, the loci of these genes are also the ones that are differentially methylated at the DNA level. While the changes in gene expression could also be influenced by retargeting of repressive histones, like H3K9me3, analyses of transposable element (TE) expression and H3K9me3 peaks overlapping DEG promoters using ENCODE data (Supplementary Data 4) suggest that such contribution, if any, is minimal. Metabolic rewiring was observed in the IF1-KO cells, but it did not involve metabolites

previously associated with DNA methylation. Rather, we found increased antioxidant pathways, including through glutathione metabolism and the activation of the pentose phosphate pathway (PPP), which are likely adaptations to increases in mitochondrial ROS production due to a higher resting ΔΨm. Activation of the PPP helps explain previously reported effects of the ΔΨm on cell proliferation[12–14]. Interestingly, CRISPR/cas9 deletion of the rate-limiting enzyme of the PPP (Glucose-6-Phosphate Dehydrogenase, G6PD) led to a complete cell cycle arrest with features of cellular senescence in the IF1-KO cells (Supplementary Fig. 8)

In all models of ΔΨm hyperpolarization described herein, we found a change in the PC/PE ratio derived from a decrease in PC. Changes in cellular phospholipids through the loss of PE to PC methylation have been previously shown to cause histone hypermethylation in yeast devoid of PEMT or under starvation, both of which accumulated SAM[40]. In yeast, the divergence of methyl groups to histones was proposed as a means to recycle SAM and maintain the cellular methionine/methyl cycle, which would otherwise be halted by accumulation of this metabolite[41,48]. In our cells, neither histones were found to be hypermethylated nor did SAM accumulate, suggesting fundamental differences in the epigenetic outcomes associated with PC/PE remodeling in yeast and in mammalian cells. As the existence of enzymatic-driven DNA methylation in yeast remains debatable[42], it is possible that in mammalian cells, the DNA rather than histones is the acceptor of PE-derived methyl groups. Intuitively, the DNA would seem a more efficient way to maintain the recycling of methyl groups from SAM to S-adenosylhomocysteine (SAH), given the abundance of cytosines (2 billion) over that of histones (~240 million) in the mammalian genome[49]. The vast source of recycling sites at the DNA level may also explain the lack of accumulation of SAM in the IF1-KO cells, although it is possible that these differences simply reflect inhibition of activity (this study) vs total loss of PEMT as in previous work[41,48]. More studies are required to fully understand this issue.

Our methionine tracing data point to decreased PE to PC methylation, implying that PEMT activity might be sensitive to ΔΨm. How a rise in resting ΔΨm may impact PEMT, which is present in the interface between the ER and mitochondria and is regulated transcriptionally or by substrate availability[44], remains unclear. Increased local ROS in the ER-mitochondrial interface may provide a novel means of local redox regulation of the enzyme; interestingly, we found 3 putative redox-sensitive cysteines that might render PEMT amenable to this type of regulation using a prediction software (pCysMod.omicsbio.info). Oxidation of the enzyme to inhibit its activity might explain why ectopically expression of PEMT in the KO cells did not reverse the DNA hypermethylation phenotype. To our knowledge, there are no publicly available reports on the redox regulation of PEMT. Alternatively, mitochondrial hyperpolarization may immobilize PEMT within the inter-organellar nanodomains, affecting its function, similar to our recent findings on the $IP_3$ receptor[50]. It is also possible that the

decrease in PEMT activity was a response to the transcriptional changes involving phospholipids, particularly the generation of PC through the Kennedy reactions, as significant compensation between these two pathways of PC production is well-established[51]. Whether the decreased PEMT activity under our experimental conditions led to activation of the Kennedy or is a response to it remains to be defined. Further studies addressing these different questions are warranted.

That a chronic rise in the $\Delta\Psi$m leads to phospholipid remodeling is clear based on our data, but why such remodeling occurs remains to be fully defined. In brown adipose tissue, PE in the mitochondrial membrane was recently shown to regulate proton flux through UCP1, presumably affecting the $\Delta\Psi$m although this was not measured[23]. While our cells do not express UCP1, we speculate that an increase in relative PE content (by decreasing PC) in the mitochondrial membrane of IF1-KO cells could be deployed in a similar fashion, maintaining an optimal hyperpolarized state to support proper mitochondrial and cellular function. How these changes impact DNA methylation and transcription at specific loci remains unknown. Along the same lines, why modulation of ETC and mitochondrial translation genes was pervasive remains unclear, but they might be deployed as additional means to prevent excessive hyperpolarization of the inner mitochondrial membrane. Irrespective, the observation that chemicals can lead to similar rewiring raises important questions about how chronic increases in $\Delta\Psi$m, whether in disease (e.g., pulmonary hypertension or cancers) or environmentally induced, may impact health outcomes. A depiction of our working model is presented in Fig. 7H.

In summary, our results provide insights into the molecular and genomic adaptations associated with a chronic rise in resting $\Delta\Psi$m. Our data adds to the growing body of evidence identifying a relationship between phospholipids and mitochondria while revealing that the $\Delta\Psi$m can serve as a signal to communicate with the epigenome. Notably, this occurs in the absence of mitochondrial dysfunction. Overall, our data raise fundamental questions about the potential insidious effects of environmental exposures that affect the $\Delta\Psi$m, including their potential to influence health outcomes. It also highlights the possibility of modulating the $\Delta\Psi$m to treat diseases, such as ovarian cancer.

## Methods

### Cell lines

HEK293T cells carrying a tetracycline (Tet)-on inducible DN-POLG and the ATP5IF1 KO isogenic derivatives were provided by Dr. Navdeep Chandel (Northwestern University); these cells were previously described[13]. pcDNA3.1-hUCP4-NE was purchased from Addgene (plasmid # 102362), and pcDNA3.1 + /C-(K)DYK containing the human PEMT transcript variant 2 mRNA sequence (NM_007169.3) was purchased from GenScript (cat # OHu13080D). IF1-KO cells were used to generate isogenic derivatives ectopically expressing IF1 (IF1-OE), UCP4 (IF1-KO UCP4), or PEMT (IF1-KO PEMT-flag). Stable expression was obtained by transfecting the cells using Lipofectamine™ 3000 (Invitrogen™) and selecting multiple clones. IF1 was also deleted using CRISPR/Cas9 (GenScript, SC1678, sgRNA1 ATP5IF1-1, lot # U2615IA200-1/H330965 and sgRNA2 ATP5IF1-2, lot # U2615IA200-1/H330965) in HeLa. After transfection, cells were incubated with puromycin 1 µg/mL for 2 days to select transfected cells. Cells were then allowed to grow in the absence of puryomycin to remove selective pressure for plasmid integration, which also contains Cas9. Cells were expanded after single clone selection in complete DMEM. HEK293 cells were maintained in DMEM high glucose (D-glucose 4.5 g/L, 25 mm) supplemented with FBS 10%, pyruvate 1 mM, uridine 50 µg/mL (Sigma-Aldrich®), and 0.16% penicillin/streptomycin, hygromycin 150 µg/mL and, blasticidin 5 µg/mL at 37 °C and 5% CO$_2$. WT and HeLa IF1-KO cells were maintained in DMEM high glucose (D-glucose 4.5 g/L, 25 mm) supplemented with FBS 10%, and 1% penicillin/streptomycin, at 37 °C and 5% CO$_2$.

Human epithelial ovarian adenocarcinoma OVCAR3 (NIH:OVCAR3, cat # HTB-161™) and Caov3 (Caov3, cat # HTB-75™) cells were obtained from the American Type Culture Collection (ATCC). Both cell lines were cultured as previously described[52]. OVCAR3 cells were grown in RPMI 1640 medium (Gibco™) supplemented with 20% fetal bovine serum (FBS, Cytiva HyClone, Marlborough, MA), 0.01 mg/mL recombinant human Insulin (Gibco™), 100 U/mL penicillin, and 100 µg/mL streptomycin (Sigma-Aldrich®). Caov3 cells were grown in DMEM high glucose (Sigma-Aldrich®) supplemented with 10% FBS, 100 U/mL penicillin, and 100 µg/mL streptomycin.

### $\Delta\Psi$m measurements in intact cells and chemical treatments

$\Delta\Psi$m was accessed using tetramethylrhodamine ethyl ester (TMRE). $5 \times 10^5$ cells were plated in a six-well plate ($5.6 \times 10^4$ cells/cm$^2$) and incubated at 37 °C in a 5% CO$_2$ humidified incubator for 24 h. Medium was removed and washed once with FBS-free DMEM. Cells were incubated in 2 mL FBS-free DMEM with TMRE 25 nM and MitoTracker® Green (MTG) 50 nM at 37 °C for 15 min. Alternatively, MitoTracker® DeepRed (MTDR) 100 nM was used since HEK293T IF1-OE cells express EGFP. OVCAR3 and Caov3 cells were treated the same way but seeded in 12-well plates at 59,375 cells/well and 118,750 cells/well, respectively. To induce mild mitochondrial uncoupling with 2,4-dinitrophenol (DNP, Sigma-Aldrich®) without toxicity, we titrated several concentrations of DNP. Next, $1.25 \times 10^5$ cells were plated in a six-well plate and incubated at 37 °C in a 5% CO$_2$ humidified incubator with 25 µM for 3 days. For phosphate starvation experiments, recently published protocols using phosphate-free medium were followed[15]. Fluorescence was analyzed in BD LSRFortessa™ Cell Analyzer using FITC channel for MTG and PE-Cy7 channel for TMRE. 10,000 events were analyzed. At the end of each measurement, 10 µM FCCP (final concentration) was added and incubated for 10 min to allow full depolarization. Mean fluorescence values were converted as the ratio of mean TMRE subtracted by mean TMRE post-FCCP/MTG.

### Mitochondrial Ca$^{2+}$ uptake and $\Delta\Psi$m measurements in permeabilized cells

$\Delta\Psi$m was recorded simultaneously with [Ca$^{2+}$]$_c$ using a multi-excitation and dual-emission fluorometer (DeltaRAM; Horiba, New Jersey, USA) as described[53,54]. Cells were harvested and washed with cold Na-Hepes-EGTA buffer containing 120 mM NaCl, 5 mM KCl, 1 mM KH$_2$PO$_4$, 0.2 mM MgCl$_2$, and 20 mM Hepes-NaOH, pH 7.4. In 37 °C and under stirring condition the same aliquots of cells (1.8 mg) were permeabilized using 30–40 µg/ml Digitonin in 1.5 ml intracellular medium buffer (ICM:120 mM KCl, 10 mM NaCl, 1 mM KH$_2$PO$_4$, 20 mM Hepes-Tris, pH 7.2) supplemented with 5 µg/ml protease inhibitors leupeptin, antipain and pepstatin for 5 min. In all the experiments, 2 mM MgATP, 2 mM succinate, and 2 µM thapsigargin (Enzo), TMRM (1.5 µM), and 1.0 µM fura2FF (Kd = 4.5 µM, TEFLabs) were present. Uncoupler, FCCP, and oligomycin, (5 µM and 5 µg/ml, respectively) were applied at the end of each run to dissipate remaining $\Delta\Psi$m. TMRM fluorescence was recorded with 540 nm excitation and 580 nm emission, whereas furaFF fluorescence was recorded at 340 nm and 380 nm excitation and 500 nm emission. Calibration for maximum and minimum fura2FF response was performed by adding 2 mM Ca$^{2+}$ and 10 mM EGTA/Tris pH 8.5, respectively.

### Histone purification

Histones were purified according to Shechter et al. [55]. Briefly, ~$5 \times 10^6$ cells were incubated in hypotonic lysis buffer (Tris-HCl 10 mM, pH 8.0, KCl 1 mM, MgCl$_2$ 1.5 mM, and DTT 1 mM) on rotator at 4 °C for 30 min. Nuclei were pelleted at 10,000 × $g$ at 4 °C for 10 min. Supernatant was discarded and nuclei resuspended in 400 µL of H$_2$SO$_4$ 0.4 M. Acid extraction was performed on rotator at 4 °C overnight. Samples were centrifuged at 16,000 × $g$ at 4 °C for 10 min. Supernatant was transferred into new tubes, and histones precipitated with 132 µL of

trichloroacetic acid 100% (w/v). Samples were incubated on ice for 30 min and centrifuged at $16,000 \times g$ at 4 °C for 10 min. Supernatant was removed and pellet was washed thrice with ice-cold acetone by centrifugation at $16,000 \times g$ at 4 °C for 5 min. Histones were air-dried and quantified using BCA.

## Native PAGE and Immunoblotting
Experiments were performed as described previously[56,57]. Briefly, 50 μg of mitochondrial fractions were thawed on ice for 30 min. Samples were mixed with 8 μL digitonin 5% (w/v), 5 μL 4X Native PAGE sample buffer, and a sufficient volume of deionized to 20 μL (subtracting the volume of the mitochondrial fraction). Samples were incubated on ice for 20 min and centrifuged at $20,000 \times g$ at 4 °C for 10 min. 15 μL of the supernatant was transferred into new tubes. 2 μL of Coomassie G-250 sample additive was added per sample, mixed, and 12 μL of each sample was loaded. Electrophoresis in Native PAGE™ 3 to 12% gels was performed in XCell™ SureLock™ with NativePAGE™ (NP) buffer (anode) and Coomassie Brilliant Blue G-250 (CBBG) 0.02% (w/v) in NP buffer (dark blue cathode) at 150 V for 30 min. After 30 min, dark blue cathode buffer was removed and replaced with CBBG 0.002% (w/v) in NP buffer (light blue cathode). BN-PAGE gels were transferred to methanol-activated PVDF membranes in Dunn carbonate buffer (10 mM $NaHCO_3$, 3 mM $Na_2CO_3$). Transfer was performed at a constant current of 300 mA at 4 °C for 1 h using XCell™ Blot Module (Invitrogen™). Proteins were fixed/denatured with 8% acetic acid for 5 min, and CBBG was removed by washing thrice with methanol. Membranes were blocked using EveryBlot Blocking Buffer (Bio-Rad©) and incubated overnight with antibodies listed in Table S. Detection was performed in Odyssey® CLx Imager (LI-COR©).

## SDS-PAGE and immunoblotting
Protein samples were solubilized in cold RIPA Lysis and Extraction Buffer supplemented with Halt™ Protease and Phosphatase Inhibitor Cocktail (Thermo Scientific™) and centrifuged at $20,000 \times g$ for 10 min. The supernatant was mixed with NuPage LDS sample buffer (Invitrogen™), resolved by 4–20 or 16% Tris-glycine SDS-PAGE, and transferred to PVDF or nitrocellulose membrane. Detection was performed in Odyssey® CLx Imager (LI-COR©). Cell lysates (25 to 40 μg of protein) were loaded per well. Primary antibodies and dilutions are as follows: ATPIF1 (Millipore, ABC137, 1:400), MICU1 (Sigma, HPA034780, 1:500), MICU2 (Abcam, ab101465 1:500 and Bethyl, A300-BL 19212, 1:1000), MCU (Sigma-Aldrich, AM Ab91189, 1:1000), EMRE (Bethyl, A300-BL 19208,1:1000), mitochondrial HSP70 (Sigma-Aldrich, G4045, 1:1000), AMPKα (Cell Signaling, 2532, 1:1000), Phospho-AMPKα (Thr172) (Cell Signaling, 2531, 1:400), β-actin (Cell Signaling, 4967, 1:1000), GAPDH (Sigma-Aldrich, G8795, 1:1000), UCP4 (*SLC25A27*) (Invitrogen, PA5-69265, 1:400), H3K4me3 (ActiveMotif, 39159, 1:1000), H3K9me3 (ActiveMotif, 39161, 1:1000), H3K27me3 (ActiveMotif, 39156, 1:1000), H3K79me3 (Cell Signaling, 4260, 1:1000), H3 (ActiveMotif, 61799, 1:1000), ATP synthase C antibody (Abcam, ab181243, 1:1000), ATP5O (Cell Signaling, 92658, 1:1000), ATP5B (Cell Signaling, 85001, 1:1000), MT-ATP8 (Cell Signaling, 96857, 1:1000), DNMT1 (Cell Signaling, 5032, 1:500), DNMT3A (Cell Signaling, 49768, 1:500), DNMT3B (Cell Signaling, 57868, 1:500), LC3A (Cell Signaling, 4599, 1:1000), TDG (Cell Signaling, 99105, 1:1000), TET1 (Cell Signaling, 40142, 1:500), TET3 (Cell Signaling, 57868, 1:500), G6PD (Invitrogen, HPA000834, 1:1000), p21 (Cell Signaling, 2947, 1:1000). Secondary antibodies and dilutions: goat-anti rabbit IRDye 680RD (LI-COR 925-681817, 1:10,000), and goat-anti mouse 680RD (LI-COR 925-68070, 1:15,000).

## OCR in intact cells
OCR was accessed in Seahorse XFe96 Analyzer (Agilent Technologies). Seahorse XFe96/XF Pro Cell Culture Microplates (Agilent Technologies) were coated with poly-D-lysine (Gibco™) following the manufacturer's protocol. Three to four independently growing flasks of each genotype (WT, IF1-KO, IF1-OE, and IF1-KO UCP4) with 60–80% confluency cells were trypsinized and cell density estimated in Bio-Rad TC20 automated cell counter. $2 \times 10^4$ cells were plated per well and incubated at 37 °C in a 5% $CO_2$ humidified incubator for 18 h. Cartridge was hydrated according to the manufacturer's instruction. Cells were washed twice by dilution with XF Assay medium, supplemented with D-glucose 10 mM, pyruvate 1 mM, and glutamine 2 mM. Cells were incubated in a $CO_2$-free BOD-type incubator at 37 °C for 1 h. MitoStress protocol was used to access mitochondrial function. Final concentration of inhibitors was: oligomycin 1 μM (Port A), FCCP 1 μM (Port B), and rotenone 1 μM plus antimycin A 1 μM (Port C) (all Sigma-Aldrich®). At the end of the assay, the medium was carefully removed, and the cells were washed with ice-cold PBS. PBS was removed, and cells were lysed with 20 μL of RIPA Lysis and Extraction Buffer supplemented with Halt™ Protease and Phosphatase Inhibitor Cocktail (Thermo Scientific™). Protein content was estimated using Pierce™ BCA Protein Assay Kits (ThermoFisher™). OCR values were corrected by protein content of each given well. The average of 6–8 replicates from the original independently growing flask was considered one independent sample ($n = 3$). OCR and ECAR results were transformed in $O_2$ pmol/min/mg protein and mpH/min/mg protein.

## Transmission electron microscopy analysis
WT, KO, and OE cells were grown in duplicates in T-175 flasks till confluency. Cells were collected by trypsinization and washed thrice in PBS. Subsequently, the cells were fixed at room temperature for 1 h in McDowell and Trump's fixative and washed thrice in 0.1 M cacodylate buffer prior post-fixation in osmium tetroxide 1% (Electron Microscopy Sciences, Hatfield, PA). Cells were rinsed in distilled water and dehydrated in an ethanol series transitioning to acetone. The samples were then infiltrated with EPON-812 Resin (Electron Microscopy Sciences). After polymerization, blocks were trimmed and semithin sections (-0.5 μm thick) were cut, mounted on glass slides, and stained with 1% toluidine blue in 1% sodium borate for quality check examination with a light microscope. After trimming, ultrathin sections (-70–100 nm thick) were cut from selected blocks, placed onto 200 mesh copper grids, and counterstained with 2% uranyl acetate and lead citrate. Digital images were captured using a Gatan Orius SC1000 camera attached to an FEI Co. Tecnai T12 transmission electron microscope operated at 80 kV. For quantification, at least 100 images for each study group were captured, and EM subcellular analysis was made using Gatan Digital Micrograph software. Statistical analysis was performed using the non-parametric Mann–Whitney $U$ test, and data are presented as mean values. $P$ values < 0.05 were considered statistically significant.

## Live-cell microscopic imaging of GSH:GSSG
Experiments were performed as described previously[58–60]. WT and IF1-KO HEK cells were transfected with either cytoplasmic or mitochondrial matrix-targeted Grx1-roGFPE2 plasmids. Epifluorescence imaging of GSSG:GSH was carried out using a fluorescence wide field imaging system consisting of a ProEM1024 EMCCD (Princeton Instruments), fitted to Leica DMI 6000B inverted epifluorescence microscope equipped with a Sutter DG4 light source. Fluorescence excitation was achieved with a custom dichroic derived from Chroma #59022 modified to enhance short-wavelength excitation and bandpass filters specific to fluorophores. Grx1-roGFP2 was imaged with dual excitation of 414/10 and 480/15 nm combined with a 515/30 nm emission filter. Typical time resolution was 10 s. Calibration of the probe was performed by adding 2 mM dithiothreitol (DTT) twice for minimal ratio value and after washout of DTT, 0.2 mM $H_2O_2$ for maximal ratio value. All the experiments were performed in ECM containing 0.25% BSA at 37 °C.

## RNA isolation, RNA-seq, and gene expression analysis
RNA was extracted from WT, IF1-KO, and IF1-OE cell lines ($n = 3$) or IF1-KO and IF1-KO-UCP4 ($n = 4$) for RNA-seq. RNA was isolated in QIAcube

using QIAshredder columns and RNeasy Mini with DNase digestion (QIAGEN). RNA concentration was estimated in NanoDrop™ 2000 and Qubit™ 3 fluorometer. RNA integrity was analyzed in an Agilent 2100 Bioanalyzer using an Agilent RNA 6000 Nano Kit, and poly-A-selected for RNA-seq library generation. Paired-end libraries were sequenced to 100 base pairs (bp) or single-end libraries were sequenced to 75 bp on a NovaSeq 6000 platform (Illumina, San Diego, CA). Raw fastq files were filtered to exclude reads with phred quality score <20 using the FASTX-Toolkit fastq_quality_filter command-line utility (http://hannonlab.cshl.edu/fastx_toolkit/). The STAR RNA-Seq aligner was used to align quality-filtered reads to the human genome (hg38), and reads with mapping quality <20 were discarded with the SAMtools view command[61,62]. Reads aligning to Gencode GRCh38 Release 32 human genes were quantified with the Subread package featureCounts utility[63]. Differential expression analysis was performed in the R statistical programming environment v4.1.2 using DeSeq2 v1.34.0 (https://www.r-project.org/)[64]. Genes with an adjusted $p$ value < 0.05 were classified as differentially expressed.

## DNA methylation arrays and data analyses

Genomic DNA was extracted from the cells, and samples were bisulfite-converted using an EZ DNA Methylation kit (Zymo Research) following the manufacturer's protocol. Differential methylation at the CpG dinucleotide level was conducted using Human Infinium Methylation EPIC 850 K v2.0 BeadChip arrays (Illumina) following the Infinium methylation protocol. Methylation analysis was performed with the R Bioconductor package ChAMP v2.22.0[65]. Raw IDAT files were loaded into R with the champ.load function and probes that failed with detection with $p$ value > 0.01 or <3 beads in 5% of samples were removed. Multi-hit, non-CpG, Chromosome X/Y, and SNP-overlapping probes were filtered prior to analysis. Beta-mixture quantile normalization was applied to correct for type-II probe bias[66]. Differentially methylated probes (DMPs) were identified with the champ.DMP function, which implements limma for calculation of the $t$ statistic and $p$ value for each probe[67]. For the promoter analysis, probes were extracted with the EPIC TSS1500 feature annotation. The mean $\beta$ value was reported for genes with more than one probe mapping to the promoter region. Hypermethylated or hypomethylated probes were defined by a change in mean $\beta$ value > 10 or <10, respectively, and an adjusted $p$ value < 0.05. The Benjamini-Hochberg procedure was applied to adjust $p$ values for multiple testing correction.

## Steady-state metabolomics data collection and analysis

Cells grown at 60–80% confluency from six independent cultures were trypsinized and cell density estimated in Bio-Rad TC20 automated cell counter. $5 \times 10^6$ cells were plated in 100 mm dishes and incubated at 37 °C and 5% $CO_2$ for 6 h to allow cells to attach. Cells were then washed once with 10 mL ice-cold PBS and placed on ice. 1 mL UHPLC grade methanol:water 4:1, pre-chilled at −80 °C, was added in each dish and incubated at −80 °C for 15 min. Dishes were then placed on dry ice, and cells were scraped off the dish. Lysates were transferred to pre-chilled 1.5 mL microtubes and centrifuged at $20,000 \times g$ a 4 °C for 10 min. 900 μL of supernatant was transferred to new 1.5 mL microtubes. Solvent was dried using Vacufuge Plus (Eppendorf©), and analytes were kept in −80 °C until Untargeted Metabolomics was performed. The samples were resuspended in acetonitrile-water (2%:98%) v/v, vortexed for 10 s, and centrifuged (Eppendorf Centrifuge 5425 R) at 14,000 relative centrifugal force (rcf) for 10 min at 4 °C. The soluble fraction of the extract was transferred to 2 mL autosampler vial (12 mm × 32 mm height vial, 12 mm screw cap, and PTFE/silicone septa, Agilent) containing a microvolume insert (Agilent).

Samples were analyzed using an ultra-high performance liquid chromatograph (Vanquish™ Horizon UHPLC, Thermo Scientific) coupled to a high-resolution mass spectrometer (Orbitrap Fusion™

Tribrid, Thermo Scientific). EASY-Max NG™ was used as the ionization source, operated in the heated-electrospray ionization (H-ESI) configuration. Prior to measurements, the mass spectrometer was calibrated using FlexMix (Thermo Scientific) following manufacture directions. EASY-IC™ (Thermo Scientific) was used during data collection. LC-MS data were collected from individual samples, system blanks, and a pooled quality control (QC), which was generated by combining aliquots from every sample in the data set into one sample. MS data were collected with an anticipated LC peak width of 8 s and a default charge of 1. MS data were acquired at 120,000 resolution from m/z 100–1000 with an RF lens of 60% and maximum injection time of 50 ms. Prior to acquiring LC-MS data for samples, liquid chromatography−tandem mass spectrometry (LC-MS/MS) data were acquired using the AcquireX (Thermo Scientific) deep scan methodology. The inclusion list was generated and updated via AcquireX with a low and high mass tolerance of 5 part-per-million (ppm) mass error. An intensity filter was applied with an intensity threshold of $2.0 \times 10^4$. Dynamic exclusion was used with the following parameters: exclude after $n = 3$ times; if it occurs within 15 s; exclusion duration of 6 s; a low mass tolerance of 5 ppm mass error; a high mass tolerance of 5 ppm mass error; and excluding isotopes.

Data files (.raw) were processed with Compound Discoverer 3.3.0.550 (ThermoFisher Scientific) to identify unique molecular features and, where possible, annotate them with chemical names. Features with distinct, measured, accurate mass, unique retention time, and MS/MS data were tabulated after removal of isotope peaks, blank contaminants, and noise artifacts from the data. Spearman's $p$ (evaluates a monotonic response), Pearson's $r$ (evaluates a linear response), and coefficient of determination (R2, evaluates fit to a linear model); for Spearman and Pearson correlations, $p = 0.05$ was used as the statistical metric for significance. For every feature in the data set, a value of these metrics was calculated to evaluate the signal response for that feature over the QC range. Any feature for which the value does not meet the filtering parameter, or any feature with a negative correlation (Spearman or Pearson), is filtered out of the dataset, ensuring that only features displaying positive correlations within the $p = 0.05$ parameter are retained for further evaluation and interpretation.

## Methyl-tracing data acquisition and analysis

WT and IF1-KO cells were grown for 4 (four) days in DMEM, high glucose, no glutamine, no methionine, no cystine (Gibco™, cat # 21013024), supplemented with dialyzed FBS 10%, pyruvate 1 mM, uridine 50 μg/mL, glutamine 4 mM (all Gibco™), cystine-HCl 0.2 mM (Thermo Scientific Chemicals, cat # J61651.09), and L-Methionine-(methyl-d3) 0.2 mM (Sigma-Aldrich, cat # 300616), 0.16% penicillin/streptomycin, hygromycin 150 μg/mL, and blasticidin 5 μg/mL. 60–80% confluent cells from three independently growing 100 mm dishes (60 cm²) were trypsinized and cell density estimated in Bio-Rad TC20 automated cell counter. Metabolite extraction was performed as described above. After extraction, samples were dried down in a centrifugal vacuum concentrator and dissolved in 4/1 acetonitrile/water mixture.

Metabolite analysis was conducted on a QExactive HF-X mass spectrometer equipped with an H-ESI II probe. The mass spectrometer was coupled to a Vanquish binary UPLC system (Thermo Fisher Scientific, San Jose, CA). For chromatographic separation prior to mass analysis, 5 μL of the sample was injected onto a BEH Z-HILIC column (100 mm, 1.7 μM particle size, 2.1 mm internal diameter, Waters). Samples were diluted 1:5 in acetonitrile for the analysis. For chromatographic separation, Mobile phase A was 15 mM ammonium bicarbonate (pH 9.0) in 90% water and 10% acetonitrile, and mobile phase B was 15 mM ammonium bicarbonate (pH 9.0) in 95% acetonitrile and 5% water. Water and acetonitrile were purchased from Fisher and were Optima LC/MS grade, analytical standards were from Ammonium bicarbonate powder was purchased from Merck. The column oven was

held at 40 °C, and autosampler at 4 °C. The chromatographic gradient was carried out at a flow rate of 0.5 ml/min as follows: 0.75 min initial hold at 95% B; 0.75–3.00 min linear gradient from 95% to 30% B, 1.00 min isocratic hold at 30% B. B was brought back to 95% over 0.50 minutes, after which the column was re-equilibrated under initial conditions. Each sample was analyzed in both positive and negative modes. Retention times were determined using authentic standards; in the absence of an analytical standard, a PRM experiment was conducted on candidate peaks (resolution 30,000, 1.0 Da isolation window, stepped N(CE) 30, 50, 150), and compared to publicly available spectra. Raw data were converted to mzML and processed using emzed[68]. Theoretical isotopologue m/z values for methyl-labeled metabolites were determined using the exact mass of the M0 isotopologue as well as CD3. Raw peak areas were calculated by integrating an RT window determined by m/z, chemical standards, fragmentation patterns, and isotopologue patterns.

### Sub-chronic chemical exposures

$2 \times 10^5$ cells were plated in six-well plates ($2.2 \times 10^4$ cells/cm²) and incubated for 24 h. Cells were then treated with telmisartan 2 µM (Cayman Chemicals, 11615) and annatto 10 µM (D.D. Williamson, Annatto EM 25 AP) for 10 days. Fresh media with chemicals of interest were replaced daily. At day 10, cells were collected and plated for Seahorse analysis, or aliquoted for PC/PE analysis, DNA, RNA, and protein isolation.

### Estimation of cellular PE and PC content

PC and PE were estimated using phosphatidylcholine and phosphotidylethanolamine assay kit (Abcam©) according to the manufacturer's instructions. Briefly, $2 \times 10^6$ cells for PC and $1 \times 10^6$ cells for PE were pelleted. PE was extracted in peroxide-carbonyl-free Triton X-100 (Sigma-Aldrich®) 5% (v/v) in deionized water. For all samples, background measurement was added to account for the intracellular levels of the products of the reaction catalyzed by PC and PE converter enzymes. Results were calculated from the standard curve provided and presented as nmol of phospholipid per $1 \times 10^6$ cells.

### Mitochondrial isolation

Mitochondria were isolated as previously described[69] with slight modifications. Briefly, WT and IF1-KO cells were grown in 2 (two) T-175 flasks until they reached 70–80% confluency. Cells were trypsinized, and cell suspensions were centrifuged at $300 \times g$ at 4 °C for 10 min. Supernatant was removed, and pellets were washed once with ice-cold MSHE buffer (mannitol 210 mM, sucrose 70 mM, HEPES-KOH 10 mM, pH 7.2, EDTA 2 mM, EGTA 1 mM, DTT 5 mM). Pellets were resuspended in 2 mL of MSHE buffer, and cells were lysed using Omni Soft Tissue Tip™ (Omni, Inc) at full speed for 5 s. Nuclei and membranes were pelleted by centrifugation at $700 \times g$ at 4 °C for 12 min. Mitochondria-enriched supernatants were transferred into new pre-chilled 1.5 mL microtubes and centrifuged at $9000 \times g$ at 4 °C for 10 min. Protein content was estimated using Pierce™ BCA Protein Assay Kits (ThermoFisher™). 50 µg of mitochondrial fractions were aliquoted in 0.6 mL microtubes, flash-frozen in liquid nitrogen, and stored at −80 °C.

### HPTLC materials and lipid analysis

Lipid standards 1-stearoyl-2-oleoyl-sn-glycero-3-phosphocholine (18:0-18:1 PC; #850467), 1-stearoyl-2-oleoyl-sn-glycero-3-phosphoethanolamine (18:0-18:1 PE; #850758), and 1-stearoyl-2-oleoyl-sn-glycero-3-phospho-L-serine (18:0-18:1 PS; #840039) were obtained from Avanti Polar Lipids. Silica gel 60 HPTLC plates (Supelco, #105641) and all solvents of HPLC grade were purchased from EMD Millipore.

Lipids from mitochondria-enriched fractions and total cellular lipids were extracted with methanol/chloroform/water (final ratio 2:2:1.8) according to Bligh and Dyer[70], with modifications[71] and analyzed by automated HPTLC. Briefly: HPTLC plates were prewashed with methanol/ethyl acetate (6:4, v/v) and activated (110 °C, 30 min). Dried lipid extracts were dissolved in 200–400 µl chloroform/methanol (2:1, v/v) and loaded onto the HPTLC plate along with a dilution series of lipid standards using the CAMAG Linomat 5 loading unit. Phospholipids were resolved in two steps: to 90 mm with dichloromethane/ethyl acetate/acetone (80:16:4,v/v) and then to 90 mm with chloroform/ethyl acetate/acetone/isopropanol/ethanol/methanol/water/acetic acid (30:6:6:6:16:28:6:2, v/v)[71] using the CAMAG automated AMD2 multi-development system. For phospholipids visualization, developed plates were submerged in a 10% $CuSO_4$ in 8% $H_3PO_4$ aqueous solution for 4 sec, followed by 10 min of drying at room temperature, and subsequent quantitative charring at 167 °C for 10–15 min. The cupric sulfate-based fluorescence signal was detected with the 540/30–590/20 filter sets. PC, PE, and PS lipid species were identified according to their positions on the HPTLC plate and quantified by comparing integrated fluorescent signals to those of the lipid standards, which were resolved in parallel. Quantifications of lipids on digital images of chromatograms were performed with ImageJ by drawing a uniform rectangular grid to encompass each band per dilution series at a given Rf value. Two-sample two-tailed t-tests were used to assess differences between conditions.

### Fluorescence-activated cell sorting

Approximately $1.6$–$2.4 \times 10^8$ HEK293 cells were loaded with MTG 50 nM and TMRE 25 nM as described above. Cells were washed once, counted, and cell density adjusted to $4$–$5 \times 10^6$ cells/mL in complete DMEM with TMRE 10 nM for a prolonged time due to sorting. Cells were isolated using a BD Symphony S6 cell sorter (Becton Dickinson Biosciences, San Jose, CA) equipped with FACSDiVa software. Initially, a "scatter" gate was set on a forward scatter (FSC-A) versus side scatter (SSC-A) dot plot to isolate the principal population of cells free of debris. Subsequently, cells were consecutively gated on a side scatter height (SSC-H) versus width (SSC-W), then a forward scatter height (FSC-H) versus width (FSC-W) dot plot to isolate single cells. The center 30% (MFI) of cells stained with MTG (Ex: 488; Em: 537) were gated and examined on a TMRE (Ex: 561/Em: 585) histogram. The bottom and top 5%, along with the center ~10% of TMRE-stained cells, were then collected, washed, and flash-frozen for further analysis. Additionally, $2$–$3 \times 10^7$ HEK293 cells from three independent growing 150 cm² dishes were loaded with MTG 50 nM and TMRE 25 nM and processed as described above. $2 \times 10^5$ cells were sorted per gating parameter as described above and plated on a pre-coated Seahorse XFe96/XF Pro Cell Culture Microplates. Cells were allowed to attach for 18 h, and MitoStress Test was performed as described above.

### Lipidomics sample preparation and data analysis

OVCAR3 and Caov3 cells were seeded in six-well plates at $3 \times 10^5$ cells/well and allowed to grow for 48 h. Cells were rinsed with PBS once, 0.25% trypsin-EDTA (Thermo Fisher Scientific) was added, and cells were allowed to detach for 4 minutes. Trypsin was neutralized, cells were washed once with PBS, and the supernatant was removed; pellets were flash-frozen and stored at −20 °C. At the time of lipidomics, passage numbers used were P9 and P10 for OVCAR3 cells and P6 and P24 for Caov3 cells. Lipidomics analysis was performed at the Department of Chemistry Mass Spectrometry Core Laboratory of the University of North Carolina. Methyl tert-butyl ether, 1 mL, was added to the cell pellets, vortexed, and then transferred to an Eppendorf tube. 300 µL of methanol with an internal standard was added, and samples were shaken for 10 min. In all, 200 µL of water was added to facilitate phase separation. The extracts were centrifuged at $20,000 \times g$ for 10 min. The top layer was removed, dried down, and reconstituted in 100 µL of isopropanol for analysis. Avanti's deuterated lipid mix, Equisplash, was used as an internal standard. This was spiked into the methanol at 1.5 µg/mL and used for extraction.

Analysis was performed using a Thermo Q Exactive Plus coupled to a Waters Acquity H-Class LC. A 100 mm × 2.1 mm, 2.1 μm Waters BEH C18 column was used for separations. The following mobile phases were used: A- 60/40 ACN/H20, B- 90/10 IPA/ACN; both mobile phases had 10 mM Ammonium Formate and 0.1% Formic Acid. A flow rate of 0.2 mL/min was used. Starting composition was 32% B, which increased to 40% B at 1 min (held until 1.5 min), then 45% B at 4 minutes. This was increased to 50% B at 5 min, 60% B at 8 min, 70% B at 11 min, and 80% B at 14 min (held until 16 min). At 16 min, the composition switched back to starting conditions (32% B) and was held for 4 min to re-equilibrate the column. Samples were analyzed in positive/negative switching ionization mode with top five data-dependent fragmentation. Raw data was analyzed by LipidSearch. Lipids were identified by MS2 fragmentation (mass error of precursor = 5 ppm, mass error of product = 8 ppm). The identifications were generated individually for each sample and then aligned by grouping the samples (OxPAPC = C, HF = S1, Con = S2). Normalization was performed using EquiSplash from Avanti. The data were exported into Excel and normalized by cell counts.

### Analysis of TEs and H3K9me3 peaks using ENCODE data

The ENCFF762ELE data set containing peak calls corresponding to Hg38 H3K9me3 enrichment in bed format was obtained from the ENCODE Project[72] (www.encodeproject.org, accessed on 13 August 2024). The UCSC liftOver command-line utility was employed to obtain Hg19 genomic coordinates for the ENCODE H3K9me3 peaks[73]. Promoters, defined as 1.5 kb upstream and downstream of TSSs, for DEGs in the IF1-KO vs. WT comparison were intersected with the H3K9me3 peaks with bedtools intersect function[74].

Quantitative comparison of TE expression between the IF1-KO and WT was performed with the REdiscoverTE computational pipeline[75]. Transcriptome expression was quantified with Salmon v1.10.0[76]. TE expression was summarized at the class and subfamily level with the REdiscoverTE rollup function. The R DESeq2 v1.44.0 package was used to test for differential expression of TEs[64].

### Statistical analysis

Statistical analysis was performed using GraphPad Prism version 9.0. When comparing two groups, an unpaired Student's $t$ test was used. For comparisons involving more than one group, one-way ANOVA with Dunnett's or Tukey's post-test was performed. All statistical tests were two-sided. In all cases, tests were always two-sided, unless otherwise stated.

### Reporting summary

Further information on research design is available in the Nature Portfolio Reporting Summary linked to this article.

## Data availability

The RNA-seq and DNA methylation data generated in this study have been deposited in the GEO database under accession numbers GSE295015, GSE295296, and GSE295300. Source data are provided with this paper.

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

## Acknowledgements
We are grateful for the technical assistance of staff members of the Epigenomics, Metabolomics and Molecular Genomics Core Facilities at NIEHS, including Gregory Solomon, Nicole Reeves, Jason Malphurs, Laura Wharey and Dr. Kevin Gerrish, Ms Maria Teresa Castromonte (Thomas Jefferson University) for the help in redox imaging, and Dr. Brandie Ehrmann from the UNC Mass Spectrometry Core, for the help with the lipidomics analysis. We are also thankful to Dr. Nicole Taube for helping with the Seahorse of PEMT-overexpressing cells. We also thank Dr. William Copeland (Genome Integrity and Structural Biology Laboratory, NIEHS) and Dr. Xiaoling Li (Signal Transduction Laboratory, NIEHS) for critical review of the manuscript. This work was funded by intramural research funds to J.H.S., and NIH grants R01DK125897-03S1 and R01GM151536-01 to G.H., and R01 CA256710 to I.R. M.O. is funded by the Carolina Cancer Nanotechnology T32 CA196589 grant, and TR by MRU IRGF.

## Author contributions
M.P.M. and J.H.S. conceptualized the study, designed the experiments, and wrote the manuscript. M.P.M. did all cell culture experiments, including generating the genetic models, ΔΨm measurements and preparing samples for the 'omics' approaches and data analyses. O.A.L. performed some RNA-seq experiments; A.B. analyzed the DNA methylation and RNA-seq data; C.N. performed the TEM and part of steady-state metabolomics; B.R. performed methionine tracing and its initial analysis; G.H. designed and K.T.H. and P.H. performed experiments in permeabilized cells and with roGFP and analyzed data. B.P.R. and M.O. performed ΔΨm and lipidomic analyses in ovarian cancer cell lines. C.D.B. set up parameters and performed the sorting of the cells based on ΔΨm. T.R. performed and analyzed the HPTLC experiments. G.H. and I.R. revised the manuscript.

## Funding

## Competing interests
The authors declare no competing interests.
