## [Peer Review file · Nature Communications]

Mitochondrial membrane hyperpolarization modulates nuclear DNA methylation and gene expression through phospholipid remodeling

Corresponding Author: Dr Janine Santos

Version 0:

Reviewer comments:

Reviewer #1

(Remarks to the Author)

Mori et al:

The authors use IF1-KO cell lines to model the impact of chronic mitochondrial hyperpolarisation. Their system is shown to significantly enhance mitochondrial membrane potential ($\Delta\Psi$ M), providing a tool to analyse this phenomenon in cells. The authors perform bulk RNA-Seq in their IF1-KO cell line, identifying downregulation of mitochondrial and respiration genes, and upregulation of phospholipid-modifying and glucose metabolism genes compared to their control cells. In addition, the authors identify significant alterations in DNA methylation profiles during chronic mitochondrial hyperpolarisation. Using lipidomics and methyl tracing experiments, the authors identify reductions in PE:PC ratios in cells with elevated $\Delta\Psi$ M, with concomitant increases in methylation of cytosines. The authors demonstrate that similar effects can be observed in cells exposed to chemical compounds and in cancer cells that weakly express IF1. Overall, the authors propose a model whereby elevated $\Delta\Psi$ M disrupts PE:PC remodelling, rerouting methyl groups for DNA methylation and altering gene expression.

The findings are of significant interest, however, as presented, some of the data do not support the claims made in the manuscript. We hope our comments and suggestions will clarify specific points and improve the manuscript.

Comments:

- It is an interesting finding that mitochondrial membrane depolarisation can signal down to the level of chromatin and change the DNA methylation landscape; it is remarkable to see promoter methylation gain. Such an association between high TMRE/MTG and global methylation has been observed before in human pluripotent stem cells and reprogramming to naïve pluripotency (PMID: 28843285). However, in this system, high TMRE/MTG was associated primarily with widespread global demethylation, but also promoter CpG island methylation or CGIs (promoter CGIs are normally not methylated). Since in this manuscript the focus is specifically on the effect of membrane potential on hypermethylation, how do the authors explain the different effects in the two experimental systems? Is this indicating that the effect is not direct and there is an intermediate mechanism that mediates the effect on DNA methylation? If so, would this be cell type specific?
- “Unlike previously reported¹⁶, we found no changes at the protein levels of the different subunits of the mitochondrial Ca²⁺ uniporter complex (mtCU)” – 1 out of 3 of the IF1-KO lysates show elevated levels; it is too strong to say “no changes at the protein levels”. Some statistics on the quantifications shown in Fig S1C would help the authors make such quantitative statements here.
- “we also did not find changes in total or phosphorylated AMPK”. This statement goes against the quantification shown in Fig S1D, where p-AMPK in the IF1-KO cells are evidently higher than in the WT cells. Although, I am confused by the quantification, which shows an increase in p-AMPK (up by more than 1.5-fold in one of the repeats), while the blot does not show such an increase by eye.
- “These data are consistent with a fraction of IF1 being bound to complex V in WT cells grown under standard cell culture conditions.” The BN-PAGE experiment in Fig 1C is insufficient evidence to suggest that IF1 is “bound”, as it is labelled on the blot. Moreover, ATP5O is migrating above 480 kDa whereas IF1 appears to migrate below the 480 kDa marker based on the blot shown. To say that IF1 is “bound”, the authors would need to perform a pulldown assay for ATP5O and confirm that IF1 is co-precipitated (or similar assay). Based on this BN-PAGE blot, I would argue that the authors could, at most, say that IF1 is “co-migrating” rather than “bound”.
- “Collectively, these data support the conclusion that the increased resting $\Delta\Psi$ M in the KO cells relies on a significant contribution of ATP hydrolysis under standard cell culture conditions.” Have the authors confirmed that ATP hydrolysis is indeed altered in their model when cultured in glucose compared to galactose (see Fig 1D)? This should be measurable

using a colorimetric assay, for example. It is too speculative to say that “By limiting the pool of glycolytic ATP that can be hydrolyzed by the ATP synthase, growth in galactose would be predicted to decrease $\Delta\Psi_m$.” – this prediction needs to be validated. This is also interesting to confirm given the reported roles of IF1 in inhibiting ATP synthase activity (discussed by the authors earlier in the paragraph).

- Figure 2D indicates enrichment of KRAB domain genes in the RNA-seq data. This is relevant as these are transcription factors (TFs) and can recruit co-repressors (including DNMTs) to the promoters. This could indicate that histone regulation (heterochromatin formation by recruitment of HP1 by KRAB TFs for example) could explain the gene expression changes. A subset of genes might be directly repressed by DNA methylation. I appreciate that the authors have looked at histones in WBs (Figure S4) – however, retargeting of repressive histone marks to different genomic locations would not be readily visible in a WB global analysis. ChIP-seq or Cut&Run of specific repressive histone marks (H3K9me2/3 for example) would be more informative. As such, this interpretation might not be correct: “these data suggest that the methylation of the DNA, but not of histones, is primarily affected by a decrease in the cellular PC/PE ratio in mammalian cells”
- “In agreement, we found that the mitochondria of WT and KO cells could further hyperpolarize when they were put under phosphate depleted conditions (Fig. S2F), recapitulating the effects recently reported for other cells¹⁵” this is interesting...perhaps IF1 is not the only inhibitor of ATP synthase? And that in fact IF1 has other roles in the signalling network? Have the authors checked DNA methylation in this condition of hyperpolarisation? According to their model, this should further modify the DNA methylation landscape, perhaps even more profoundly?
- Do the authors think that there is a global hypermethylation effect? If true global methylation levels are desired perhaps best is to analyse genomic methylation using mass spectrometry or HPLC or next best Luminometric methylation assay (LUMA) or LINE-1 methylation assay.
- In Fig 2E, it would be interesting to also compare WT versus RSC for DEGs; this would give weight to the argument that the rescue cell line (in a IF1-KO background) is an adequate control that mimics WT conditions.
- “We also stably decreased the $\Delta\Psi_m$ of IF1-KO cells genetically by re-introducing IF1 (Fig. 3E)”. The authors should acknowledge that their rescue cell line is depolarising the mitochondrial membrane, i.e. the $\Delta\Psi_m$ is significantly lower in the RSC cell line compared to the WT, and explain the caveats associated with this reconstituted cell line (for example, Fig S2A shows that IF1 levels in the RSC cell line are ~6-fold higher than in the WT). In effect, the IF1-KO versus RSC comparison is measuring hyperpolarisation versus depolarisation.
- Although found to be statistically insignificant, the authors do observe an increase in mitochondrial perimeter in the IF1-KO cell lines compared to the controls (Fig S2E), which could still be biologically significant. Furthermore, it would be interesting to explore how increased $\Delta\Psi_m$ influences mitophagy (e.g. using fluorescent tags such as mt-Keima to monitor mitophagic flux), as this could be a biologically important consequence of increased $\Delta\Psi_m$ that is unexplored in this paper. This is especially interesting given that depolarisation has been linked to mitophagy (as discussed by the authors in the introduction).
- It would be valuable to supplement the RNA-seq data with proteomics; this would confirm which transcriptional changes are having an effect at the protein level in the IF1-KO cell line. However, this is only a suggestion.
- Related to repression by DNA methylation, the representation in Figure 3A as log fold change of DNA methylation is perhaps not the best. Drawing a direct relationship between DNA methylation and gene expression is also not always linear. Moreover, a difference from 3% DNA methylation to 7% (or 1% to 4%) measured by EPIC (which doesn't have the best sensitivity within this range) will surely not have such a significant effect on gene expression within the cell population. In fact, the threshold should be increased to 10% for differentially methylated CpGs (not “mean β value >0 or <0 ”, as currently listed in the methods). Can the authors display, at least in the supplementary data, scatterplots of the EPIC array either the full data or specifically the 4 clusters (WT vs IF1-KO) and colour code downregulated and upregulated genes as a different way of looking at the data?
- Is there a particular reason to not discuss Cluster 1 genes? To get a better feel for where DNA methylation changes, can the authors show pie charts of the different genomic elements as represented in the EPIC array (promoter, exon, intron etc) and how it changes in the WT and IF1-KO? This would display better DNA methylation re-distribution.
- “We then profiled nuclear DNA methylation at a single nucleotide resolution;” Do they mean EPIC? Usually WGBS is referred to single nucleotide resolution. EPIC analyses CpGs but it only represents ~3% of the genome.
- The authors should quantify the levels of DNMTs and TETs in Fig S4A to back up the claim that they see “no changes” in their levels. By eye, the levels of DNMT3B are lower in the IF1-KO cell line compared to the WT control.
- “Strikingly, KO cells with a lower $\Delta\Psi_m$ either through IF1 reintroduction or UCP4 ectopic expression had genome wide TSS methylation levels that were close to those observed in the WT counterparts (Fig. 3G, Fig. S3D).” Representing as a violin plot is disrupting the pairwise comparison. Are the ones gaining methylation in IF1-KO the same that lose methylation in the rescue samples? The labels in Fig S3D are unclear. What is the exact meaning of “somewhat”, “not full”, “no” – are these judgements made based on quantitative thresholds? Add legend in Fig 4A to explain the meaning of the colours in the bar chart.
- Could the authors comment on the inability of the rescue cell line to return cofactor levels back to those seen in WT cells in Fig 4B, and what this means about the suitability of their rescue cell line as a control? Similarly, in Fig 4C, the analysis should be expanded to include their RSC cell line using the data from Fig 4B. This is currently not addressed in the text either.
- The plot in Fig 4G highlights the inability of the RSC cell line to recapitulate the WT phenotype with regards to phospholipid levels; this should be addressed and discussed in the text. Furthermore, the PC/PE ratios shown in Fig 4H show similar values for the WT and RSC cell lines, however Fig 4G shows a remarkable difference between the WT and RSC cell lines; it is unclear how similar ratios for these two cell lines have been calculated based on this data.
- “Remarkably, concomitant to the decreased labeling of PC (Fig. 4J) were increases in labeling of 5-meC (Fig. 4K).” This is quite cool but don't really get it. Rerouting methyl groups in the one-carbon metabolism is one thing and de-novo methylation is another. The latter occurs if DNMTs are recruited to sites which have lower methylation levels. In other words, if a locus does not recruit DNMTs, no additional methyl donor amount would change methylation level.
- “Like in the KO cells, this chronic rise in resting $\Delta\Psi_m$ did not lead to mitochondrial dysfunction as judged by lack of effects

on oxygen consumption and extracellular acidification (Fig. 5C and D).” Disagree – Fig 5D shows statistically significant increases in several mitochondrial parameters with telmisartan.

- “analysis of phospholipids showed that the levels of PC were decreased in the treated cells affecting the cellular PC/PE ratio (Fig. 5G), and that the nuclear DNA was hypermethylated (Fig. 5H).” The effect is really small and also the median of the control seems higher than the WT in Fig 3G. Are these the same cell type?
- For Fig 6B, having the PE data is critical for the authors’ conclusion that high $\Delta\Psi$ triggers a lower PC/PE ratio. While lower PC is seen in the upper 5% of cells, PE could also be reduced in this population. The experiment should be repeated with more material to capture both PE and PC levels. If this is not possible the authors should give strong reasons why not.
- To support the model presented in Fig 6F, an important experiment would be to test whether disrupting PEMT activity can block PE/PC remodelling and consequently whether this alters DNA methylation, irrespective of changing $\Delta\Psi$. This would provide more evidence about the sequence of biological events triggered by elevated $\Delta\Psi$, and whether DNA methylation changes sit downstream of phospholipid remodelling in this setting.

Minor comments and edits:

- In the introduction: “These effects were observed to be dependent and independent of the ETC, but the higher $\Delta\Psi$ restored mitochondrial protein import in a mutant yeast strain¹⁵.” This sentence is unclear: how can it be both dependent and independent of the ETC, and what mutant are the authors referring to?
- It seems like different cell types were used for different experiments throughout the manuscript. Can the authors mention which cell type was used in figure legends?
- “Uncoupling through proton dissipation is a known mechanism to regulate the $\Delta\Psi$.” – missing reference(s).
- “IF1 was thought to bind to the ATP synthase to inhibit its hydrolase activity only under stress conditions” – how is stress defined here?
- “If decreases in PC levels due to a higher resting $\Delta\Psi$ mediate nuclear DNA hypermethylation in the KO cells, then chronically increasing the $\Delta\Psi$ of WT cells should affect the PC/PE ratio and hypermethylated the nuclear DNA.” Delete character in red for clarity.
- I would suggest moving Fig 3D to the supplementary, as DNP-treatment is not pursued beyond measuring its effect on $\Delta\Psi$ and appears out of place in the main figures.
- “Curiously, DML-containing nuclear-encoded mitochondrial genes enriched for regulation of the $\Delta\Psi$ in the KO cells, irrespective of the means utilized to decrease $\Delta\Psi$ (Fig. 3J).” – add the text in red for clarity.
- In Fig 5D-H, legends should be added for clarity.
- Missing Mw markers on blots in: Fig S2A & Fig S4A.
- Raw uncropped blots should be provided by the authors for full data transparency.

Reviewer #2

(Remarks to the Author)

In this study, Mori et al established a genetic model of mitochondrial hyperpolarization with increased mitochondrial membrane potential ($\Delta\Psi$). They found increased $\Delta\Psi$ was associated phospholipid remodeling, hypermethylation of nuclear DNA, and transcriptional regulation of genes involved in mitochondrial, carbohydrate, and lipid metabolism. These findings highlight an operational communication between polarization states in the mitochondria and transcriptional responses in the nucleus, revealing a potentially novel retrograde signaling pathway of great interest to the mitochondrial field. However, I have several concerns regarding the metabolic mechanisms underlying this mitochondria-to-nucleus communication.

1. The study did not clearly explain how the alteration in $\Delta\Psi$ led to changes in phosphatidylcholine (PC) and phosphatidylethanolamine (PE) content. Was this due to an effect on phosphatidylserine (PS) decarboxylation in the mitochondria as a result of hyperpolarization?
2. It remained puzzling how $\Delta\Psi$ could cause DNA hypermethylation without noticeable changes in S-adenosylmethionine (SAM) and other metabolites related to methylation and demethylation. The extensive consumption of SAM at the DNA level might explain the lack of SAM accumulation in IF1-KO cells. If this is the case, would treating IF1-KO cells with DNA methylation inhibitors result in increased cellular SAM levels?
3. The connection between $\Delta\Psi$ and DNA methylation might be mediated by metabolites such as SAM. Although total SAM levels remain similar, it might differ in distinct compartments or have different their metabolizing rates. It is essential for this study to provide some insights into this question.
4. In the tracing experiment, the relative changes do not reflect the proportional contribution to the total PC from the PE methylation pathway. It is often thought that PE methylation activity is low in non-liver cells. Please consider evaluating the contribution of PE methylation to total cellular PC.
5. Minor:

The study should present an overall change in lipid composition to fully assess the process of phospholipid remodeling in response to a rise in resting $\Delta\Psi$.

The legend of the y-axis did not show properly in Fig. 1B.

Check Fig. S1C. It looks like that the band of EMRE is positioned upside down.

On Page 12, “A premise of this hypothesis is that PC methylation should be decreased while...” should be PE methylation.

There are missing callouts for Fig. 5E-F.

Reviewer #3

(Remarks to the Author)

This manuscript by Mori et al inspects the effects of mitochondrial membrane hyperpolarization in a number of human cell models. Notably, they find that mitochondrial hyperpolarization leads to epigenetic changes, specifically hypermethylation, of the nuclear genome, as well as considerable transcriptional changes that include downregulation of mitochondrial genes and upregulation of phospholipid and glucose metabolism. Moreover, the authors report phospholipid remodeling in all tested models with increased mitochondrial membrane potential, and suggest this remodeling may link the membrane potential changes to the observed epigenetic alterations. The manuscript is highly interesting, compelling and significant to researchers interested in mitochondrial biology, nuclear epigenetics, signaling and/or cellular adaptations. The conclusions are largely supported by the experimental data. However, although the results convincingly show decreased levels of phosphatidylcholine (PC) as well as hypermethylation in multiple models with increased mitochondrial hyperpolarization, it should be noted that none of the data show causality, or even temporal precedence, of the lower PC levels relative to genome hypermethylation. This caveat should be made clear in both the abstract and the discussion, or the knowledge gap at least partially addressed with the experiment(s) suggested below. With that said, the presented data is novel and interesting and in my opinion warrants publication.

Major comments:

-The data in Fig. 1a suggest that the ATPase activity of ATP synthase has a considerable impact on mitochondrial membrane potential (MMP) even in cells grown under normal conditions, which is somewhat surprising. At the same time, MMP determination by TMRE is sensitive to artefacts induced during handling. To confirm the validity of the MMP measurements, please provide control measurements of TMRE intensity from WT and IF1-ko cells following full uncoupling vs no treatment.

-Fig. 1C, right panels. While ATP5O runs above the indicated 480 kDa marker line, IF1 runs somewhat lower. Please show the marker bands, rather than just their annotation, to be able to judge whether the two bands truly run at the same height.

-The so-called rescue (RSC) is in fact not only restoration of endogenous levels of IF1 but a 6-fold overexpression (Fig. S2A), which results in considerable drop in MMP to below 0.5-fold of wt levels (Fig. 3E) and massive transcriptional changes. To clearly indicate this fact, it should be labelled “IF1-OE” rather than “RSC” throughout the manuscript.

-p9, 1st paragraph. The sentence “Collectively, these data suggest that the differential nuclear DNA methylation regulated the transcriptional response to the increased resting delta-psi in IF1-ko cells” should be rephrased to indicate that the differential DNA methylation can only account for a part of the transcriptional response. Similarly, on p10, line 4, the phrase “Collectively, these data strongly support the hypothesis that the increased delta-psi is the upstream signal regulating the epigenetic landscape and the transcriptional output of the KO cells” must be rephrased to indicate that the regulation is only observed at a subset of genes (the ones the authors decided to focus on in Fig 3).

-p9, 3rd line from the bottom: the statement that the methylation changes “were observed at specific loci, including those involved in mitochondrial and phospholipid genes” should be supported by data; for example by including GO terms corresponding to the genes that reverse in TSS methylation status in Fig 3H in RSC or UCP4 cells. Similarly, it would be beneficial to indicate in the supplemental the GO terms or functional annotations of the cluster 1-4 genes shown in Fig 3A.

-Fig. 3F: please indicate which clone of the IF1-ko UCP4 cells was used to allow for comparison with the protein levels in Fig S2B.

-Fig 4: Please also show the GSSG/GSH ratio for RSC cells (Fig 4C). For the NADP(H) content, it would be informative to also plot the NADPH/NADP+ ratio in Fig 4E.

-p10, first line: the sentence stating that the ectopic expression of UCP4 “essentially mirrors” the effects of IF1 re-introduction should be rephrased to reflect that the normalization or reversal is only partial.

- Please describe why the chemicals in Fig 6a were chosen from the Tox21 dataset. If for example some others were tested but did not cause an MMP increase as expected, this would be valuable information for other researchers in the field. Also please in the legend indicate the treatment time for the experiment in Fig. 6A, and include the cell viability data following chronic exposure to at least telmisartan and anatto.

-p. 18, first sentence of the second paragraph. “Our methionine tracing data point to the PE to PC methylation reaction as decreased, implying that PEMT activity might be sensitive to delta-psi”. An alternative, or additional, explanation that should be mentioned is that the PE to PC reaction is affected by the transcriptional changes in lipid metabolism genes shown in this study.

On a related note, and most important for the revision of the paper is that as such, the data presented here do not show any causal relationship between the PE/PC ratio and the methylation state of nuclear DNA. The authors explain why PEMT overexpression is not a feasible approach to experimentally prove causality. In absence of a better approach, a time course

experiment with MMP-increasing drugs where MMP as well as the expression level and TSS methylation status of some known target genes (chosen from the genome-wide omics data) could help show the temporal order of events and support the presented working model where PE/PC remodeling is an early event rather than a consequence of the extensive transcriptional changes seen in mt-hyperpolarized cells.

-Are protein levels of PEMT affected in IF1-KO or RSC cells?

-Fig. 6B: please show PtdCho also in the lower 5% of cells.

Minor comments:

-The supplemental table files should be labelled S1, S2 and S3, as cited in the text.

-p4, first paragraph. The sentence "These effects were observed to be dependent and independent of the ETC.." is unclear.

-Fig. 1B some symbols in the y-axis labels don't display correctly.

-The key/legend in many of the heatmaps (2a, 2h, 3a, 3h-j..) does not display any color.

-p9, second paragraph and Fig. 3D. I would suggest to move the DNP data to the supplemental, since it is not used for addressing methylation.

-Fig 4A: color code (black for wt, red of ko, blue for RSC) should be provided in figure legend as it is not in the graph.

-p17, 5th sentence of the Discussion: "Even though metabolic rewiring was observed, it did not involve classic metabolites previously associated with DNA methylation but rather those associated with redox homeostasis, including through the pentose phosphate pathway". Which PPP metabolites do you mean? NADP(H), others?

-Fig. 3H: please describe in the legend what the dotted rectangle represents.

-p35, the first sentence of "RNA isolation, RNA-Seq and gene expression analysis" needs to be edited for clarity.

-Legend to Fig. S2. The panel C description is erroneously labelled as D).

Reviewer #4

(Remarks to the Author)

Reviewer #5

(Remarks to the Author)

The manuscript by Mori and colleagues details how the electrochemical status (hyperpolarisation) of mitochondria may regulate gene expression through epigenetic processes and in particular how phospholipid remodelling potentially regulates this process. I found this an intriguing pathway in part because it requires regulation across the cell, moving from mitochondria to cell membranes (and indeed potentially the plasma cell membrane) to the nucleus to exert its effects. The authors build up this mechanism in part through transcriptomics and sequencing to follow the epigenetic changes, along with knockout and rescue cell lines. This data is challenging to interpret as there are a lot of other pathways that are impacted, making it difficult to judge whether or not this is a downstream event. However, the authors follow-up their sequencing with more targeted approaches. I particularly found the PC labelling studies to follow labelling into DNA (cytosine) particularly compelling.

I have some specific points that I would like the authors to address.

Page 3: the authors provide a good introduction to IF1 (ATPIF1-ATP5IF1) but I wonder if there are any common SNPs found in these genes and whether they tell us anything about functional consequences.

Page 6: There's a typo in "...synthase monomers as judge by antibodies against subunit ATP5O...." Should be 'judged.'

Page 6. In the section for the chronic loss of IF6, the authors state that over 6000 genes are altered. Are we really just following the impact of hyperpolarisation or could the cells have growth differences? This is a large number of genes - are we really comparing like with like?

Page 6. "Genes associated with glycolysis were upregulated in the IF1-KO cells (Table S1), consistent with unrestrained hydrolysis of glycolytic ATP contributing to the maintenance of a higher resting $\Delta\Psi_m$." I am confused by this statement. Are they suggesting that mitochondrial function regulates glycolysis? Unconstrained glycolysis is regulated by trehalose in yeast

and there is regulation between the glycolysis and the TCA cycle. If glycolysis is unconstrained there are serious issues for cell function.

Page 10 "We did not find changes in the abundance of metabolites that might influence DNMT or TET activity, including SAM, 5'-deoxy-5'-methylthioadenosine (5'-MTA), α -KG or its competitive inhibitors succinate and fumarate (Fig. 4A) through whole cell steady-state metabolomics analysis." Isn't this surprising given the large number of transcriptional changes detected?

Page 12. "...the two major phospholipid components of membranes, including that of mitochondria, were reversibly altered between the KO and RSC cells (Fig. 4G)." It would be useful to understand which cell membrane is most involved in this mechanism. As the authors have already isolated mitochondria could they also not check the ratio of PC to PE to understand whether changes are occurring at mitochondria or at the plasma membrane?

Page 16: For the TMRE fluorescence experiments how was the data normalised?

Figure 3: I am probably missing where this is defined but what is the dashed box in sub-panel A?

Figure 4 legend: There is a typo for "...(mito, probe is tarted to the mitochondria)."

Page 37: "5x10⁶ cells were plates in 100 mm dishes and incubated at 37 °C and 5% CO₂ for 6 h to allow cells to attach."
Typo – should be plated.

Page 39 "WT and IF1-KO cells were grown for 4 (four) days in DMEM..." There's no need to define 4!

Figure 2A: What does this show and does it mean that pretty much everything is changed transcriptionally? Does this make sense that so many pathways have changed given the metabolomics detects so few samples?

Figure 2E: how do the authors justify that actually most genes are not in common in terms of comparing the two genetic modifications?

Figure 4B: why does the RSC mutant look more like the KO than the WT?

Version 1:

Reviewer comments:

Reviewer #1

(Remarks to the Author)

The authors have addressed the majority of the points raised and overall the data highlights an interesting link between mitochondrial polarisation and DNA methylation. However we recommend the following changes to make the manuscript suitable for publication:

1. It is interesting that the authors do not see increases in PC content after PEMT-OE (Figure 4N) yet see increases in DNA methylation in this cell line (Figure 4M). This indicates that changes in the levels of DNA methylation are perhaps not being driven by altered PC/PE ratios. It would be useful to have data on PE levels in this system, however the kit has been discontinued as highlighted by the authors. The authors instead use oxygen consumption as a proxy for PE level changes, however observe no changes, implying that PE levels are also not altered by PEMT-OE. In this case, it is unclear why DNA methylation is changing if it is dependent on altered PE/PC ratios (see working model in Figure 6H). Given this, the authors should highlight the limitations of their working model (Figure 6H) and recognise that DNA methylation may be altered through an alternative mechanism than PE/PC ratios. This should also be discussed in the text.
2. Page 11/12: "As no significant changes at the transcription (Table S1) or protein levels of DNMT, TETs or TDG". This sentence should be revisited as the authors show in Figure S5A that DNMT1 levels in the IF1-KO cell line are indeed statistically significantly increased compared to the control. This is interesting as previous reports (<https://doi.org/10.1038/s41594-021-00603-8>) have linked DNMT1 to de novo methylation activity at TE sites, which could be driving the increase DNA methylation observed in Figure 3F. The authors should discuss this.

Minor comments:

1. Page 15: "We found that the epigenome hypermethylated by overexpression of PEMT (Fig. 5M)"; this should be Fig. 4M?
2. Page 16: "Loss of PE in mitochondria has been shown to cause mitochondrial dysfunction(45 and references therein)"; the reference should be formatted correctly to match the rest of the text.
3. Figure 3A: I recommend updating the colours used in the heatmap so that different palettes are used for the mRNA and 5mC readings. This will make the heatmap easier to understand.
4. Figure S1C: Are the p.values for MCU and MICU1 missing decimals? The accompanying text on page 5 should read: "we found no significant changes at the protein levels...". There are changes, just not statistically significant ones.
5. Figure S1D: The accompanying text on page 5 should read: "we also did not find significant changes in total or phosphorylated AMPK". There are changes, just not statistically significant ones.
6. Figure S5L: add legend title.
7. Figure 4M: add legend title.
8. Figure 5D-H: add legends.

9. Figure S2A: add Mw markers to blot.

Reviewer #2

(Remarks to the Author)

The authors have adequately addressed my previous concerns. The data and conclusions are clearly presented and supported. I recommend the manuscript for publication.

Reviewer #3

(Remarks to the Author)

The authors have satisfactorily addressed my comments and concerns. Congratulations on an interesting study.

Reviewer #4

(Remarks to the Author)

Reviewer #5

(Remarks to the Author)

I thank the authors for addressing the comments I made in my previous review. On the whole they have addressed the majority of my concerns. While I am still a little confused why so many transcriptional changes result in a relatively small number of metabolomic changes, I am happy with the other responses. This is an interesting paper and should raise a lot of interest in those interested in mitochondrial function and its potential role in epigenetic regulation.

Reviewer #1 (Remarks to the Author)

Mori et al: The authors use IF1-KO cell lines to model the impact of chronic mitochondrial hyperpolarisation. Their system is shown to significantly enhance mitochondrial membrane potential ($\Delta\Psi M$), providing a tool to analyse this phenomenon in cells. The authors perform bulk RNA-Seq in their IF1-KO cell line, identifying downregulation of mitochondrial and respiration genes, and upregulation of phospholipid-modifying and glucose metabolism genes compared to their control cells. In addition, the authors identify significant alterations in DNA methylation profiles during chronic mitochondrial hyperpolarization. Using lipidomics and methyl tracing experiments, the authors identify reductions in PE:PC ratios in cells with elevated $\Delta\Psi M$, with concomitant increases in methylation of cytosines. The authors demonstrate that similar effects can be observed in cells exposed to chemical compounds and in cancer cells that weakly express IF1. Overall, the authors propose a model whereby elevated $\Delta\Psi M$ disrupts PE:PC remodelling, rerouting methyl groups for DNA methylation and altering gene expression. The findings are of significant interest, however, as presented, some of the data do not support the claims made in the manuscript. We hope our comments and suggestions will clarify specific points and improve the manuscript.

We appreciate the Reviewer's insightful suggestions and positive comments.

Comments:

1) It is an interesting finding that mitochondrial membrane depolarization can signal down to the level of chromatin and change the DNA methylation landscape; it is remarkable to see promoter methylation gain. Such an association between high TMRE/MTG and global methylation has been observed before in human pluripotent stem cells and reprogramming to naïve pluripotency (PMID: 28843285). However, in this system, high TMRE/MTG was associated primarily with widespread global demethylation, but also promoter CpG island methylation or CGIs (promoter CGIs are normally not methylated). Since in this manuscript the focus is specifically on the effect of membrane potential on hypermethylation, how do the authors explain the different effects in the two experimental systems?

Response: We thank the Reviewer for bringing the paper of Takashima et al to our attention. Please note that because of different assay strategies (dequenching, our study vs self-quenching, theirs), high TMRE was interpreted as opposing changes of the mitochondrial membrane potential. Takashima et al associated mitochondrial depolarization (decreased membrane potential) with widespread demethylation, whereas we provided evidence that increased membrane potential (hyperpolarization) resulted in general hypermethylation. Both Takashima et al and we showed locus-specific methylation changes going in opposite directions to the global pattern. We are thus pleased that the conclusions of their studies are complementary with ours.

2) Is this indicating that the effect is not direct and there is an intermediate mechanism that mediates the effect on DNA methylation? If so, would this be cell type specific?

Response: We would argue that in both cases the data indicate an intermediate. Our results strongly suggest that the effects observed involve re-routing of methyl groups from PE methylation to DNA. As PE methylation is driven by the enzyme phosphatidylethanolamine N-methyltransferase (PEMT), our data suggest that the activity of this enzyme is modulated by hyperpolarization of the mitochondrial membrane. While we do not know yet how this might take place, it is notable that PEMT resides on the interface between the ER and mitochondrial membranes, and we have shown that other proteins are

affected by changes in that interface between the two organelles (PMID: 36351901). Thus, it is feasible to speculate that the increased mitochondrial membrane potential modulates PEMT enzymatic activity by changing the local redox environment or by, for example, trapping the enzyme in areas of the ER-mitochondrial contact sites. Interestingly, we found that PEMT has up to 3 putative redox sensitive cysteines, making it feasible that it is redox regulated using a publicly available prediction software. It is also important to point out that modulation of PEMT activity might be a response to or a cause of compensation to changes in the Kennedy pathway, the major source of PC in most cells. Significant feedback regulation has been shown to occur between these two pathways (PMID:24184426), and our RNA-seq suggests upregulation of several genes involved in PC metabolism through the Kennedy reactions (see **new data in revised Fig.S5N**). These possibilities were included in the discussion (**Page 22**). Whether this is applicable to all cells is too early to speculate, and likely depend on the extent to which cells use PEMT as a significant contributor for PC production. As per below, liver and kidney cells have high PEMT transcript levels and, at least the liver, is known to produce significant levels of PC through this enzyme (PMID: 22877991).

3) “Unlike previously reported (16), we found no changes at the protein levels of the different subunits of the mitochondrial Ca²⁺ uniporter complex (mtCU)” – 1 out of 3 of the IF1-KO lysates show elevated levels; it is too strong to say “no changes at the protein levels”. Some statistics on the quantifications shown in Fig S1C would help the authors make such quantitative statements here.

Response: As per the request, we have added statistics, which confirm the lack of significant changes, to the revised Fig. S1C.

4) “we also did not find changes in total or phosphorylated AMPK”. This statement goes against the quantification shown in Fig S1D, where p-AMPK in the IF1-KO cells are evidently higher than in the WT cells. Although, I am confused by the quantification, which shows an increase in p-AMPK (up by more than 1.5-fold in one of the repeats), while the blot does not show such an increase by eye.

Response: As per the request, we have re-quantified the images and re-plotted the graphs, including statistics (revised Fig. S1D). The graphs now reflect the total amount of AMPK relative to actin, and the levels of AMPK-P relative to the total amount of actin-normalized AMPK.

5) “These data are consistent with a fraction of IF1 being bound to complex V in WT cells grown under standard cell culture conditions.” The BN-PAGE experiment in Fig 1C is insufficient evidence to suggest that IF1 is “bound”, as is it labelled on the blot. Moreover, ATP5O is migrating above 480 kDa whereas IF1 appears to migrate below the 480 kDa marker based on the blot shown. To say that IF1 is “bound”, the authors would need to perform a pulldown assay for ATP5O and confirm that IF1 is co-precipitated

(or similar assay). Based on this BN-PAGE blot, I would argue that the authors could, at most, say that IF1 is “co-migrating” rather than “bound”.

Response: We apologize for the confusion and have included additional data and revised the text accordingly (Page 6). Blue-native-PAGE (BN-PAGE) is not intended to show protein-protein interactions, but to identify protein complexes in their native state. In our case, the antibody for ATP50 (α denotes sensitive to oligomycin), which is a subunit of the F_0 peripheral stalk, was employed to visualize the ATP synthase complex. The holoenzyme can be present as dimers or monomers and is composed of 18 subunits – plus or minus IF1. As such, antibodies against any of these subunits can be used to identify the ATP synthase complex. To further make this point, in the **new Fig. 1C and Fig. S1E**, we used antibodies against the ATP synthase c-ring and ATP8 (mtDNA-encoded ATP synthase subunit 8), both of which are part of the F_0 rotor. We also used antibodies against ATPB (ATP synthase F1 subunit β) that is part of the F_1 catalytic head and thus of another component of the holoenzyme. Irrespective of the antibody used, the new data show that the BN-PAGE results are identical - IF1 is not present on the ATP synthase complex in the KO cells. Since IF1 is not stable when not bound to the ATP synthase, the conclusion of these experiments is correct. The **revised text on page 6** now reads: “Using blue native gels and antibodies against subunit ATP50 (Fig. 1C) or ATP8, ATPB or ATP c-ring subunit (Fig. S1C) to visualize the holoenzyme, we found that IF1 was detected in WT cells, but not KO cells.”.

6) “Collectively, these data support the conclusion that the increased resting $\Delta\Psi_M$ in the KO cells relies on a significant contribution of ATP hydrolysis under standard cell culture conditions.” Have the authors confirmed that ATP hydrolysis is indeed altered in their model when cultured in glucose compared to galactose (see Fig 1D)? This should be measurable using a colorimetric assay, for example. It is too speculative to say that “By limiting the pool of glycolytic ATP that can be hydrolyzed by the ATP synthase, growth in galactose would be predicted to decrease $\Delta\Psi_M$.” – this prediction needs to be validated. This is also interesting to confirm given the reported roles of IF1 in inhibiting ATP synthase activity (discussed by the authors earlier in the paragraph).

Response: The Reviewer raises an excellent point. To address this, we assayed ATP hydrolysis through in-gel assays. Briefly, mitochondria were isolated from WT and IF1-KO cells, run on a BN-PAGE and then incubated for 16 h with buffer containing 35 mM Tris, 270 mM glycine, 14 mM $MgSO_4$, 0.2% $Pb(NO_3)_2$ and 10 mM ATP at pH 7.8. Subsequently, the gel was fixed with 50% methanol and 10% acetic acid. The reaction between inorganic phosphate and lead nitrate forms lead phosphate. The inorganic phosphate is produced by ATP hydrolysis, which is catalyzed by Complex V when working in reverse mode. The lead phosphate then accumulates on the enzyme's band, which can be visualized on the KO samples but not on the WT counterparts (**new Fig. 1D**). Text has also been added to the manuscript (**new text on Page 6**) to report on these new experiments.

7) Figure 2D indicates enrichment of KRAB domain genes in the RNA-seq data. This is relevant as these are transcription factors (TFs) and can recruit co-repressors (including DNMTs) to the promoters. This could indicate that histone regulation (heterochromatin formation by recruitment of HP1 by KRAB TFs for example) could explain the gene expression changes. A subset of genes might be directly repressed by DNA methylation. I appreciate that the authors have looked at histones in WBs (Figure S5) – however, retargeting of repressive histone marks to different genomic locations would not be readily visible in a WB global analysis. ChIP-seq or Cut&Run of specific repressive histone marks (H3K9me2/3 for example) would be more informative. As such, this interpretation might not be correct: “these data suggest that the methylation of the DNA, but not of histones, is primarily affected by a decrease in the cellular PC/PE ratio in mammalian cells”

Response: We appreciate the comment of the Reviewer and addressed this possibility using *in silico* approaches. To define whether retargeting of the repressive histone marks H3K9me3 played a significant role in influencing the gene expression changes identified, we analyzed the expression of transposable elements (TEs). TE loci are transcriptionally repressed due to their heterochromatic state that is largely set by the presence of H3K9me3 (PMID: 35338361). Additionally, a large fraction of KRAB-ZFP target sites is located within TEs (PMID: 31540910 and references therein). Hence, analysis of TE expression as a proxy of heterochromatin remodeling through the involvement of KRAB is an appropriate means to address the concern of the Reviewer. As previously we showed that TE expression and the generation of fusion transcripts was increased with loss of H3K9me3 in a model of cocaine exposure (PMID: 27415830), we surmised that if H3K9me3 would be retargeted in IF1-KO cells, then expression of TEs would be broadly changed in the KO versus the WT controls. We used RepeatMasker to map reads to TEs, restricting the analysis to intergenic regions to avoid confounding signals from co-transcription of protein-coding genes. We analyzed ~4 million TE sequences classified into the main classes LINE, SINE, long terminal repeats (LTRs), and DNA transposons; we also performed the analysis at the subfamily level, as recently described (PMID: 31745090). We found minor changes in TE expression at the class (table below) or at the subfamily levels (**new Table S4**), collectively suggesting that heterochromatin changes are a minor effect associated with chronic loss of IF1.

ID	baseMean	log2FoldChange	lfcSE	stat	pvalue	Fold_Change	padj
DNA	47939.73092	-0.120553116	0.1005093	-1.199422503	0.230363702	-1.087151587	0.488119579
DNA?	104.1134772	-0.362376219	0.571926355	-0.633606435	0.5263377	-1.285541531	0.780530805
LINE	314216.1652	-0.508285683	0.103040878	-4.932854712	8.10E-07	-1.422359039	9.79E-06
LTR	217638.6838	0.151472283	0.09812679	1.543638412	0.122675954	1.110702375	0.319452544
LTR?	436.6014444	-0.315079761	0.288534263	-1.0920012	0.274832574	-1.244080432	0.550252245
SINE	559258.5053	1.082504535	0.10150385	10.66466484	1.49E-26	2.11770926	1.01E-24
SINE?	0.325139217	2.099047145	4.984204173	0.42113988	0.673652942	4.284263294	0.880121092

As H3K9me3 can repress genes encoding silencing factors and be present at gene body and promoter regions, also affecting gene expression (PMID: 31540910 and references therein), we next asked whether promoter H3K9me3 peak coordinates identified in HEK293 cells through the ENCODE Project coincided with those of the differentially expressed genes (DEGs). The ENCODE Project reported 3,788 H3K9me3 promoter peaks in this cellular background. We determined average H3K9me3 promoter peak width based on the publicly available ENCODE data, then overlapped their coordinates (+/-1.5Kb) with those of the promoters of the DEGs. We found that 93 out of the 6,553 (0.01%) DEGs overlapped with H3K9me3 peak coordinates (**new Table S4**). Given over 50% of DEGs had the promoters differentially methylated, we conclude that heterochromatin changes and/or H3K9me3 retargeting have little, if any, influence on the transcriptional changes driven by chronic loss of IF1. These new data are included and discussed in the revised manuscript (**Page 21**).

8) *"In agreement, we found that the mitochondria of WT and KO cells could further hyperpolarize when they were put under phosphate depleted conditions (Fig. S2F), recapitulating the effects recently reported for other cells¹⁵" this is interesting...perhaps IF1 is not the only inhibitor of ATP synthase? And that in fact IF1 has other roles in the signalling network? Have the authors checked DNA methylation in this condition of hyperpolarisation? According to their model, this should further modify the DNA methylation landscape, perhaps even more profoundly?*

Response: The $\Delta\Psi_m$ is maintained by the flux of protons through the electron transport chain, it is also maintained by ETC-independent mechanisms including ATP hydrolysis by reverse rotation of the F1

subunit of complex V, the exchange of ATP/ADP by the adenine nucleotide translocator (ANT), the exchange of ions such as Na⁺, K⁺ or Ca²⁺ for H⁺ through different ion transporters on the mitochondrial membrane, the opening of the mPTP (mitochondrial permeability transition pore) as well as through uncoupling proteins. These mechanisms operate independently of each other, and in different cellular contexts. Phosphate depletion for 72 h was shown to hyperpolarize the mitochondrial membrane both by modulating ETC function as well as by altering other mechanisms, although the exact process was not identified (PMID: 38251707). While it is possible that further hyperpolarization of the mitochondrial membrane might impact DNA methylation more, testing this in the context of phosphate depletion is challenging given the confounding effects associated with the many processes in the cells that are phosphate-dependent (including protein phosphorylation) and that are impacted by its removal. Also, at least in our hands, loss of cell viability becomes evident at 72h of phosphate depletion. These significant technical challenges hamper the unequivocal interpretation of such experiments, and we believe will not adequately address this interesting question.

9) Do the authors think that there is a global hypermethylation effect? If true global methylation levels are desired perhaps best is to analyse genomic methylation using mass spectrometry or HPLC or next best Luminometric methylation assay (LUMA) or LINE-1 methylation assay.

Response: We would like to clarify what we mean by global phenotype: changes in DNA methylation when considering the different genomic regions that are covered by the Illumina array, including promoters, 5' and 3' UTRs, gene bodies and intergenic regions. Based on this criterion, average DNA methylation values was overall increased in the KO cells (see Table below) although both hyper and hypomethylation changes were identified at the locus level (Table S2). Estimates suggest that only 1-5% of the genome is amenable to changes in DNA methylation after the landscape is set early in development, findings that we have corroborated previously in the mouse liver using WGBS (PMID:

Means	Feature	ko.mean	wt.mean	diff.mean	p.value	effect.size	alternative
	1stExon	0.5351703	0.453751	0.0814192	4.20E-29	0.2760179	two.sided
	3'UTR	0.6948496	0.6256567	0.0691929	7.60E-41	0.2665405	two.sided
	5'UTR	0.5862812	0.518112	0.0681692	2.55E-83	0.2355359	two.sided
	Body	0.6393725	0.5754339	0.0639386	0	0.2283931	two.sided
	ExonBnd	0.7254582	0.6674032	0.058055	4.43E-08	0.2328262	two.sided
	IGR	0.5862575	0.5325107	0.0537468	1.74E-216	0.1880556	two.sided
	TSS1500	0.5470964	0.4772081	0.0698883	1.90E-143	0.237614	two.sided
	TSS200	0.5227778	0.4383732	0.0844046	2.00E-74	0.2866383	two.sided
	All DMPs	0.5982983	0.5354602	0.0628381	< 2.2e-16	0.2180837	one.sided (KO>WT)

32937126). Thus, we use the term global to refer to what is captured by the Illumina 850K array, which in turn reflects the areas of the genome that are amenable to changes. To avoid confusion to the reader,

we have revised the text (**Page 9**) and added a **new Fig. S3B** to the manuscript showing all genomic regions where methylation was evaluated.

10) *In Fig 2E, it would be interesting to also compare WT versus RSC for DEGs; this would give weight to the argument that the rescue cell line (in a IF1-KO background) is an adequate control that mimics WT conditions.*

Response: We have now added the data (**revised Table S1**) and revised the text accordingly (**Page 8**).

11) *“We also stably decreased the $\Delta\Psi$ M of IF1-KO cells genetically by re-introducing IF1 (Fig. 3E)”. The authors should acknowledge that their rescue cell line is depolarising the mitochondrial membrane, i.e. the $\Delta\Psi$ M is significantly lower in the RSC cell line compared to the WT, and explain the caveats associated with this reconstituted cell line (for example, Fig S2A shows that IF1 levels in the RSC cell line are ~6-fold higher than in the WT). In effect, the IF1-KO versus RSC comparison is measuring hyperpolarisation versus depolarisation.*

Response: In writing the manuscript, we have debated on whether calling the rescue as such, or as an overexpression isogenic line. Based on comments from this and Reviewer 3, we have revised the text and replaced RSC for OE (for overexpression). As requested, we discussed the caveats in the use of this model as a rescue given earlier data, at least in cancer cells, of the effects of IF1 overexpression (see also response to point 19 below). Those revisions appear throughout the revised manuscript.

12) *Although found to be statistically insignificant, the authors do observe an increase in mitochondrial perimeter in the IF1-KO cell lines compared to the controls (Fig S2E), which could still be biologically significant. Furthermore, it would be interesting to explore how increased $\Delta\Psi$ M influences mitophagy (e.g. using fluorescent tags such as mt-Keima to monitor mitophagic flux), as this could be a biologically important consequence of increased $\Delta\Psi$ M that is unexplored in this paper. This is especially interesting given that depolarisation has been linked to mitophagy (as discussed by the authors in the introduction).*

Response: To address whether the IF1KO cells can undergo mitophagy despite their hyperpolarized state, WT, KO and OE cells were treated with FCCP, which fully uncouples mitochondria and is regularly used as a positive control to induce mitophagy. Western blots probing LC3-II cleavage, which is used to report induction of mitophagy, were performed. All cells equally showed LC3-II cleavage, indicating proficiency in the ability to undergo mitophagy. These data have been included in this revised version as **new Fig. S2F**. The revised text is presented on **Page 9**.

13) *It would be valuable to supplement the RNA-seq data with proteomics; this would confirm which transcriptional changes are having an effect at the protein level in the IF1-KO cell line. However, this is only a suggestion.*

Response: We agree that proteomics can help further understand the impact of the transcriptional changes identified in this study. However, in the context of epigenetic outcomes, determining the transcriptional output seems more informative. Given the time frame and cost to perform proteomics, and the comment that this was a suggestion, we opted not to include it in this revision.

14) *Related to repression by DNA methylation, the representation in Figure 3A as log fold change of DNA methylation is perhaps not the best. Drawing a direct relationship between DNA methylation and gene expression is also not always linear. Moreover, a difference from 3% DNA methylation to 7% (or 1% to 4%) measured by EPIC (which doesn't have the best sensitivity within this range) will surely not have such a significant effect on gene expression within the cell population. In fact, the threshold should be increased to 10% for differentially methylated CpGs (not “mean β value >0 or <0 ”, as currently listed in*

the methods). Can the authors display, at least in the supplementary data, scatterplots of the EPIC array either the full data or specifically the 4 clusters (WT vs IF1-KO) and colour code downregulated and upregulated genes as a different way of looking at the data?

Response: We thank the Reviewer for the comment; while some loci have low differences in the level of methylation based on the initial analysis, we still found that most loci had $\Delta\beta$ values ranging from +/-10-80% (Table S2). Nevertheless, we followed the suggestion of the Reviewer and re-analyzed the data establishing a threshold of at least 10% difference. By doing so, the overall genome-wide differences increased (**revised Fig. 3F**), and the interpretation of the data did not change despite the expected decrease in the number of differentially methylated genes, as judged by promoter methylation. We have updated the heatmap, which we believe allows easy visualization of both patterns, and generated the scatter plots as requested, color coding their expression (up or down regulated; **new Fig. S3C**).

15) Is there a particular reason to not discuss Cluster 1 genes? To get a better feel for where DNA methylation changes, can the authors show pie charts of the different genomic elements as represented in the EPIC array (promoter, exon, intron etc) and how it changes in the WT and IF1-KO? This would display better DNA methylation re-distribution.

Response: We did not discuss the genes from cluster 1 or 4 because those have the DNA hypermethylated and genes upregulated or vice-versa. As indicated in the paper, for simplistic purposes we chose to focus on those that the directionality and DNA methylation would be consistent (**Page 10**) based on the criterion we applied. We have included in this revision the pathways enriched from genes on clusters 1 and 4 (**revised Table S2**). As per the request, pie charts of the different genomic regions and the respective changes in DNA methylation the cells were added (**new Fig. S3B**).

16) The authors should quantify the levels of DNMTs and TETs in Fig S4A to back up the claim that they see “no changes” in their levels. By eye, the levels of DNMT3B are lower in the IF1-KO cell line compared to the WT control.

Response: We have re-done the immunoblots, now showing the membranes with 3 independent replicates; per the request, we also added the bar graphs (**revised Fig. S5A**). Only DNMT1 levels were found to be statistically increased, but since this enzyme is involved in DNA methylation maintenance and not *de novo* methylation, the statement made on **page 12** remains correct: “As no significant changes at the transcription (Table S1) or protein levels of DNMT, TETs or TDG (Fig. S5A) **that could explain the DNA hypermethylation phenotype were observed...**”.

17) “Strikingly, KO cells with a lower $\Delta\psi M$ either through IF1 reintroduction or UCP4 ectopic expression had genome wide TSS methylation levels that were close to those observed in the WT counterparts (Fig. 3G, Fig. S3D).” Representing as a violin plot is disrupting the pairwise comparison. Are the ones gaining methylation in IF1-KO the same that lose methylation in the rescue samples?

Response: The violin plots depict the average probe signals across genome-wide TSSs, whether they overlap or not with the promoters of genes that are also differentially expressed. Data on Table S3 reported on pair-wise comparisons at the locus level, and while the heatmaps on Fig. 3G were restricted in showing the same loci that reversed in both the UCP and KO cells, we now have added heatmaps depicting all changes that reverse, at the locus level, for each comparison (KO vs OE and KO vs UCP4 - **new Fig. S4E**). In this new figure, each line represents the exact same locus across the genotypes.

18) *The labels in Fig S3D are unclear. What is the exact meaning of “somewhat”, “not full”, “no” – are these judgements made based on quantitative thresholds? Add legend in Fig 4A to explain the meaning of the colours in the bar chart.*

Response: we agree with the Reviewer that this is confusing. For clarity, we have removed those labels keeping only the numbers of loci that have been fully, partially or not rescued. The color of the bars was included as requested, with black representing WT, red the KO and blue the OE, which is consistent throughout the manuscript.

19) *Could the authors comment on the inability of the rescue cell line to return cofactor levels back to those seen in WT cells in Fig 4B, and what this means about the suitability of their rescue cell line as a control?*

Response: IF1 has been shown to inhibit ATP synthesis when overexpressed, effects that have been mostly studied in different cancer cells (PMID: 37568591). Our data show that under the same experimental conditions that the metabolomics analysis was done, that is, cells grown in glucose-rich media, basal and ATP-driven respiration were significantly decreased in the OE cells relative to the WT cells (Fig. S2C). Based on the data from the current manuscript and previous work from us and others (PMID: 26725009, PMID: 27307216, PMID: 26924217 AND PMID: 29668680), we interpret these data to indicate that increased ATP hydrolysis and membrane hyperpolarization, as well as loss of ATP synthesis and the associated depolarization, lead to (some) common metabolic phenotypes. These are, nevertheless, initiated by distinct stimuli and serve different purposes in that cell-specific context.

20) *Similarly, in Fig 4C, the analysis should be expanded to include their RSC cell line using the data from Fig 4B. This is currently not addressed in the text either.*

Response: the data has been included per the request.

21) *The plot in Fig 4G highlights the inability of the RSC cell line to recapitulate the WT phenotype with regards to phospholipid levels; this should be addressed and discussed in the text. Furthermore, the PC/PE ratios shown in Fig 4H show similar values for the WT and RSC cell lines, however Fig 4G shows a remarkable difference between the WT and RSC cell lines; it is unclear how similar ratios for these two cell lines have been calculated based on this data.*

Response: The data on the original Fig. 4G depicted absolute values of a few phospholipids that were captured using metabolomics, which is not an ideal technique to detect and differentiate between phospholipids heads (these data have now been moved to Fig. S5C-E). We have generated new data determining total and mitochondrial levels of PE and PC using high performance thin layer chromatography (HPTLC), which is a highly quantitative and suitable method to detect specific phospholipids. Using this approach, we were able to quantify the total or mitochondria levels of PC or PE accurately. The HPTLC data confirmed that total levels of PC were decreased in the KO cells, but were not different between the OE and WT, whether in whole cells or in isolated mitochondrial membranes (**new Fig. 4G**). This approach also established that there were no significant increases in PE (**new Fig. 4H**). Given the ratio of PC and PE is calculated based on the absolute values of these phospholipids in each cell line, the data also confirmed a decreased in the PC/PE ratio in the KO relative to wild type controls, which reversed in the OE (**new Fig. 4I and representative HPTLC gel on Fig. S5F**).

23) *“Remarkably, concomitant to the decreased labeling of PC (Fig. 4J) were increases in labeling of 5-mC (Fig. 4K),” This is quite cool but don't really get it. Rerouting methyl groups in the one-carbon metabolism is one thing and de-novo methylation is another. The latter occurs if DNMTs are recruited to*

sites which have lower methylation levels. In other words, if a locus does not recruit DNMTs, no additional methyl donor amount would change methylation level.

Response: Our data show that the total amount of methyl donor, SAM, was not significantly changed between the phenotypes under our experimental conditions (Fig. 4A). However, the methyl tracing experiments strongly suggest that SAM is being re-routed from PE methylation into cytosines (Fig. 4J-L). Based on these results, we conclude that SAM from PE-PC reaction is re-routed onto the DNA, where it can be recycled through the action of *de novo* DNMTs, as previously described to histone methyltransferases in yeast. Why some genomic loci were affected while others were not is currently unknown, but we believe beyond the scope of this manuscript. We also invite the Reviewer to read Reviewer 2, comment #3, and our response to it (below).

24) *“Like in the KO cells, this chronic rise in resting $\Delta\Psi_M$ did not lead to mitochondrial dysfunction as judged by lack of effects on oxygen consumption and extracellular acidification (Fig. 5C and D).” Disagree – Fig 5D shows statistically significant increases in several mitochondrial parameters with telmisartan.*

Response: Some aspects of respiration were higher with telmisartan, which is inconsistent with mitochondria dysfunction. We have revised the sentence above to read: “Like in the KO cells, this chronic rise in resting $\Delta\Psi_M$ did not lead to mitochondrial dysfunction **as judged by loss** in oxygen consumption or **increased** extracellular acidification (Fig. 5C and D).”

25) *“analysis of phospholipids showed that the levels of PC were decreased in the treated cells affecting the cellular PC/PE ratio (Fig. 5G), and that the nuclear DNA was hypermethylated (Fig. 5H).” The effect is really small and also the median of the control seems higher than the WT in Fig 3G. Are these the same cell type?*

Response: The effect on DNA hypermethylation was smaller when cells were treated for 10 days with the chemicals, relative to the effects upon chronic genetic ablation of IF1; this is likely due to the timing associated with the epigenetic/phospholipid adaptation. We note that by applying the 10% threshold for the DNA methylation analysis as suggested by this Reviewer, changes in DNA methylation upon the treatments were more pronounced (see **revised figure 5H**). Controls were cells exposed to the vehicle (DMSO), which was utilized to solubilize the drugs; DMSO was not present in other experiments.

26) *For Fig 6B, having the PE data is critical for the authors’ conclusion that high $\Delta\Psi_M$ triggers a lower PC/PE ratio. While lower PC is seen in the upper 5% of cells, PE could also be reduced in this population. The experiment should be repeated with more material to capture both PE and PC levels. If this is not possible the authors should give strong reasons why not.*

Response: The Reviewer raises a good point, but we think it is unlikely given that loss in PE leads to mitochondrial dysfunction by changing properties of the inner membrane that negatively impact the ETC (PMID: 23250747; PMID: 22971339). Unfortunately, the limited number of cells sorted precluded us from performing lipidomics or HPTLC, and the PE kit that we previously used has been discontinued. In trying to get some clues about potential PE decreases in the cells, we analyzed mitochondrial function as a proxy. Analysis of oxygen consumption using the Seahorse Flux Analyzer in the three different sorted populations revealed that only the 5% depolarized cells had impaired mitochondrial function (**new Fig. 6C and D**). Thus, while indirect, these data coupled to the lack of effects on PE in all other hyperpolarized models lead us to assume that PE levels are not diminished in the 5% hyperpolarized sorted cells.

27) *To support the model presented in Fig 6F, an important experiment would be to test whether inhibiting PEMT activity can block PE/PC remodelling and consequently whether this alters DNA*

methylation, irrespective of changing $\Delta\Psi_M$. This would provide more evidence about the sequence of biological events triggered by elevated $\Delta\Psi_M$, and whether DNA methylation changes sit downstream of phospholipid remodelling in this setting.

Response: as noted in the original submitted manuscript, these experiments are challenging given the significant feedback regulation between the Kennedy pathway, which produces PC in the ER, and PEMT activity - which produces PC from PE methylation in the interface between the ER and mitochondria. In this context, it is well established that modulation of PEMT leads to compensation by the Kennedy pathway, overall causing no net changes in total PC (PMID: 24184426). If this holds under our experimental conditions, then we would expect no impact on the PC/PE ratio or on the epigenome. Based on this comment and a similar suggestion from Reviewer 3 (point 9), we went ahead and overexpressed PEMT in the KO cells. In our hand, the antibodies for PEMT proved to be unspecific, which led us to generate a PEMT-flag tagged construct whose overexpression was confirmed using immunoblots (**new Fig. S5K**). We then performed DNA methylation analysis using the Illumina array and found that the DNA methylation landscape did not reverse in these cells; only 320 TSS loci were significantly different between the KO and the PEMT-overexpressing counterparts, and they were mostly further hypermethylated (**new Figure 4M and Table S3**). We then determined total PC levels, and found no changes upon overexpression of PEMT, whether in the WT or IF1-KO cells (**new Fig. 4N**). Because the PE kit has been discontinued but because loss of mitochondrial PE leads to mitochondrial dysfunction (see comment above), we determined oxygen consumption as a proxy for changes in PE. No differences in mitochondrial function as gauged by respiration rates were identified as a function of PEMT overexpression in any of the genotypes (**new Fig. 4O**) collectively indicating that the PC/PE ratio is critical in impacting the methylation status of the nuclear genome under our experimental conditions.

Minor comments and edits: - **all minor comments and edits have been addressed.**

28) *In the introduction: “These effects were observed to be dependent and independent of the ETC, but the higher $\Delta\Psi_M$ restored mitochondrial protein import in a mutant yeast strain¹⁵.” This sentence is unclear: how can it be both dependent and independent of the ETC, and what mutant are the authors referring to?*

Response: See response above to point 8; regardless, we have revised the sentence for clarity.

29) *It seems like different cell types were used for different experiments throughout the manuscript. Can the authors mention which cell type was used in figure legends?*

Response: All cells are isogenic derivatives of the WT cells and thus in the HEK293 background. The ovarian cancer cells are the exception. Figure legends have been updated as requested.

30) *“Uncoupling through proton dissipation is a known mechanism to regulate the $\Delta\Psi_M$.” – missing reference(s).*

Response: the reference has been added

31) *“IF1 was thought to bind to the ATP synthase to inhibit its hydrolase activity only under stress conditions” – how is stress defined here?*

Response: In the context of ATP hydrolysis, stress is defined as ischemia or oxygen deprivation as well as ETC inhibition. This has been now included in the text.

32) *“If decreases in PC levels due to a higher resting $\Delta\Psi_M$ mediate nuclear DNA hypermethylation in the KO cells, then chronically increasing the $\Delta\Psi_M$ of WT cells should affect the PC/PE ratio and hypermethylated the nuclear DNA.” Delete character in red for clarity.*

Response: Deleted as requested

33) *I would suggest moving Fig 3D to the supplementary, as DNP-treatment is not pursued beyond measuring its effect on $\Delta\Psi_M$ and appears out of place in the main figures.*

Response: This figure has been moved to the supplement, as requested, it is now Fig. S4A.

34) *“Curiously, DML-containing nuclear-encoded mitochondrial genes enriched for regulation of the $\Delta\Psi_M$ in the KO cells, irrespective of the means utilized to decrease $\Delta\Psi_M$ (Fig. 3J).” – add the text in red for clarity.*

Response: text has been added

35) *In Fig 5D-H, legends should be added for clarity.*

Response: Added

36) *Missing Mw markers on blots in: Fig S2A & Fig S4A.*

Response: added

37) *Raw uncropped blots should be provided by the authors for full data transparency.*

Response: Data has been provided, can be found at the end of the supplemental figures.

Reviewer #2 (Remarks to the Author)

In this study, Mori et al established a genetic model of mitochondrial hyperpolarization with increased mitochondrial membrane potential ($\Delta\Psi_M$). They found increased $\Delta\Psi_M$ was associated phospholipid remodeling, hypermethylation of nuclear DNA, and transcriptional regulation of genes involved in mitochondrial, carbohydrate, and lipid metabolism. These findings highlight an operational communication between polarization states in the mitochondria and transcriptional responses in the nucleus, revealing a potentially novel retrograde signaling pathway of great interest to the mitochondrial field. However, I have several concerns regarding the metabolic mechanisms underlying this mitochondria-to-nucleus communication.

We thank the Reviewer for the constructive criticism and positive comments.

1. The study did not clearly explain how the alteration in $\Delta\Psi_M$ led to changes in phosphatidylcholine (PC) and phosphatidylethanolamine (PE) content. Was this due to an effect on phosphatidylserine (PS) decarboxylation in the mitochondria as a result of hyperpolarization?

Response: In response to this comment, we have analyzed total and mitochondrial levels of PE, PC and PS using high performance thin layer chromatography (HPTLC), which allows for quantitative analysis of phospholipids. The data from this approach confirmed that while PC levels were decreased in the KO cells, there were no changes in the levels of PE or decreases in the levels of PS, which were rather increased in both whole cells and in isolated mitochondria membranes (**new figure S5F and G**). Increased PS is also consistent with the upregulation of phosphatidylserine synthase 2 transcript – the enzyme that synthesizes PS from PE (Table S1). Given PE levels were not significantly different, and our carbon tracing experiments, we conclude that the decreased levels of PC that are associated with mitochondrial

hyperpolarization result from decreased activity of phosphatidylethanolamine-N-transferase (PEMT) – the enzyme that methylates PE into PC. Why the phospholipid changes occur are still unclear, but we believe that their relative composition on the mitochondrial membrane can facilitate regulation of proton flux and thus allow maintenance of a chronic hyperpolarized state (see our model in Fig. 6H).

2. It remained puzzling how $\Delta\Psi_M$ could cause DNA hypermethylation without noticeable changes in S-adenosylmethionine (SAM) and other metabolites related to methylation and demethylation. The extensive consumption of SAM at the DNA level might explain the lack of SAM accumulation in IF1-KO cells. If this is the case, would treating IF1-KO cells with DNA methylation inhibitors result in increased cellular SAM levels?

Response: The Reviewer brings up an interesting point, but we think that it is unlikely that inhibition of DNMTs alone would allow accumulation of SAM based on the various levels of regulation of the methyl cycle – since it is critical in maintaining cellular homeostasis (PMID: 15809266). Changes in SAM are expected to set in motion a cascade of reactions to adjust methyl group donation and homocysteine recycling to maintain the cycle running. For example, SAM serves as an allosteric activator of cystathionine β -synthase (CBS), an enzyme that promotes the transsulfuration pathway when SAM levels are elevated. This ensures the recycling of SAM into redox reactions, including through the generation of glutathione (GSH), preventing hypermethylation. Similarly, the availability of cofactors such as folate (vitamin B9), vitamin B12, and vitamin B6 is critical for maintaining the proper flow through the SAM cycle. Deficiencies in any of these co-factors can impair methionine re-methylation and lead to elevated homocysteine levels. Also, the inhibition of methyltransferases, including DNMTs, by the SAM byproduct SAH (S-adenosyl-homocysteine) serves as an intrinsic feedback mechanism to prevent excessive methylation, ensuring that SAM-dependent reactions occur in a controlled manner. With this level of control and regulation, it is likely that cells will compensate exogenous DNMT inhibition by adjusting other reactions. Another concern with this approach to increase SAM is the fact that DNMT inhibitors, such as 5'-aza-deoxycytidine and its derivatives, cause DNA-protein crosslinks (PMID: 12154409; PMID: 22349820; PMID: 23609537, PMID: 28655905; PMID: 36681662), which in turn activate DNA repair pathways that themselves can consume SAM (PMID: 39651281).

3. The connection between $\Delta\Psi_M$ and DNA methylation might be mediated by metabolites such as SAM. Although total SAM levels remain similar, it might differ in distinct compartments or have different their metabolizing rates. It is essential for this study to provide some insights into this question.

Response: we had not considered this possibility, and we thank the Reviewer for bringing it up. However, we believe that the lack of differences in our metabolomics data, which provides measurements of steady state levels, is consistent with the reactions involved in maintaining the methyl cycle (see response to point above) adapting to maintain cellular homeostasis in the KO cells. In terms of compartmentalization, SAM is primarily generated in the cytosol, from where it is available for the many reactions that utilize it, including in the nucleus (PMID: 23425511). A smaller pool of SAM is present in the mitochondria (PMID: 14674884). The mitochondrial solute carrier 25A26 (SLC2A26) imports cytosol-synthesized SAM in exchange for matrix SAH; while disease mutations on this gene exist and have been shown to cause mitochondrial dysfunction (PMID: 32340404), there has been no reported effects on the cytosolic levels of SAM. Importantly, decreases in mitochondrial SAM were shown to impair OXPHOS assembly and induce complex I instability (PMID: 33608280). Conceptually, a decrease in mitochondrial SAM could increase the cytosolic pool in the KO cells, providing the means to differently impact the epigenome without altering its total levels. While this is possible, we think this is unlikely for several reasons: (1) the lack of mitochondrial dysfunction in the KO cells (Fig. S2B and C) would be inconsistent

with a decrease in the mitochondrial SAM pool; (2) SLCs are known to be regulated transcriptionally, based on substrate availability or membrane fluidity (PMID: 32810346). Our RNA-seq results show that while ~100 genes coding for SLCs were differentially expressed between the WT and the KO cells, SLC25A26 was not among them (Table S1); (3) SAM is not limiting in the cytosol of KO cells (as per metabolomics) and thus should not impact the transporter function of SLC25A26 and (4) PC is the major phospholipid present in mitochondrial membrane, including the inner membrane where ETC and SLCs are present. While its decrease can potentially alter membrane fluidity, the lack of OXPHOS dysfunction in the KO cells is inconsistent with the PC changes observed impacting the function of proteins that are embedded in inner mitochondrial membrane. We have included this in the revised manuscript (**Page 12**).

4. In the tracing experiment, the relative changes do not reflect the proportional contribution to the total PC from the PE methylation pathway. It is often thought that PE methylation activity is low in non-liver cells. Please consider evaluating the contribution of PE methylation to total cellular PC.

Response: This is another excellent comment. We refer the Reviewer to the responses to point 2 of Reviewer 1 (above) and to point 9 of Reviewer 3 (below).

5. Minor – **all issues have been addressed as requested.**

The legend of the y-axis did not show properly in Fig. 1B.

Response: this has been corrected.

Check Fig. S1C. It looks like that the band of EMRE is positioned upside down.

Response: Blots were checked, and the band is correct – we refer to the non-cropped blots now shown at the end of the Supplemental Material.

On Page 12, “A premise of this hypothesis is that PC methylation should be decreased while...” should be PE methylation.

Response: thank you for noticing it, we have fixed the error.

There are missing callouts for Fig. 5E-F. – F

Response: this has been also fixed.

Reviewer #3 (Remarks to the Author)

This manuscript by Mori et al inspects the effects of mitochondrial membrane hyperpolarization in a number of human cell models. Notably, they find that mitochondrial hyperpolarization leads to epigenetic changes, specifically hypermethylation, of the nuclear genome, as well as considerable transcriptional changes that include downregulation of mitochondrial genes and upregulation of phospholipid and glucose metabolism. Moreover, the authors report phospholipid remodeling in all tested models with increased mitochondrial membrane potential and suggest this remodeling may link the membrane potential changes to the observed epigenetic alterations. The manuscript is highly interesting, compelling and significant to researchers interested in mitochondrial biology, nuclear epigenetics, signaling and/or cellular adaptations. The conclusions are largely supported by the experimental data. However, although the results convincingly show decreased levels of phosphatidylcholine (PC) as well as hypermethylation in multiple models with increased mitochondrial hyperpolarization, it should be noted that none of the data show causality, or even temporal precedence, of the lower PC levels relative to genome hypermethylation. This caveat should be made clear in both the abstract and the discussion, or the

knowledge gap at least partially addressed with the experiment(s) suggested below. With that said, the presented data is novel and interesting and in my opinion warrants publication.

We appreciate the Reviewer's thoughtful suggestions and favorable overall opinion.

Major comments:

1) -The data in Fig. 1a suggest that the ATPase activity of ATP synthase has a considerable impact on mitochondrial membrane potential (MMP) even in cells grown under normal conditions, which is somewhat surprising. At the same time, MMP determination by TMRE is sensitive to artefacts induced during handling. To confirm the validity of the MMP measurements, please provide control measurements of TMRE intensity from WT and IF1-ko cells following full uncoupling vs no treatment.

Response: we thank the comment, and while we have added the data as requested (Fig. S1B), we also call attention for data on Fig. 1B, which used TMRM in permeabilized cells and FCCP.

2) -Fig. 1C, right panels. While ATP5O runs above the indicated 480 kDa marker line, IF1 runs somewhat lower. Please show the marker bands, rather than just their annotation, to be able to judge whether the two bands truly run at the same height.

Response: Differently than SDS-PAGE/immunoblot, NativeMark™, the molecular weight marker used of blue native PAGE/immunoblot is not detected after transfer onto the PVDF membrane. However, to address the Reviewer's concern, we have stained the gel with Colloidal blue staining (ThermoFisher Scientific, cat # LC6025) and superimposed the gel and the membrane to get an estimate of the molecular weight marker on the membrane (**revised Fig. 1C and D**). We have also added data using antibodies against other subunits of the ATP synthase to confirm that the results are consistent independent of the antibody tested. The **new data** has been added as **Fig. S1E**.

3)-The so-called rescue (RSC) is in fact not only restoration of endogenous levels of IF1 but a 6-fold overexpression (Fig. S2A), which results in considerable drop in MMP to below 0.5-fold of wt levels (Fig. 3E) and massive transcriptional changes. To clearly indicate this fact, it should be labelled "IF1-OE" rather than "RSC" throughout the manuscript.

Response: as per the request of the Reviewer, the change has been made throughout the manuscript. We invite the Reviewer to also see the response to point 11 from Reviewer 1.

4) -p9, 1st paragraph. The sentence "Collectively, these data suggest that the differential nuclear DNA methylation regulated the transcriptional response to the increased resting delta-psi in IF1-ko cells" should be rephrased to indicate that the differential DNA methylation can only account for a part of the transcriptional response. Similarly, on p10, line 4, the phrase "Collectively, these data strongly support the hypothesis that the increased delta-psi is the upstream signal regulating the epigenetic landscape and the transcriptional output of the KO cells" must be rephrased to indicate that the regulation is only observed at a subset of genes (the ones the authors decided to focus on in Fig 3).

Response: this has been changed as per the request, now on Pages 10 and 11, respectively.

5) -p9, 3rd line from the bottom: the statement that the methylation changes "were observed at specific loci, including those involved in mitochondrial and phospholipid genes" should be supported by data; for example by including GO terms corresponding to the genes that reverse in TSS methylation status in Fig 3H in RSC or UCP4 cells. Similarly, it would be beneficial to indicate in the supplemental the GO terms or functional annotations of the cluster 1-4 genes shown in Fig 3A.

Response: the loci that were differentially methylated in the KO and reversed in either the OE or UCP4 were shown in Table S3 (mitochondrial and phospholipid genes in bold) and are now also presented as a heatmap (**new Fig. S4E**), which displays the DNA methylation status of the same locus (each line) across the different genotypes. As per the request, the GO terms corresponding to clusters 1-4 from Figure 3A are now present in the **revised Table S2**.

6) -Fig. 3F: please indicate which clone of the IF1-ko UCP4 cells was used to allow for comparison with the protein levels in Fig S2B.

Response: clone B1 was the one selected and utilized throughout the manuscript.

7) -Fig 4: Please also show the GSSG/GSH ratio for RSC cells (Fig 4C). For the NADP(H) content, it would be informative to also plot the NADPH/NADP+ ratio in Fig 4E.

Response: data has been added as per the request.

8) -p10, first line: the sentence stating that the ectopic expression of UCP4 “essentially mirrors” the effects of IF1 re-introduction should be rephrased to reflect that the normalization or reversal is only partial.

Response: revised as per the request, now on page 11.

- Please describe why the chemicals in Fig 6a were chosen from the Tox21 dataset. If for example some others were tested but did not cause an MMP increase as expected, this would be valuable information for other researchers in the field. Also please in the legend indicate the treatment time for the experiment in Fig. 6A, and include the cell viability data following chronic exposure to at least telmisartan and annatto.

Response: We have included in the text, **page 16**, the reasoning behind selecting telmisartan (a blood pressure medication) and annatto (a food additive), which can provide a physiological setting where the membrane potential might be chronically elevated in humans. As requested, we have added a table with the chemicals that were tested and the extent to which our analysis recapitulated or not the data from Tox21, both when chemicals were tested acutely or chronically (**new Fig. S6A**). We have revised the legends as requested and included viability data (**new Fig S6B**).

9) -p. 18, first sentence of the second paragraph. “Our methionine tracing data point to the PE to PC methylation reaction as decreased, implying that PEMT activity might be sensitive to delta-psi”. An alternative, or additional, explanation that should be mentioned is that the PE to PC reaction is affected by the transcriptional changes in lipid metabolism genes shown in this study.

Response: This is an excellent comment, particularly the possibility that changes in PEMT activity might be responding to the transcriptional changes in phospholipids. As mentioned above (see response to point 27 of Reviewer 1) it has been well established that there is significant compensation between the two main pathways to generate PC, the Kennedy - which occurs in the ER - and PEMT, which methylates PE in the interface between the mitochondria and ER membranes. In looking at our RNA-seq data, we found that the metabolism of PC through reactions associated with the Kennedy pathway were indeed modulated in the KO cells and the changes are reversed in the OE counterparts (**revised Fig. S5M**). Thus, while these data suggest feedback regulation between the two pathways, it is unclear whether the latter is the cause of, or a consequence to, the decreased PEMT activity. We have included this both in the results (**Page 15**) and in the discussion (**Page 22**).

On a related note, and most important for the revision of the paper is that as such, the data presented here do not show any causal relationship between the PE/PC ratio and the methylation state of nuclear

DNA. The authors explain why PEMT overexpression is not a feasible approach to experimentally prove causality. In absence of a better approach, a time course experiment with MMP-increasing drugs where MMP as well as the expression level and TSS methylation status of some known target genes (chosen from the genome-wide omics data) could help show the temporal order of events and support the presented working model where PE/PC remodeling is an early event rather than a consequence of the extensive transcriptional changes seen in mt-hyperpolarized cells.

Response: we refer the Reviewer to our answer to point 27 from Reviewer 1, who had a similar comment. Despite the knowledge of compensation between the Kennedy and PEMT pathways to generate PC, and our RNA-seq data (see comment above), we did try to perform the requested experiment that included overexpressing PEMT. We note that DNA methylation was not reversed, but neither were the levels of PC increased (**new Fig. 4M and N**). Also, mitochondrial function was not changed in the PEMT-overexpressing cells (**new Fig. 4O**), which would be inconsistent with decreases in mitochondrial PE. Incidentally, we think that these results help shed light into the question raised by this Reviewer, as they suggest that it is indeed the PC/PE ratio that influences the epigenetic status of the cells under our experimental conditions, although we note that we identified cysteines that might be redox regulated in PEMT. If oxidation renders the enzyme inhibited, this could also explain the lack of reversal on PC and the epigenome as, presumably, the ectopically expressed protein would be oxidized in the IF1-KO hyperpolarized background. We have added this to the discussion as well.

10) -Are protein levels of PEMT affected in IF1-KO or RSC cells?

Response: the accurate answer to this question is that we do not know; commercially available antibodies against PEMT are extremely unspecific, at least in our hands. Dr Mori (first author of this manuscript) met Dr Rene Jacobs, who studies PEMT and has generated antibodies against the protein, in a conference. Dr Jacobs then sent us the antibodies, but in our hands, it was very unspecific. Even using cell KO for PEMT (which we could not confirm with the antibodies), using mouse liver and kidney lysates as positive controls (see Fig below), we could not interpret the data. Thus, this is question that remains un-addressed. Given no changes in the transcription of PEMT (Table S1), which is a known means of its regulation, we believe its amounts are not different among the cells.

-Fig. 6B: please show PtdCho also in the lower 5% of cells.

Response: data has been included. The PC content in those cells was decreased but, unlike the other sorted cells, the depolarized population had dysfunctional mitochondria (**new Fig. 6B-D**). We note that decreases in PC do not affect ETC function while decreases in PE or cardiolipin do (PMID: 39178855). Thus, these data suggest that there is no linear relationship between the levels of PC *alone* and the degree of polarization of the mitochondrial membrane, which is likely influenced by the relative abundance of different phospholipids, including PC, PE and potentially cardiolipin.

Minor comments: all minor comments have been addressed as requested

The supplemental table files should be labelled S1, S2 and S3, as cited in the text.

Response: this has been revised as per the request.

-p4, first paragraph. The sentence "These effects were observed to be dependent and independent of the ETC.." is unclear.

Response: We agree that it sounded dubious, thus we have modified the phrase for clarity.

-Fig. 1B some symbols in the y-axis labels don't display correctly.

Response: we have revised the manuscript to assure all labels display properly

-The key/legend in many of the heatmaps (2a, 2h, 3a, 3h-j..) does not display any color.

Response: we have revised the manuscript to assure all labels display properly;

-p9, second paragraph and Fig. 3D. I would suggest to move the DNP data to the supplemental, since it is not used for addressing methylation.

Response: data has been moved and is now Fig. S4A.

-Fig 4A: color code (black for wt, red of ko, blue for RSC) should be provided in figure legend as it is not in the graph.

Response: Color coding was added.

-p17, 5th sentence of the Discussion: "Even though metabolic rewiring was observed, it did not involve classic metabolites previously associated with DNA methylation but rather those associated with redox homeostasis, including through the pentose phosphate pathway". Which PPP metabolites do you mean? NADP(H), others?

Response: yes, primarily the redox pair NADP(H)

-p35, the first sentence of "RNA isolation, RNA-Seq and gene expression analysis" needs to be edited for clarity.

Response: Edited as requested.

-Legend to Fig. S2. The panel C description is erroneously labelled as D).

Response: we have fixed the legend as per the comment above.

Reviewer #4 (Remarks to the Author)

I co-reviewed this manuscript with one of the Reviewers who provided the listed reports. This is part of the Nature Communications initiative to facilitate training in peer review and to provide appropriate recognition for Early Career Researchers who co-review manuscripts.

Reviewer #5 (Remarks to the Author):

The manuscript by Mori and colleagues details how the electrochemical status (hyperpolarisation) of mitochondria may regulate gene expression through epigenetic processes and in particular how phospholipid remodelling potentially regulates this process. I found this an intriguing pathway in part because it requires regulation across the cell, moving from mitochondria to cell membranes (and indeed potentially the plasma cell membrane) to the nucleus to exert its effects. The authors build up this mechanism in part through transcriptomics and sequencing to follow the epigenetic changes, along with knockout and rescue cell lines. This data is challenging to interpret as there are a lot of other pathways that are impacted, making it difficult to judge whether or not this is a downstream event. However, the authors follow-up their sequencing with more targeted approaches. I particularly found the PC labelling studies to follow labelling into DNA (cytosine) particularly compelling.

We thank the Reviewer for raising important points and for the positive comments.

I have some specific points that I would like the authors to address.

1) Page 3: the authors provide a good introduction to IF1 (ATPIF1-ATP5IF1) but I wonder if there are any common SNPs found in these genes and whether they tell us anything about functional consequences.

Response: This is an interesting point, according to the reference genome, there are 3 known isoforms of ATPIF1, which differ primarily in their size given alternative use of 5' or 3' UTRs or splice sites. Thus, 3 protein isoforms are generated: i) isoform 1 (NP_057395.1; 12.2 kDa), ii) isoform 2 (NP_835497.1; 7.9 kDa), and iii) isoform 3 (NP_835498.1; 6.6 kDa). Only isoform 1 has been shown to be functional, at least in inhibiting complex V. As for polymorphism, to our knowledge only one missense variant SNP causing an A-G transition (AGG)>(GGG) that results in a change from arginine to glycine has been reported, but without functional consequences.

2) Page 6: There's a typo in "...synthase monomers as judge by antibodies against subunit ATP50..." Should be 'judged.'

Response: We thank the Reviewer and corrected the typo.

3) Page 6. In the section for the chronic loss of IF6, the authors state that over 6000 genes are altered. Are we really just following the impact of hyperpolarisation or could the cells have growth differences? This is a large number of genes - are we really comparing like with like?

Response: The population doubling of the cells is not statistically different although they are not the same (19.7h for the KO and 20.2h for the WT). Such a small difference in doubling time (~30 min) is unlikely underlying the transcriptional changes.

We were also surprised with the number of differential expressed genes given the cells were not submitted to any type of stress. However, we also note that such a large number of genes might reflect how we analyze the data which, unlike others, do not rely on a simple cut off based on 2-fold changes to define a gene as differentially expressed. Instead, in all our studies we use the entire dataset to evaluate data dispersion and estimate the effect size, which helps defining what signal is considered true relative to (and above) the background noise. Then, we apply p-values and adjusted p-values as the thresholds. By doing so, we believe we have a better ability to let the data guide further analysis, rather than assume that biological relevance is only present with a fold change arbitrarily set to >1.5- 2-fold.

We find this approach more appropriate as instead of identifying less genes with higher fold differences, it usually allows identification of several genes within the same pathway (for example, 30% of all ETC genes in this dataset) which may have smaller changes individually but are more likely to be biologically relevant given the engagement of several players in the same pathway.

4) Page 6. *“Genes associated with glycolysis were upregulated in the IF1-KO cells (Table S1), consistent with unrestrained hydrolysis of glycolytic ATP contributing to the maintenance of a higher resting $\Delta\Psi_M$.” I am confused by this statement. Are they suggesting that mitochondrial function regulates glycolysis? Unconstrained glycolysis is regulated by trehalose in yeast and there is regulation between the glycolysis and the TCA cycle. If glycolysis is unconstrained there are serious issues for cell function.*

Response: we apologize if this statement was confusing to this Reviewer. ATP hydrolysis is unrestrained in these cells because of loss of IF1, which is known for inhibiting this specific activity. For clarity, we have revised the text to indicate “unrestrained hydrolysis of glycolytic ATP by the ATP synthase”.

5) Page 10 *“We did not find changes in the abundance of metabolites that might influence DNMT or TET activity, including SAM, 5'-deoxy-5'-methylthioadenosine (5'-MTA), α -KG or its competitive inhibitors succinate and fumarate (Fig. 4A) through whole cell steady-state metabolomics analysis.” Isn't this surprising given the large number of transcriptional changes detected?*

Response: this was not surprising given that many changes, including in TCA cycle genes, seemed compensatory. For example, while expression of pyruvate dehydrogenase decreased in the KO cells, potentially decreasing flux through the TCA cycle, the levels of glutaminase (GLS2) - a mitochondrial enzyme that provides an alternative entry point to the TCA cycle feeding into alpha-ketoglutarate (α -K) through glutamine, was 2-fold increased. Consistent with glutamine helping fuel the cycle in the KO cells, oxoglutarate dehydrogenase was upregulated, this enzyme generates succinyl-CoA and eventually succinate from α -KG. Similarly, succinate dehydrogenase, which in the TCA cycle generates fumarate from succinate, was downregulated, but so was fumarate hydratase that hydrates fumarate into malate. The net effect of such changes would be maintenance of succinate and fumarate levels. Thus, it seems that the transcriptional changes involved compensatory adaptations that were deployed to maintain proper mitochondrial function and cellular homeostasis.

6) Page 12. *“...the two major phospholipid components of membranes, including that of mitochondria, were reversibly altered between the KO and RSC cells (Fig. 4G).” It would be useful to understand which cell membrane is most involved in this mechanism. As the authors have already isolated mitochondria could they also not check the ratio of PC to PE to understand whether changes are occurring at mitochondria or at the plasma membrane?*

Response: the Reviewer raises an interesting point, and we have now added data on isolated mitochondrial membranes. We used high performance thin layer chromatography (HPTLC) for those experiments, showing that changes in whole cells and mitochondria paralleled each other (**new Fig. 4G-I**). This is consistent with a recent study that showed that changes in whole cell phospholipids reflects (or is reflected by) changes in organellar membrane phospholipids composition, including in the ER, mitochondria, Golgi and the peroxisomes (PMID: 38129691).

7) Page 16: *For the TMRE fluorescence experiments how was the data normalised?*

Response: the data from TMRE fluorescence was normalized by co-staining the cells with mitotracker green, which is primarily insensitive to changes in mitochondrial membrane potential and thus can be

used as a proxy for mitochondrial content. As the dyes fluoresce in different channels (red and green), cells were concomitantly loaded, and the ratio between the two fluorophores calculated.

8) *Figure 3: I am probably missing where this is defined but what is the dashed box in sub-panel A?*

Response: we apologize for the lack of clarity, assuming the Reviewer is referring to figure 3, panel A. The dashed boxes are highlighting genes that are hypermethylated at the DNA level and downregulated at the transcript level, as well as those that are hypomethylated at the DNA and upregulated in expression. The figure legend has been revised to clarify what the lines refer to.

9) *Figure 4 legend: There is a typo for “...(mito, probe is tarted to the mitochondria).”*

Response: We thank the Reviewer and corrected the typo.

10) *Page 37: “5×10⁶ cells were plates in 100 mm dishes and incubated at 37 °C and 5% CO₂ for 6 h to allow cells to attach.” Typo – should be plated.*

Response: We thank the Reviewer and corrected the typo.

11) *Page 39 “WT and IF1-KO cells were grown for 4 (four) days in DMEM....” There’s no need to define 4.*

Figure 2A: What does this show, and does it mean that pretty much everything is changed transcriptionally? Does this make sense that so many pathways have changed given the metabolomics detects so few samples?

Response: We refer the Reviewer to the responses to points 3 and 5 above. There are several adaptations occurring at the transcriptional level, many of which are reflected in the metabolome of the cells. However, we also believe that these data indicate that the large transcriptional responses are reflective of the chronic hyperpolarized state of the mitochondrial membrane and the broad impact that it has in cellular homeostasis. For example, it is interesting that the maintenance of a high mitochondrial membrane potential had been associated with cell cycle and cell replication, and many genes that are upregulated in the cells are involved in cell cycling (cyclins, for instance) and DNA replication (many DNA polymerases are upregulated). We were not surprised with the relatively few changes in steady state metabolism in these cells given our previous work. In an earlier study, we characterized how cells respond to step-wise and progressive loss of mitochondrial function through depletion of the mitochondrial DNA. In that study, we reported that ~ 200 metabolites changed over the course of progressive mitochondrial DNA loss, which was paralleled with increasing mitochondrial dysfunction, while over 2K genes were differentially expressed at the same time (PMID: 29668680 and PMID: 30737248). Thus, it seems that the metabolome of cells is quicker to adapt to mitochondrial changes relative to the transcriptome.

12) *Figure 2E: how do the authors justify that actually most genes are not in common in terms of comparing the two genetic modifications?*

Response: The Reviewer raises an excellent point. In addition to inhibiting the hydrolytic activity of the ATP synthase, IF1 has also been shown to inhibit ATP synthesis when overexpressed (PMID: 37568591). Immunoblots revealed that the level of ectopically expressed IF1 in the KO background were 6X higher than the endogenous levels detected in the WT controls (Fig. S2A). Thus, the ectopically expressed IF1

in these cells restored the inhibition of ATP hydrolysis, but also had effects that reflected its overexpression, including on ATP synthesis. In the WT cells, IF1 overexpression likely had minor effects on ATP hydrolysis, as the endogenous protein is sufficient to keep that activity restrained. As per the paper: “RNA-seq analysis identified over 9,000 DEGs between the KO and OE (Table S1). Of these, 4,185 were DEGs between KO vs WT and broadly enriched for mitochondria (Fig. 2D). The expression of 2,365 out of the 4,185 common genes was fully reversed by the re-introduction of IF1 (Fig. 2E)”. We interpret the changes in the

4,185 genes to be broadly a response to IF1, with the 2,365 specifically responsive to the changes in the ATP hydrolytic activity. We have added the comparison between WT and OE to the revised manuscript. Consistent with the overexpression of IF1 having effects on the ATP synthesis, pathway enrichment analysis shows that the only common pathway enriched between the WT vs OE and the KO vs OE relates to ATP synthesis (**revised Table S2**).

13) *Figure 4B: why does the RSC mutant look more like the KO than the WT?*

Response: we refer the Reviewer to our response to comment 19 from Reviewer 1.

The authors have addressed the majority of the points raised and overall the data highlights an interesting link between mitochondrial polarisation and DNA methylation. However we recommend the following changes to make the manuscript suitable for publication:

1. It is interesting that the authors do not see increases in PC content after PEMT-OE (Figure 4N) yet see increases in DNA methylation in this cell line (Figure 4M). This indicates that changes in the levels of DNA methylation are perhaps not being driven by altered PC/PE ratios. It would be useful to have data on PE levels in this system, however the kit has been discontinued as highlighted by the authors. The authors instead use oxygen consumption as a proxy for PE level changes, however observe no changes, implying that PE levels are also not altered by PEMT-OE. In this case, it is unclear why DNA methylation is changing if it is dependent on altered PE/PC ratios (see working model in Figure 6H). Given this, the authors should highlight the limitations of their working model (Figure 6H) and recognise that DNA methylation may be altered through an alternative mechanism than PE/PC ratios. This should also be discussed in the text.

Response: We appreciate the reviewer's concern but would like to clarify a few points that were apparently misconstrued based on how we reported the data. As indicated on page 15, over 8,774 TSSs were differentially methylated by loss of IF1 (Table S2) while only 320 were changed by overexpression of PEMT (Table S3). Out of these, 190 were already changed in the KOs, demonstrating that – in effect - the overexpression of PEMT only added 130 promoters to the changes already provided by IF1 deletion. Assuming, simplistically, that the genome contains 20K protein coding promoters, IF1 loss impacted 44% of genome-wide promoters, while PEMT overexpression changed an additional 0.65% promoters. The range of DNA methylation changes (or $\Delta\beta$) ranged ± 0.2 when PEMT was overexpressed (Table S3) while it varied between -0.73 to $+0.78$ by loss of IF1 alone (Table S2). Thus, we interpreted the data to indicate that constitutive overexpression of PEMT on top of IF1 loss did not result in a DNA methylation profile adjustment large enough, in magnitude (± 0.7 vs ± 0.2) or number of TSSs (8,774 vs 130), to supersede the effect of IF1 deletion alone. We have revised the text on Page 15 to make this clearer and hope the reviewer, like the other 3 reviewers, will agree with the interpretation of the data.

2. Page 11/12: "As no significant changes at the transcription (Table S1) or protein levels of DNMT, TETs or TDG". This sentence should be revisited as the authors show in Figure S5A that DNMT1 levels in the IF1-KO cell line are indeed statistically significantly increased compared to the control. This is interesting as previous reports (<https://doi.org/10.1038/s41594-021-00603-8>) have linked DNMT1 to de novo methylation activity at TE sites, which could be driving the increase DNA methylation observed in Figure 3F. The authors should discuss this.

Response: We appreciate the concern of the reviewer and thank for the comment; we are familiar with the data that DNMT1 can occasionally serve as a *de novo* methyltransferase, specifically at TE loci. However, we note that data presented in Figure 3F strictly report on DNA

methylation at transcription start sites of protein coding genes, which does not include TEs which are primarily (but not exclusively) present in intergenic regions. As far as we are aware, the Illumina array does not cover TE loci, unless if coinciding with the intergenic regions covered. Importantly, if increased DNMT1 levels explained the hypermethylation phenotype observed under our experimental conditions, one would expect its levels to be decreased when re-introducing IF1 since the DNA generally hypomethylated in those cells. Yet, as per the data below, the levels of DNMT1 were not different between the KO (red bar) and the IF1-overexpressing counterpart (blue bar). Therefore, an increase in DNMT1 protein cannot account for the DNA methylation changes at promoter regions, which is the focus of our studies, reported herein.

Minor comments:

1. Page 15: "We found that the epigenome hypermethylated by overexpression of PEMT (Fig. 5M)"; this should be Fig. 4M?

Response: we thank the reviewer for identifying this error, which was an inadvertent mistake. The sentence has now been corrected.

2. Page 16: "Loss of PE in mitochondria has been shown to cause mitochondrial dysfunction(45 and references therein)"; the reference should be formatted correctly to match the rest of the text.

Response: We have revised the text accordingly.

3. Figure 3A: I recommend updating the colours used in the heatmap so that different palettes are used for the mRNA and 5meC readings. This will make the heatmap easier to understand.

Response: As per the request, the colours for the RNA-seq data in this figure have been changed.

4. Figure S1C: Are the p.values for MCU and MICU1 missing decimals? The accompanying text on page 5 should read: “we found no significant changes at the protein levels...”. There are changes, just not statistically significant ones.

Response: We apologize that the p-values were seemingly missing decimals, the numbers are now corrected and read p-0.58 and 0.34, respectively. We respectfully disagree with the reviewer about an effect in the absence of statistical significance; by definition, the lack of statistical significance indicates that there is no effect (or that it is likely random), which is what the null hypothesis tests. As such, the sentence as written is correct.

5. Figure S1D: The accompanying text on page 5 should read: “we also did not find significant changes in total or phosphorylated AMPK”. There are changes, just not statistically significant ones.

Response: See answer above.

6. Figure S5L: add legend title. The legend for this panel was present in the revised version, the title is one for the entire Fig. S5.

7. Figure 4M: add legend title. The legend for this panel was present in the revised version; the title for the figure is one for the whole Fig. 4.

8. Figure 5D-H: add legends. Same as above

9. Figure S2A: add Mw markers to blot. – markers have been added